# GRADIENT DESCENT DYNAMICS OF RANK-ONE MATRIX DENOISING

**Zeyan Zhuang & Shenghui Song**
Department of Electronic and Computer Engineering
The Hong Kong University of Science and Technology
`zzhuangac@connect.ust.hk, eeshsong@ust.hk`

## ABSTRACT

Matrix denoising is a crucial component in machine learning, offering valuable insights into the behavior of learning algorithms (Bishop & Nasrabadi, 2006). This paper focuses on the rectangular matrix denoising problem, which involves estimating the left and right singular vectors of a rank-one matrix that is corrupted by additive noise. Traditional algorithms for this problem often exhibit high computational complexity, leading to the widespread use of gradient descent (GD)-based estimation methods with a quadratic cost function. However, the learning dynamics of these GD-based methods, particularly the analytical solutions that describe their exact trajectories, have been largely overlooked in existing literature. To fill this gap, we investigate the learning dynamics in detail, providing convergence proofs and asymptotic analysis. By leveraging tools from large random matrix theory, we derive a closed-form solution for the learning dynamics, characterized by the inner products of the estimates and the ground truth vectors. We rigorously prove the almost sure convergence of these dynamics as the signal dimensions tend to infinity. Additionally, we analyze the asymptotic behavior of the learning dynamics in the large-time limit, which aligns with the well-known Baik-Ben Arous-Péchée phase transition phenomenon (Baik et al., 2005). Experimental results support our theoretical findings, demonstrating that when the signal-to-noise ratio (SNR) surpasses a critical threshold, learning converges rapidly from an initial value close to the stationary point. In contrast, estimation becomes infeasible when the ratio of the inner products between the initial left and right vectors and their corresponding ground truth vectors reaches a specific value, which depends on both the SNR and the data dimensions.

## 1 INTRODUCTION

Matrix denoising involves recovering a signal matrix $\boldsymbol{P} \in \mathbb{R}^{p \times n}$ from a noisy observation $\boldsymbol{X} = \boldsymbol{P} + \boldsymbol{Z}$, which is a fundamental challenge in statistics with broad applications in image processing (Pedersen et al., 2009; Cordero-Grande et al., 2019), genomics (Leek, 2011), wireless communications (Couillet & Hachem, 2013), and other fields. This model is typically referred to as Johnstone's spiked model (Johnstone & Paul, 2018), when the noise $\boldsymbol{Z}$ is a random matrix whose dimensions $p$, $n$ are large and comparable, while $\boldsymbol{P}$ is a deterministic matrix with rank $r \ll \min(p, n)$. Extensive research (Ding & Yang, 2021; Couillet & Liao, 2022; Bao et al., 2022; Liu et al., 2025a) has shown that in high-dimensional settings, the left and right singular vectors of $\boldsymbol{X}$ corresponding to its top singular values could be utilized as the estimate for $\boldsymbol{P}$.

However, when the dimensions $p$ and $n$ are very large, the singular value decomposition (SVD) of $\boldsymbol{X}$ suffers from intolerable storage and computational overhead. To address this issue, iterative optimization and learning methods have been developed (Björck et al., 2015), among which gradient descent (GD) plays a crucial role not only in matrix denoising but also many machine learning problems. In particular, understanding the dynamics of GD is essential in explaining the remarkable performance of today's deep neural networks. Inspired by this, we aim to analyze the learning dynamics of the GD-based rank-one signal estimation algorithm in this work.

## 1.1 RELATED WORKS

Research on matrix denoising typically studies low-rank deformed random matrix models. Two fundamental and widely studied variants are the spiked Wigner model (Benaych-Georges & Nadakuditi, 2011) and the spiked Wishart model (Benaych-Georges & Nadakuditi, 2012). The seminal work by Baik, Ben Arous, and Péché (BBP) (Baik et al., 2005) revealed that the extreme eigenvalues and associated eigenvectors of $X$ undergo a BBP phase transition as the signal-to-noise ratio (SNR) varies. This pioneering result has since been generalized to a wide range of statistical models for signal estimation, including those with correlated noise (Zhang & Mondelli, 2024; Ding & Yang, 2021; Gavish et al., 2023), random features (Liao & Couillet, 2018), puncturing (Couillet et al., 2021), random projections (Yang et al., 2021), among others. A key commonality across these studies is the use of extreme eigenvalues and eigenvectors for signal recovery.

The aforementioned studies primarily focus on the "static" or asymptotic performance of matrix denoising. Iterative algorithms for such low-rank signal recovery problems have also been established. Under the information-theoretic framework (Korada & Macris, 2009; Lelarge & Miolane, 2017), it can be shown that approximate message passing (AMP) can achieve minimal mean-square-error, and the corresponding SNR boundary has been determined (Barbier et al., 2016; Lesieur et al., 2017). However, AMP generally requires specific prior distributions and is often tailored to particular problem settings.

Another effective iterative algorithm is GD. Unlike AMP, GD does not rely on prior information and can be applied to a broad range of problems (Pretorius et al., 2018). The dynamics of GD are also crucial for understanding machine learning processes. Nevertheless, results on the learning dynamics of GD-based estimation are very limited. The most relevant work in the literature is (Bodin & Macris, 2021), where the authors studied the rank-one matrix denoising problem for the deformed Wigner model. In particular, a closed-form solution for the learning dynamics is obtained for the case where $P$ and $Z$ are symmetric. However, the symmetric structure for the noise may not be available in practice, where the observed data typically consists of a sequence $\{x_j\}$, and the data matrix $X = [x_1, \ldots, x_n]$ is rectangular (the deformed Wishart model). Unfortunately, the overall learning dynamics for this deformed Wishart model with general structured $P$ and $Z$ have not been fully understood in the literature. This work aims to fill this research gap.

From a technical perspective, obtaining analytical solutions for the dynamical behavior of the Wishart model is significantly more challenging than that for the Wigner model, due to its structural asymmetry and the higher order of the governing differential equations. To overcome these difficulties, we employ methods from large random matrix theory (RMT) to construct approximate solutions. For a comprehensive treatment of the relevant techniques, we refer the readers to (Bai et al., 2010; Pastur & Shcherbina, 2011; Yao et al., 2015; Erdős & Yau, 2017; Vershynin, 2018; Couillet & Liao, 2022; Tao, 2023). Our approach was specifically motivated by the almost sure boundedness of extreme eigenvalues of random matrices (Yin et al., 1988; Bai & Yin, 1993) and strong convergence results for the resolvents (Bai & Silverstein, 1998; 1999).

## 1.2 CONTRIBUTIONS

The main contributions of this work are summarized as follows.

- We obtain the deterministic approximations for the evolution of the inner products between the learned vectors and the ground truth, which are in closed-form. Moreover, we prove that, as the dimensions of the observation matrix approach infinity, the empirical evolution processes will converge almost surely to the deterministic approximations.

- The learning dynamics are described and analyzed through a set of differential equations that involve complex variables and contour integral conditions. We investigate the analytical properties of these equations. Specifically, we demonstrate that the solution admits integral representations with respect to certain class of transition kernels. Furthermore, we establish the existence and uniqueness of the solutions for the concerned equations.

- We investigate the asymptotic behavior of the learning dynamics in the large-time limit. It is observed that there exists a critical threshold, below which the estimation becomes quite challenging. This phenomenon is analogous to the well-known BBP phenomenon. However, the BBP phenomenon concerns the square of the inner product, which is unsigned, but we derive a signed result and reveal its relationship with the initial conditions.

## 1.3 NOTATIONS AND ORGANIZATION

**Notations:** Throughout the paper, lowercase and uppercase boldface letters represent vectors and matrices, respectively. $\boldsymbol{I}_n$ represents the identity matrix of size $n$. For two vectors $\boldsymbol{a}$ and $\boldsymbol{b}$, their inner product is defined as $\langle \boldsymbol{a}, \boldsymbol{b} \rangle = \boldsymbol{a}^\top \boldsymbol{b}$. Let $\mathbb{R}_+$ and $\mathbb{R}_*$ denote the sets $\mathbb{R}_+ = \{x : x \geqslant 0\}$ and $\mathbb{R}_* = \{x : x > 0\}$, respectively. The notation $\mathcal{C}(A)$ represents the set of continuous functions defined on $A$. The norms $\|\cdot\|_2$, $\|\cdot\|_F$, $\|\cdot\|_{\mathcal{C}(A)}$, and $\|\cdot\|_{TV}$ denote the $\ell_2$-norm for vectors, Frobenius norm for matrices, supremum norm of continuous functions on $A$, and the total variation of signed measures, respectively. The indicator function is denoted as $\mathbb{1}\{\cdot\}$ and the Dirac measure at a point $x$ is denoted as $\delta(x)$. Notation $(f * g)(t) = \int_0^t f(t-a)g(a)\mathrm{d}a$ represents the convolution of $f(t)$ and $g(t)$. The function $(x)^+ = \max(x, 0)$. The sign function is denoted as $\mathrm{Sgn}(x) = \mathbb{1}\{x > 0\} - \mathbb{1}\{x < 0\}$ and $\xrightarrow{a.s.}$ represents almost sure convergence.

**Organization:** The rest of the paper is organized as follows. In Section 2, we introduce the problem. In Section 3, we state the main results, including the closed-form deterministic approximation for the learning dynamics and the asymptotic behavior with long-term learning. The implications and potential applications of the main results are also discussed. Experiment results are given in Section 4 and Section 5 concludes the paper.

## 2 PROBLEM SETUP

We consider the following rank-one Jonstone's spiked model

$$\boldsymbol{X} = \boldsymbol{Z} + \lambda \cdot \boldsymbol{u}^* \boldsymbol{v}^{*\top} \in \mathbb{R}^{p \times n}, \tag{1}$$

where $\boldsymbol{X}$ is the observation and $\boldsymbol{Z} = (Z_{ij})$ denotes the random noise matrix that contains independent and identically distributed (i.i.d.) elements with mean zero and variance $n^{-1}$. The unit vectors $\boldsymbol{u}^* \in \mathbb{R}^p$ and $\boldsymbol{v}^* \in \mathbb{R}^n$ represent the directions of the left and right ground truth signals, respectively. The parameter $\lambda \in (0, \infty)$ denotes the SNR (Zhang & Mondelli, 2024). The main target is to estimate the signal components $(\boldsymbol{u}^*, \boldsymbol{v}^*)$ from the observation matrix $\boldsymbol{X}$ in high dimensions. To this end, we consider the following cost function (Nadakuditi, 2014)

$$\mathcal{H}(\boldsymbol{u}, \boldsymbol{v}) = \frac{1}{2} \left\| \boldsymbol{X} - \boldsymbol{u}\boldsymbol{v}^\top \right\|_F^2. \tag{2}$$

According to the Eckart-Young-Mirsky (EYM) theorem (Eckart & Young, 1936), the optimal solution to the optimization problem

$$(\boldsymbol{u}_{\mathrm{eym}}, \boldsymbol{v}_{\mathrm{eym}}) = \underset{\|\boldsymbol{u}\|_2 = \|\boldsymbol{v}\|_2 = 1}{\arg\min} \mathcal{H}(\boldsymbol{u}, \boldsymbol{v}), \tag{3}$$

is given by $(\boldsymbol{u}_{\mathrm{eym}}, \boldsymbol{v}_{\mathrm{eym}}) = (\widehat{\boldsymbol{u}}_1, \widehat{\boldsymbol{v}}_1)$, where $\widehat{\boldsymbol{u}}_1$ and $\widehat{\boldsymbol{v}}_1$ denote the left and right singular vectors of $\boldsymbol{X}$ corresponding to the largest singular value.

When both the dimensions $p$ and $n$ are very large, iterative methods such as power iteration and GD-based optimization are utilized to compute the top singular vectors (Martinsson & Tropp, 2020). While power iteration exhibits rapid convergence under high SNR conditions (Wu & Zhou, 2024), its effectiveness is often problem-specific and relies on favorable spectral gaps. In contrast, GD offers wider applicability and greater flexibility across diverse problem structures. Given the initial point $(\boldsymbol{u}_0, \boldsymbol{v}_0)$ with $\|\boldsymbol{u}_0\|_2 = \|\boldsymbol{v}_0\|_2 = 1$, when the step size is small, the discrete process of GD can be approximated by the gradient flow (Li et al., 2017; Liu, 2017), given by

$$\frac{\mathrm{d}}{\mathrm{d}t} \begin{bmatrix} \boldsymbol{u}_t \\ \boldsymbol{v}_t \end{bmatrix} = -\mathrm{grad}(\mathcal{H}(\boldsymbol{u}_t, \boldsymbol{v}_t)) = - \begin{bmatrix} \mathrm{proj}_{\boldsymbol{u}_t} \left( \nabla_{\boldsymbol{u}} \mathcal{H}(\boldsymbol{u}_t, \boldsymbol{v}_t) \right) \\ \mathrm{proj}_{\boldsymbol{v}_t} \left( \nabla_{\boldsymbol{v}} \mathcal{H}(\boldsymbol{u}_t, \boldsymbol{v}_t) \right) \end{bmatrix}, \tag{4}$$

where $\mathrm{grad}(\cdot)$ represents the Riemannian gradient operator (Gess et al., 2024) and $\mathrm{proj}_{\boldsymbol{x}}(\boldsymbol{y}) = (\boldsymbol{I} - \boldsymbol{x}\boldsymbol{x}^\top)\boldsymbol{y}$ denotes the projection onto the tangent space. We note that (4) enforces the unit norm constraint on $\boldsymbol{u}_t$ and $\boldsymbol{v}_t$, and the detailed proof is given in Appendix C. During the learning process, the inner products between the ground truth and the estimates, i.e., $q_u(t) = \langle \boldsymbol{u}^*, \boldsymbol{u}_t \rangle$ and $q_v(t) = \langle \boldsymbol{v}^*, \boldsymbol{v}_t \rangle$, are of interest. In particular, by $\|\boldsymbol{x}^* - \boldsymbol{x}\|_2^2 = 2 - 2\langle \boldsymbol{x}, \boldsymbol{x}^* \rangle$, the inner product is equivalent to the distance.

In this work, we aim to derive the deterministic approximations for $q_u(t)$ and $q_v(t)$. To facilitate the analysis, we make the following assumptions.

**Assumption 1.** *As $n \to \infty$, the ratio $p/n \to c > 0$.*

This assumption is common in the study of high-dimensional random matrices (Bai et al., 2010; Cui & Zdeborová, 2023; Ham et al., 2025). Here, $p = p(n)$ can be viewed as a sequence indexed by $n$. In the following, we use $p, n \to \infty$ to denote this asymptotic regime.

**Assumption 2.** *The random matrix $\boldsymbol{Z}$ has i.i.d. entries such that $\mathbb{E}[Z_{ij}] = 0$, $\mathbb{E}[|\sqrt{n}Z_{ij}|^2] = 1$, and $\mathbb{E}[|\sqrt{n}Z_{ij}|^4] < \infty$.*

More rigorously, the elements $Z_{ij} = n^{-1/2}X_{ij}$ are sampled from a double array $\{X_{ij}\}_{i,j \geqslant 1}$, where $X_{ij}$s are standardized i.i.d. random variables. This assumption is general and distribution-independent. The fourth-order moment condition is to ensure that the largest singular value of $\boldsymbol{Z}$ is almost surely bounded (Yin et al., 1988; Bai & Silverstein, 1998).

## 3 MAIN RESULTS

### 3.1 LEARNING DYNAMICS

Our goal is to obtain a closed-form solution for the inner products $q_u$ and $q_v$ in high dimensions. We note that $q_u$ and $q_v$ are random processes, and will demonstrate that these two random processes will almost surely converge to two deterministic processes, which have closed-form expressions. These deterministic processes can be described by the integration of certain "basis" functions with respect to the famous Marčenko-Pastur (MP) measure (Marčenko & Pastur, 1967)

$$\mu(\mathrm{d}x) = (1 - c^{-1})^+ \delta(0) + \frac{\sqrt{[x - (1 - \sqrt{c})^2]^+ [(1 + \sqrt{c})^2 - x]^+}}{2\pi c x} \mathrm{d}x. \tag{5}$$

To simplify the notation, we define the co-MP measure $\underline{\mu}(\mathrm{d}x) = (1 - c)\delta(0) + c\mu(\mathrm{d}x)$ and the basis functions

$$\ell_{1,x}(t) = \cosh(\sqrt{x}t), \quad \ell_{2,x}(t) = \frac{1}{2\sqrt{x}}\sinh(2\sqrt{x}t),$$

$$\ell_{3,x}(t) = \frac{1}{\sqrt{x}}\sinh(\sqrt{x}t), \quad \ell_{4,x}(t) = \cosh(2\sqrt{x}t). \tag{6}$$

**Theorem 1.** *(Deterministic Approximations for $q_u$ and $q_v$) Let $\boldsymbol{u}_0 \in \mathbb{R}^p$ and $\boldsymbol{v}_0 \in \mathbb{R}^n$ be the initial vectors with unit norms and define $\alpha_u = q_u(0) = \langle \boldsymbol{u}_0, \boldsymbol{u}^* \rangle$ and $\alpha_v = q_v(0) = \langle \boldsymbol{v}_0, \boldsymbol{v}^* \rangle$. Under Assumptions 1 and 2, we have, for any $T > 0$*

$$\sup_{0 \leqslant t \leqslant T} |q_u(t) - \widetilde{q}_u(t)| \xrightarrow[p,n \to \infty]{a.s.} 0, \quad \sup_{0 \leqslant t \leqslant T} |q_v(t) - \widetilde{q}_v(t)| \xrightarrow[p,n \to \infty]{a.s.} 0, \tag{7}$$

*where $\widetilde{q}_u(t) = \widehat{q}_u(t)/\sqrt{\widehat{p}(t)}$ and $\widetilde{q}_v(t) = \widehat{q}_v(t)/\sqrt{\widehat{p}(t)}$. Here, the deterministic functions $\widehat{q}_u(t)$, $\widehat{q}_v(t)$, and $\widehat{p}(t)$ are defined as*

$$\widehat{q}_u(t) = -\frac{\alpha_v}{\lambda}\int_{\mathbb{R}} x\left(\ell_{3,x} * \ell_{3,\vartheta_\lambda}\right)(t)\underline{\mu}(\mathrm{d}x) + \alpha_v\lambda\ell_{3,\vartheta_\lambda}(t)$$

$$\quad - \frac{\alpha_u c(1 + \lambda^2)}{\lambda^2}\int_{\mathbb{R}}(\ell_{1,\vartheta_\lambda} * \ell_{3,x})(t)\mu(\mathrm{d}x) + \alpha_u\ell_{1,\vartheta_\lambda}(t), \tag{8}$$

$$\widehat{q}_v(t) = -\frac{\alpha_u c}{\lambda}\int_{\mathbb{R}} x(\ell_{3,x} * \ell_{3,\vartheta_\lambda})(t)\mu(\mathrm{d}x) + \alpha_u\lambda\ell_{3,\vartheta_\lambda}(t)$$

$$\quad - \frac{\alpha_v(\lambda^2 + c)}{\lambda^2}\int_{\mathbb{R}}(\ell_{1,\vartheta_\lambda} * \ell_{3,x})(t)\underline{\mu}(\mathrm{d}x) + \alpha_v\ell_{1,\vartheta_\lambda}(t), \tag{9}$$

$$\widehat{p}(t) = 1 + 2\int_0^t \widehat{g}_{u,v}(a)\mathrm{d}a$$

$$\quad + \int_0^t \mathrm{d}a \int_{\mathbb{R}} \mu(\mathrm{d}x)\left\{2\alpha_u\lambda(\widehat{q}_v \cdot \ell_{1,x})(a) + 2\lambda^2\widehat{q}_v(a) \cdot (\widehat{q}_v * \ell_{1,x})(a)\right\}, \tag{10}$$

$$\widehat{g}_{u,v}(t) = \int_{\mathbb{R}} x \ell_{2,x}(t)[\mu + \underline{\mu}](\mathrm{d}x)$$

$$+ 2 \int_{\mathbb{R}} x \left\{ \lambda \alpha_u [(\widehat{q}_v \cdot \ell_{1,x}) * \ell_{2,x}](t) + \lambda^2 [[\widehat{q}_v \cdot (\widehat{q}_v * \ell_{1,x})] * \ell_{2,x}](t) \right\} \mu(\mathrm{d}x)$$

$$+ 2 \int_{\mathbb{R}} x \left\{ \lambda \alpha_v [(\widehat{q}_u \cdot \ell_{1,x}) * \ell_{2,x}](t) + \lambda^2 [[\widehat{q}_u \cdot (\widehat{q}_u * \ell_{1,x})] * \ell_{2,x}](t) \right\} \underline{\mu}(\mathrm{d}x)$$

$$+ \int_{\mathbb{R}} x \left\{ \lambda \alpha_u [(\widehat{q}_v \cdot \ell_{3,x}) * \ell_{4,x}](t) + \lambda^2 [[\widehat{q}_v \cdot (\widehat{q}_v * \ell_{3,x})] * \ell_{4,x}](t) \right\} \mu(\mathrm{d}x)$$

$$+ \int_{\mathbb{R}} x \left\{ \lambda \alpha_v [(\widehat{q}_u \cdot \ell_{3,x}) * \ell_{4,x}](t) + \lambda^2 [[\widehat{q}_u \cdot (\widehat{q}_u * \ell_{3,x})] * \ell_{4,x}](t) \right\} \underline{\mu}(\mathrm{d}x), \quad (11)$$

with $\vartheta_\lambda = (1 + \lambda^2)(c + \lambda^2)/\lambda^2$.

*Proof:* The proof of Theorem 1 is given in Appendix D. $\qquad\square$

The expression for the evolution dynamics $\widetilde{q}_u(t)$ and $\widetilde{q}_v(t)$ involves a large number of integrals and convolution operations. However, since the MP measures $\mu$ and $\underline{\mu}$ have bounded support and density functions, the terms $\widehat{q}_u(t)$, $\widehat{q}_v(t)$, and $\widehat{p}(t)$ can be computed numerically for finite $t$. Specifically, by equally partitioning the intervals $[0, t]$ and $[(1 - \sqrt{c})^2, (1 + \sqrt{c})^2]$ into $N_t$ and $N_x$ grid cells respectively and applying the standard rectangle method to approximate the integration, the overall computational complexity is at most $O(N_t^2 N_x)$. Crucially, this complexity is polynomial and depends only on the discretization parameters $N_t$ and $N_x$, which are not prohibitively large for the moderate values of $t$.

An interesting observation is that when both $\alpha_u$ and $\alpha_v$ are zero, the inner products will be approximately zero throughout the learning process. This phenomenon can be intuitively explained by the gradient flow (4). When $\boldsymbol{u}_t$ and $\boldsymbol{u}^*$, as well as $\boldsymbol{v}_t$ and $\boldsymbol{v}^*$, are orthogonal, their gradients are given by

$$\frac{\mathrm{d}\boldsymbol{u}_t}{\mathrm{d}t} = (\boldsymbol{I}_p - \boldsymbol{u}_t \boldsymbol{u}_t^\top) \boldsymbol{Z} \boldsymbol{v}_t, \quad \frac{\mathrm{d}\boldsymbol{v}_t}{\mathrm{d}t} = (\boldsymbol{I}_n - \boldsymbol{v}_t \boldsymbol{v}_t^\top) \boldsymbol{Z}^\top \boldsymbol{u}_t. \quad (12)$$

This indicates that vectors $\boldsymbol{u}_t$ and $\boldsymbol{v}_t$ will only run along the directions spanned by projection of the noise and fail to learn ground truth. In Section 3.2, we will provide a more general condition under which learning fails in the large-time limit.

Theorem 1 has several potential applications, as discussed in following remarks.

**Remark 1.** *(Early Stopping Strategy Design) Consider the task of estimating the high-dimensional signal directions $\boldsymbol{u}^*$ and $\boldsymbol{v}^*$. Assume we know the SNR $\lambda$ and the observation matrix $\boldsymbol{X}$. Additionally, there is a performance requirement*

$$\min\left\{ \langle \boldsymbol{u}_t, \boldsymbol{u}^* \rangle, \langle \boldsymbol{v}_t, \boldsymbol{v}^* \rangle \right\} \geqslant \gamma, \quad (13)$$

*for a given threshold $\gamma \in (0, 1)$. Let $\widehat{c} = p/n$ be the estimator for $c$. According to Theorem 1, the deterministic approximations only depend on $(c, \lambda, \alpha_u, \alpha_v)$. Thus, the following stopping time estimator can be constructed*

$$\widehat{t}(\alpha_u, \alpha_v) = \inf\left\{ t \geqslant 0 : \min\{\widetilde{q}_u(t), \widetilde{q}_v(t)\} \geqslant \gamma \right\}. \quad (14)$$

*We note that $\widehat{t}(\alpha_u, \alpha_v)$ can be computed by line search method after obtaining $\widetilde{q}_u$ and $\widetilde{q}_v$. As a result, the computational complexity remains dominated by $O(N_t^2 N_x)$. Furthermore, if we know the prior information about $(\boldsymbol{u}^*, \boldsymbol{v}^*)$ (Aubin et al., 2021), we can design the statistically optimal stopping time. For example, set $\widehat{t}^* = \mathbb{E}_{\alpha_u, \alpha_v}[\widehat{t}(\alpha_u, \alpha_v)]$. We note that in practice, knowing the exact SNR may be too restrictive. Fortunately, this method can also be extended to situations where $\lambda \in I$ for some known interval $I$. From a robustness perspective, we can set $\widehat{t}_R^* = \sup_{\lambda \in I} \widehat{t}^*$ to determine the stopping time.*

**Remark 2.** *(SNR Estimation) Consider the task of estimating the SNR $\lambda$. Assume the signal directions $(\boldsymbol{u}^*, \boldsymbol{v}^*)$ are known. Applying gradient flow, we obtain the following estimator*

$$\widehat{\lambda}(\boldsymbol{u}^*, \boldsymbol{v}^*) = \arg\min_{\lambda > 0} \|\langle \boldsymbol{u}_t, \boldsymbol{u}^* \rangle - \widetilde{q}_u(t)\|_{\mathcal{C}([0,T])} + \|\langle \boldsymbol{v}_t, \boldsymbol{v}^* \rangle - \widetilde{q}_v(t)\|_{\mathcal{C}([0,T])}, \quad (15)$$

*for given $T > 0$.*

We also have the following technical remarks.

**Remark 3.** *The expressions for $\widetilde{q}_u$ and $\widetilde{q}_v$ can be further simplified. For instance, in (8), the term $(\ell_{3,x} * \ell_{3,\vartheta_\lambda})(t)$ in the integration can be computed directly since the basis functions are linear combinations of exponential functions. Similarly, the term $(\widehat{q}_v * \ell_{1,x})(t)$ on the right-hand side (RHS) of (10) can also be simplified by interchanging the order of the MP integral and convolution according to Fubini's theorem. These simplifications can significantly reduce the complexity of the numerical evaluation.*

**Remark 4.** *(On the Role of $\widehat{g}_{u,v}$) The function $\widehat{g}_{u,v}$ can be used to characterize the correlation between the estimates $(\boldsymbol{u}_t, \boldsymbol{v}_t)$ and the random noise $\boldsymbol{Z}$. In particular, it can be verified by the proof in Appendix D that*

$$\sup_{0 \leqslant t \leqslant T} |\widetilde{g}_{u,v}(t) - \langle \boldsymbol{u}_t, \boldsymbol{Z}\boldsymbol{v}_t \rangle| \xrightarrow[p,n\to\infty]{a.s.} 0, \tag{16}$$

*where $\widetilde{g}_{u,v}(t) = \widehat{g}_{u,v}(t)/\widehat{p}(t)$.*

**Remark 5.** *In the field of statistical physics, the Crisanti-Horner-Sommers-Cugliandolo-Kurchan (CHSCK) equation is utilized to describe the behavior of Langevin dynamics in the spherical p-spin glass model (Sarao Mannelli et al., 2019; 2020), which accounts for more complex mixed matrix-tensor structure and gradient noise. The loss function considered in the concerned rank-one matrix recovery problem can likewise be interpreted as the Hamiltonian of a corresponding spin glass system. However, the CHSCK equation is implicit and its solution is purely numerical, as it involves full-time correlations and requires solving functional fixed-point equations for the Lagrange multipliers. In contrast, Theorem 1 provides a closed-form expression for the dynamics, offering a more tractable framework for understanding the learning behavior. Furthermore, Theorem 1 has the potential to be extended to the case of stochastic gradient descent within the framework of stochastic calculus on manifolds (Da Prato & Zabczyk, 2014; Gess et al., 2024), yielding more general results, which are left for future investigation.*

**Remark 6.** *(Comparison with (Bodin & Macris, 2021)) The proof strategy of Theorem 1 is similar to that in (Bodin & Macris, 2021). Specifically, we first construct a system of differential equations for the characteristic functions using the resolvent, and then solve it via the Laplace transform. However, the analytical properties and the existence and uniqueness of the solutions to the characteristic equation were not investigated in (Bodin & Macris, 2021). We note that these properties are important (Hachem et al., 2007). Specifically, the governing system (45) describes the dynamics of the inner products, so the uniqueness of the solution to (4) does not guarantee that of the solution to the constructed system. We show that this solution admits an integral representation via a transition kernel and is unique under the given representation (cf. Theorem 3). Substituting $(\boldsymbol{u}_t, \boldsymbol{v}_t)$ from (4) into the constructed system automatically satisfies the integral representation, which provides a rigorous theoretical guarantee. This framework can also be applied to study the learning dynamics of other problems (Bordelon et al., 2020; Loureiro et al., 2021; Paquette et al., 2024; 2021). Furthermore, in contrast to the Wigner model, the rectangular model involves correlation terms between the random matrix and its resolvent. The almost sure convergence of the bilinear form is established in (36), which is not yet available in the literature. Additionally, the poles of the Laplace transforms are of higher order and are implicitly defined for several terms, which makes the problem more challenging.*

Theorem 1 is derived for the continuous gradient flow. However, in practical applications, GD with a finite learning rate is widely employed. A natural question is whether this framework can be extended to the discrete-time setting of GD. The following remark shows that the gap between the discrete and continuous processes is small. The detailed proof is given in Appendix E.

**Remark 7.** *(Continuous to Discrete Process) Consider the initial point $(\widetilde{\boldsymbol{u}}_0, \widetilde{\boldsymbol{v}}_0)$ such that $\|\widetilde{\boldsymbol{u}}_0\|_2 = \|\widetilde{\boldsymbol{v}}_0\|_2 = 1$. Assume the following canonical Riemannian GD update*

$$\begin{bmatrix} \widetilde{\boldsymbol{u}}_{k+1} \\ \widetilde{\boldsymbol{v}}_{k+1} \end{bmatrix} = \mathrm{retr}_{(\widetilde{\boldsymbol{u}}_k, \widetilde{\boldsymbol{v}}_k)} \left( -\eta \cdot \mathrm{grad} \mathcal{H}(\widetilde{\boldsymbol{u}}_k, \widetilde{\boldsymbol{v}}_k) \right) = \begin{bmatrix} \mathrm{retr}_{\widetilde{\boldsymbol{u}}_k} \left( -\eta \cdot \mathrm{proj}_{\widetilde{\boldsymbol{u}}_k} \nabla_{\boldsymbol{u}} \mathcal{H}(\widetilde{\boldsymbol{u}}_k, \widetilde{\boldsymbol{v}}_k) \right) \\ \mathrm{retr}_{\widetilde{\boldsymbol{v}}_k} \left( -\eta \cdot \mathrm{proj}_{\widetilde{\boldsymbol{v}}_k} \nabla_{\boldsymbol{v}} \mathcal{H}(\widetilde{\boldsymbol{u}}_k, \widetilde{\boldsymbol{v}}_k) \right) \end{bmatrix}, \tag{17}$$

*where the retraction mapping $\mathrm{retr}_{(\boldsymbol{u},\boldsymbol{v})}$ is given by $\mathrm{retr}_{(\boldsymbol{u},\boldsymbol{v})}(\boldsymbol{x}_u, \boldsymbol{y}_v) = (\mathrm{retr}_{\boldsymbol{u}}(\boldsymbol{x}_u), \mathrm{retr}_{\boldsymbol{v}}(\boldsymbol{y}_v)) = (\frac{\boldsymbol{u}+\boldsymbol{x}_u}{\|\boldsymbol{u}+\boldsymbol{x}_u\|_2}, \frac{\boldsymbol{v}+\boldsymbol{y}_v}{\|\boldsymbol{v}+\boldsymbol{y}_v\|_2})$, and $\eta$ represents the learning rate. Then, for any given $T > 0$ and $\eta > 0$, there exists a constant $C > 0$ such that the following bound*

$$\max_{0 \leqslant k \leqslant \lfloor \frac{T}{\eta} \rfloor} \left\{ \|\widetilde{\boldsymbol{u}}_k - \boldsymbol{u}_{k\eta}\|_2, \|\widetilde{\boldsymbol{v}}_k - \boldsymbol{v}_{k\eta}\|_2 \right\} \leqslant C(\eta + \eta^2), \tag{18}$$

*holds with probability 1 for large $p$ and $n$. Here, $(\boldsymbol{u}_t, \boldsymbol{v}_t)$ denotes the solution of (4) under the same initialization, i.e., $(\boldsymbol{u}_0, \boldsymbol{v}_0) = (\widetilde{\boldsymbol{u}}_0, \widetilde{\boldsymbol{v}}_0)$. The above result demonstrates that in high-dimensional settings, the continuous gradient flow serves as a good approximation to the discrete GD over a finite time interval $[0, T]$. Therefore, we can characterize the behavior of GD by studying the dynamics of the system (4).*

As discussed in Section 2, the loss function $\mathcal{H}(\boldsymbol{u}, \boldsymbol{v})$ reaches the global minimum at $(\widehat{\boldsymbol{u}}_1, \widehat{\boldsymbol{v}}_1)$ by the EYM theorem. We have the following corollary regarding the learning dynamics of the gap between the loss function and its optimal value.

**Corollary 1.** *(Deterministic Approximation for the Loss Function) With the same settings as Theorem 1, we denote*

$$\overline{\mathcal{H}}(\boldsymbol{u}, \boldsymbol{v}) = \frac{1}{2} \left\| \boldsymbol{X} - \boldsymbol{u}\boldsymbol{v}^\top \right\|_F^2 - \frac{1}{2} \left\| \boldsymbol{X} - \widehat{\boldsymbol{u}}_1 \widehat{\boldsymbol{v}}_1^\top \right\|_F^2. \tag{19}$$

*Further define $\lambda_c = \max(c^{1/4}, \lambda)$ and $\vartheta_{\lambda,c} = (1 + \lambda_c^2)(c + \lambda_c^2)/\lambda_c^2$. Then, we have, for any $T > 0$,*

$$\sup_{0 \leqslant t \leqslant T} \left| \overline{\mathcal{H}}(\boldsymbol{u}_t, \boldsymbol{v}_t) - \widetilde{\overline{\mathcal{H}}}(t) \right| \xrightarrow[p,n\to\infty]{a.s.} 0, \tag{20}$$

*where $\widetilde{\overline{\mathcal{H}}}(t) = \sqrt{\vartheta_{\lambda,c}} - \widetilde{g}_{u,v}(t) - \lambda \widetilde{q}_u(t)\widetilde{q}_v(t)$.*

*Proof:* By algebra, we have

$$\overline{\mathcal{H}}(\boldsymbol{u}_t, \boldsymbol{v}_t) = \frac{1}{2} \operatorname{Tr}(\boldsymbol{X} - \boldsymbol{u}_t\boldsymbol{v}_t^\top)(\boldsymbol{X}^\top - \boldsymbol{v}_t\boldsymbol{u}_t^\top) - \frac{1}{2} \operatorname{Tr}(\boldsymbol{X} - \widehat{\boldsymbol{u}}_1\widehat{\boldsymbol{v}}_1^\top)(\boldsymbol{X}^\top - \widehat{\boldsymbol{v}}_1\widehat{\boldsymbol{u}}_1^\top)$$
$$= \langle \widehat{\boldsymbol{u}}_1, \boldsymbol{X}\widehat{\boldsymbol{v}}_1 \rangle - \langle \boldsymbol{u}_t, \boldsymbol{X}\boldsymbol{v}_t \rangle = \sigma_1 - \langle \boldsymbol{u}_t, \boldsymbol{Z}\boldsymbol{v}_t \rangle - \lambda q_u(t)q_v(t), \tag{21}$$

where $\sigma_1$ denotes the largest singular value of $\boldsymbol{X}$. According to (Liu et al., 2025b, Theorem 2), the largest eigenvalue of $\boldsymbol{X}\boldsymbol{X}^\top$ converges to $\vartheta_{\lambda,c}$ almost surely, as $p, n \to \infty$. This implies $\sigma_1 \to \sqrt{\vartheta_{\lambda,c}}$ almost surely, by the continuous mapping theorem (Van der Vaart, 2000, Theorem 2.3). Using Theorem 1 and (16), (20) is proved. $\qquad\square$

We note that the loss functions $\mathcal{H}$ in (2) and $\overline{\mathcal{H}}$ in (19) are equivalent since the second term on the RHS of (19) can be viewed as a constant. From Theorem 1 and Corollary 1, it can be observed that the learning dynamics is rotational invariant. In particular, for any $\overline{\alpha}_u, \overline{\alpha}_v \in [-1, 1]$, all initial points $(\boldsymbol{u}_0, \boldsymbol{v}_0)$ satisfying $\langle \boldsymbol{u}_0, \boldsymbol{u}^* \rangle = \overline{\alpha}_u$ and $\langle \boldsymbol{v}_0, \boldsymbol{v}^* \rangle = \overline{\alpha}_v$ exhibit asymptotically same learning dynamics, i.e., the same evolution of the inner products and the loss function.

## 3.2 Large-Time Limit

The following theorem demonstrates the asymptotic behavior of the deterministic approximations $\widetilde{q}_u$ and $\widetilde{q}_v$, as $t \to \infty$.

**Theorem 2.** *(Asymptotics of the Learning Dynamics) Define $\mathcal{I} = \operatorname{Sgn}\{\frac{\alpha_v}{\sqrt{1+\lambda^2}} + \frac{\alpha_u}{\sqrt{\lambda^2+c}}\}$. As $t \to \infty$, we have*

$$\lim_{t\to\infty} \widetilde{q}_u(t) = \widetilde{q}_u^\infty = \mathcal{I} \cdot \sqrt{\frac{1 - c\lambda^{-4}}{1 + c\lambda^{-2}}} \cdot \mathbb{1}\{\lambda > c^{1/4}\}, \tag{22}$$

$$\lim_{t\to\infty} \widetilde{q}_v(t) = \widetilde{q}_v^\infty = \mathcal{I} \cdot \sqrt{\frac{1 - c\lambda^{-4}}{1 + \lambda^{-2}}} \cdot \mathbb{1}\{\lambda > c^{1/4}\}. \tag{23}$$

*Proof:* The proof of Theorem 2 is given in Appendix F. $\qquad\square$

From Theorem 2, we can observe that when $\lambda < c^{1/4}$, the gradient flow estimation is asymptotically trivial as $\widetilde{q}_u, \widetilde{q}_v \to 0$. When $\lambda$ exceeds $c^{1/4}$, a phase transition occurs, indicating that $c^{1/4}$ is the critical threshold for estimation. This phenomenon is consistent with the well-known BBP phase transition. In the following remarks, we compare Theorem 2 with the BBP phenomenon and discuss the potential applications. We note that although Theorem 2 is established for a continuous process, the BBP transition phenomenon can also be extended to GD with a small learning rate by (18).

**Remark 8.** *(Comparison with BBP Phase Transition) Recall $\widehat{\boldsymbol{u}}_1$ and $\widehat{\boldsymbol{v}}_1$ are the left and right singular vectors of $\boldsymbol{X}$ corresponding to the largest singular value. According to the BBP phase transition (Baik et al., 2005), we have*

$$|\langle \widehat{\boldsymbol{u}}_1, \boldsymbol{u}^* \rangle|^2 \xrightarrow[p,n\to\infty]{a.s.} \frac{1 - c\lambda^{-4}}{1 + c\lambda^{-2}} \cdot \mathbb{1}\{\lambda > c^{1/4}\}, \tag{24}$$

$$|\langle \widehat{\boldsymbol{v}}_1, \boldsymbol{v}^* \rangle|^2 \xrightarrow[p,n\to\infty]{a.s.} \frac{1 - c\lambda^{-4}}{1 + \lambda^{-2}} \cdot \mathbb{1}\{\lambda > c^{1/4}\}. \tag{25}$$

*The major difference between (24)-(25) and (22)-(23) lies in the sign and the case $\frac{\alpha_v}{\sqrt{1+\lambda^2}} + \frac{\alpha_u}{\sqrt{\lambda^2+c}} = 0$. This indicates that if we have prior information about the initial points $\alpha_u$ and $\alpha_v$, it is possible to determine the confidence intervals for the signs, thereby estimating the directions of the ground truth with gradient flow. In the experiments, it will be shown that when $\frac{\alpha_v}{\sqrt{1+\lambda^2}} + \frac{\alpha_u}{\sqrt{\lambda^2+c}} = 0$, the estimation becomes difficult.*

**Remark 9.** *(Estimation for $\lambda \leqslant c^{1/4}$) Even when $\widetilde{q}_v, \widetilde{q}_u \to 0$, there is still a chance to estimate the signals. Assuming we have prior information about $\alpha_u$ and $\alpha_v$, similar to Remark 1, we can set*

$$\widehat{t}(\alpha_u, \alpha_v) = \arg\max_{t \geqslant 0} \left\{ |\widetilde{q}_u(t)| + |\widetilde{q}_v(t)| \right\}, \tag{26}$$

*and design the learning time according to $\widehat{t}(\alpha_u, \alpha_v)$. We note that when the prior distribution of the ground truth $(\boldsymbol{u}^*, \boldsymbol{v}^*)$ is known, the critical threshold for iterative algorithms like AMP may be lower than $c^{1/4}$ (Lelarge & Miolane, 2017).*

We also have the following phase transition phenomenon regarding the asymptotic behavior of the loss function.

**Corollary 2.** *Define $\mathcal{J} = \mathbb{1}\{\frac{\alpha_v}{\sqrt{\lambda^2+1}} + \frac{\alpha_u}{\sqrt{\lambda^2+c}} = 0\}$. As $t \to \infty$, we have*

$$\lim_{t\to\infty} \widetilde{\overline{\mathcal{H}}}(t) = \mathcal{J} \cdot \left[ \sqrt{\vartheta_{\lambda,c}} - 1 - \sqrt{c} \right]. \tag{27}$$

*Proof:* The proof of Corollary 2 is similar to that of Theorem 2 and thus omitted. □

Recalling $\vartheta_{c,\lambda} = (1 + \lambda_c^2)(c + \lambda_c^2)/\lambda_c^2$, it can be observed that when the SNR is smaller than the critical value, $\vartheta_{c,\lambda} = (1 + \sqrt{c})^2$ and the loss tends to 0 as the learning time increases. With random initialization, we often have $\mathbb{P}(\mathcal{J} = 0) = 1$, indicating that GD can reach the global optimum and avoid saddle points in spherical low-rank signal estimation. In the case of tensors, GD could be more efficient, as the number of saddles on the tensor manifold grows exponentially with dimensionality (Subag & Zeitouni, 2017; Auffinger et al., 2013). Furthermore, Corollary 2 implies that if $\mathcal{J} \neq 1$ and $\lambda > c^{1/4}$, we have

$$\lim_{t\to\infty} \widetilde{g}_{u,v}(t) = \frac{1}{\lambda} \left[ \sqrt{\frac{\lambda^2+c}{\lambda^2+1}} + \sqrt{\frac{\lambda^2+1}{\lambda^2+c}} \right]. \tag{28}$$

Note that $\widetilde{g}_{u,v}(t)$ approximates the correlation between the estimates $(\boldsymbol{u}_t, \boldsymbol{v}_t)$ and the noise, i.e., $\widetilde{g}_{u,v}(t) \approx \langle \boldsymbol{u}_t, \boldsymbol{Z}\boldsymbol{v}_t \rangle$. As $\lambda \to \infty$, the RHS of the above equation tends to 0, indicating that higher SNR leads to reduced dependence between $(\boldsymbol{u}_t, \boldsymbol{v}_t)$ and the noise.

## 4 EXPERIMENTS

In this section, we verify the accuracy of the theoretical results. The simulation results are generated using GD, where the update rule follows (17) with a learning rate of $\eta = 0.005$.

**Accuracy of the Deterministic Approximations:** Figure 1 shows the loss function and inner product dynamics compared with their deterministic approximations. The parameters are set as $p = 900$ and $n = 1200$. The ground truth vectors are set as follows: the components of $\boldsymbol{u}^*$ and $\boldsymbol{v}^*$ are independently drawn from a standard Gaussian and Bernoulli distribution with $p = 0.3$, respectively. Both vectors are then normalized. The initial points are set as $\boldsymbol{u}_0 = (10\boldsymbol{u}^* + \boldsymbol{z}_1)/\|10\boldsymbol{u}^* + \boldsymbol{z}_1\|_2$ and $\boldsymbol{v}_0 = (5\boldsymbol{v}^* + \boldsymbol{z}_2)/\|5\boldsymbol{v}^* + \boldsymbol{z}_2\|_2$, where $\boldsymbol{z}_1 \in \mathbb{R}^p$ and $\boldsymbol{z}_2 \in \mathbb{R}^n$ are random vectors with i.i.d. standardized Gaussian elements. In Figures 1a-1c, the SNR is set to $\lambda = 0.3$, which is below the critical threshold $c^{1/4}$. In contrast, in Figures 1d-1f, the SNR is set to $\lambda = 1.5 > c^{1/4}$. By comparing

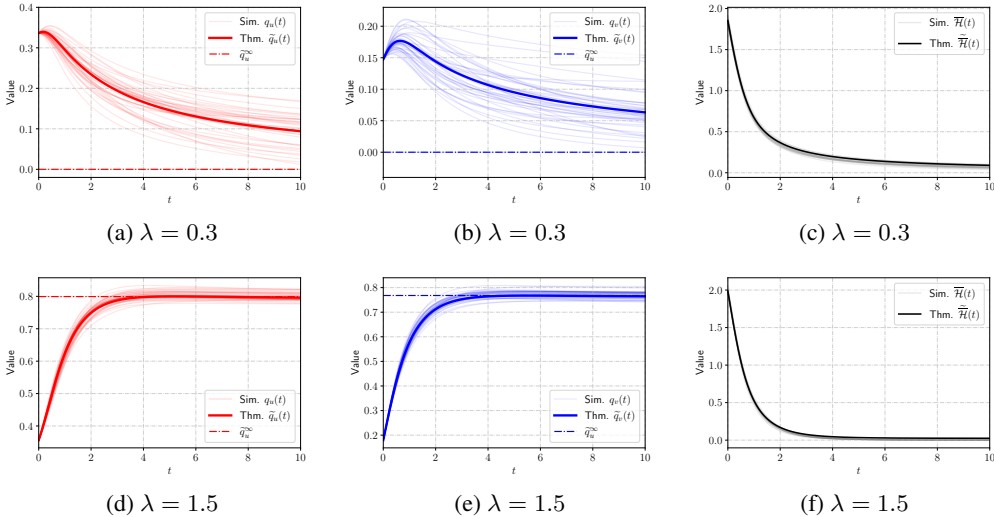

Figure 1: Theoretical and Empirical Learning Dynamics.

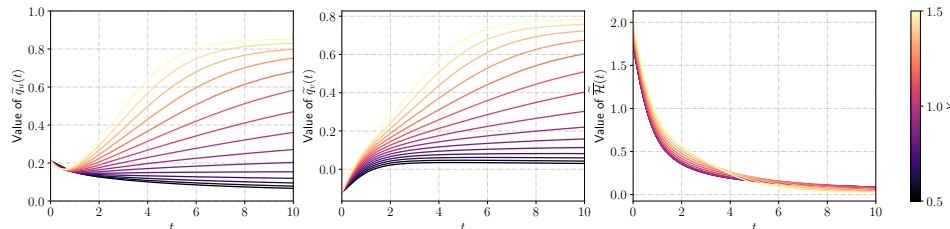

Figure 2: Learning Dynamics with Different SNR Values $\lambda$.

the dark-colored lines and the light-colored lines, it can be observed that the deterministic approximations $\widetilde{q}_u$, $\widetilde{q}_v$, and $\overline{\widetilde{\mathcal{H}}}$ are accurate, which validates the accuracy of Theorem 1 and Corollary 1. Additionally, it can be seen that the deterministic approximations become more accurate as the SNR increases. This is because, when the SNR is lower, the random noise $\mathbf{Z}$ dominates, thus $q_u$, $q_v$ have more randomness. By comparing the asymptotic values (dash-dotted lines) with the deterministic approximations, the accuracy of Theorem 2 is verified.

**Impact of the SNR:** Figure 2 illustrates the theoretical learning curves with different SNR values. The parameters are set as $c = 0.5$, $\alpha_u = 0.211$, and $\alpha_v = -0.121$. It can be observed that when $\lambda \leqslant c^{1/4} = 0.841$, $\widetilde{q}_u$ and $\widetilde{q}_v$ approach to 0 as $t \to \infty$. However, when $\lambda > c^{1/4}$, they converge to $\widetilde{q}_u^\infty$ and $\widetilde{q}_v^\infty$, respectively, which are positive for the concerned case. This validates the accuracy of Theorem 2 in terms of the sign of the limits, because $\frac{\alpha_v}{\sqrt{1+\lambda^2}} + \frac{\alpha_u}{\sqrt{c+\lambda^2}}$ is strictly positive for all $\lambda \in [0.1, 2]$.

It can be observed that as the SNR increases, the GD-based algorithm learns the hidden information $(\mathbf{u}^*, \mathbf{v}^*)$ faster. From the third sub-figure in Figure 2, we note that although the loss may be larger during the early learning stage with higher SNR, it eventually decreases to a lower level during the training. This indicates that learning is a global process and a rapid initial decrease in loss does not necessarily imply better final performance. Conversely, for data with high potential (correspondingly, high SNR), the loss may not decrease significantly in the initial stages of learning.

**Impact of the Initial Points:** Figure 3 illustrates the learning curves of $\widetilde{q}_u$ with different initial values $\alpha_u$. The parameters are set as $c = 0.5$ and $\alpha_v = 0.4$. In Figure 3a, the SNR is set to $\lambda = 1.5 > c^{1/4}$, while in Figure 3b, $\lambda = 0.2 < c^{1/4}$. The solid red curve corresponds to the case where $\frac{\alpha_v}{\sqrt{1+\lambda^2}} + \frac{\alpha_u}{\sqrt{c+\lambda^2}} = 0$, while the dashed red curve represents the line $y = \sqrt{\vartheta_{\lambda,c}} - 1 - \sqrt{c}$. The trend of the loss functions validates the accuracy of Corollary 2.

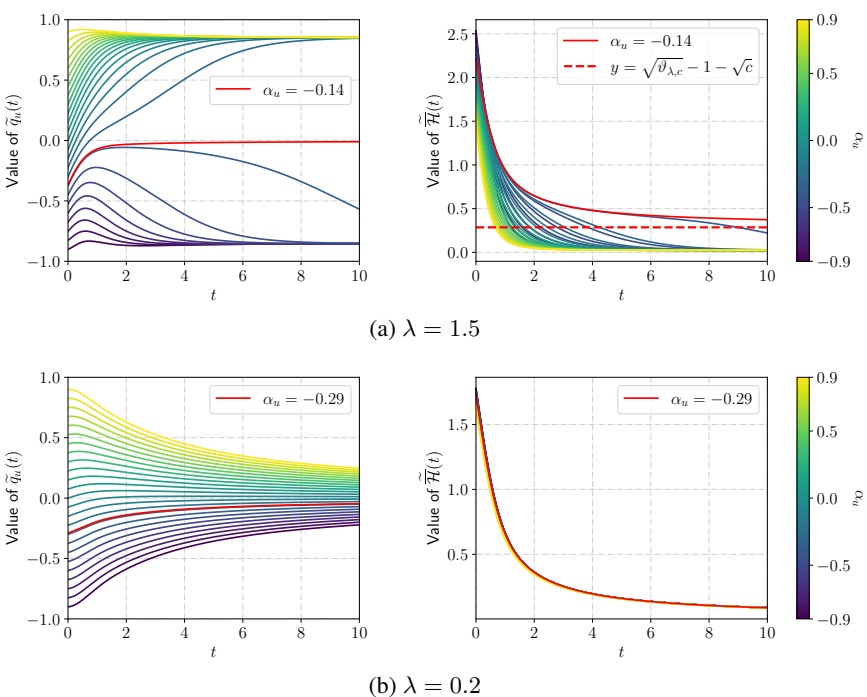

(a) $\lambda = 1.5$

(b) $\lambda = 0.2$

Figure 3: Learning Dynamics with Different Initial Values $\alpha_u$.

It can be observed that the initial value affects the speed of convergence for $\lambda > c^{1/4}$. When $\alpha_u$ is close to $-\alpha_v \sqrt{\frac{\lambda^2+c}{\lambda^2+1}}$, the gradient flow converges more slowly, whereas when the initial value $\alpha_u$ is near $\widetilde{q}_u^\infty$, the gradient flow converges faster. This also provides insights on the learning algorithms. Given the complex geometric structure of the loss surface, if the initial point is unfortunately located near some "bad" regions, it may take significant efforts for the model to learn the hidden information from the data, even though the loss decreases with GD.

## 5   Conclusions

In this work, we studied the GD dynamics of the rank-one matrix denoising problem. In particular, we obtained closed-form deterministic approximations for the inner products between the learned vectors and the ground truth, and proved that the random learning curves converge to the approximations almost surely. Additionally, we derived the asymptotic behavior of the learning dynamics, which is consistent with the BBP phase transition phenomenon. Simulations validated the accuracy of the theoretical results. Furthermore, it was observed that the gradient flow converges faster when the SNR is higher and when the initial point is close to the stationary points. In contrast, there exist "bad points" where learning is disrupted. If the initial value is near the bad points, it takes a longer time to learn the hidden information even with high SNR.

The main results of this work, i.e., Theorems 1 and 2, have many potential applications and can be further extended. Given the prior information about the initial points and the ground truth, these results can be utilized to design early stopping strategies and develop SNR estimation algorithms. In this work, we have established the almost sure convergence of the gradient flow dynamics, which corresponds to the "law of large numbers." The "central limit theorem," i.e., the asymptotic fluctuation of $q_h(t)$ around $\widetilde{q}_h(t)$, $h \in \{u, v\}$, remains an interesting open problem. Such fluctuations capture higher-order information about the test errors of learning algorithms. Moreover, the data model considered in this work is rank-one and linear. Extending this analysis to multi-rank models and nonlinear data structures, such as random feature models, represents a more general and challenging direction. From a practical standpoint, investigating these aspects would yield deeper insights into the dynamics of learning.

## 6 ACKNOWLEDGMENT

This work was supported in part by a grant from the NSFC/RGC Joint Research Scheme sponsored by the Research Grants Council of the Hong Kong Special Administrative Region, China and National Natural Science Foundation of China (Project No. N_HKUST656/22).

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

APPENDIX

## A  THE USE OF LARGE LANGUAGE MODELS (LLMS)

In this work, we utilized LLMs for grammar checking and text polishing to enhance the readability.

## B  MATHEMATICAL TOOLS

In this section, we introduce the mathematical tools used in our analysis along with related discussions.

### B.1  RESULTS ON CALCULUS OF FUNCTIONS

**Lemma 1.** *(Grönwall's Lemma) Let $T > 0$ and $h$ be a nonnegative bounded measurable function on $[0,T]$ such that, for every $t \in [0,T]$,*

$$h(t) \leqslant a + b \int_0^t h(t)\mathrm{d}t, \tag{29}$$

*for constants $a \geqslant 0$ and $b \geqslant 0$. Then, we have, for every $t \in [0,T]$,*

$$h(t) \leqslant a \exp[bt]. \tag{30}$$

**Lemma 2.** *(Final Value Theorem) Let $f \in \mathcal{C}(\mathbb{R}_+)$ be such that $\lim_{t\to\infty} e^{-at} f(t) = A$ exists for some $a > 0$ and $A \in \mathbb{R}$. Moreover, assume that $g \in \mathcal{C}(\mathbb{R}_+)$ satisfies $\int_0^\infty e^{-at}|g(t)|\mathrm{d}t < \infty$. Then, we have*

$$\lim_{t\to\infty} e^{-at}(f * g)(t) = A \cdot \int_0^\infty e^{-at}g(t)\mathrm{d}t. \tag{31}$$

**Lemma 3.** *Let $f \in \mathcal{C}([a,b])$ be such that $f(b) \neq 0$ and $\alpha > -1$. Then,*

$$\lim_{t\to\infty} \frac{\int_a^b f(x)\exp[\sqrt{x}t](b-x)^\alpha \mathrm{d}x}{f(b)\exp[\sqrt{b}t]\Gamma(\alpha+1)\left(\frac{2\sqrt{b}}{t}\right)^{\alpha+1}} = 1, \tag{32}$$

*where $\Gamma(z) = \int_0^\infty t^{z-1}\exp[-t]\mathrm{d}t$ denotes the Gamma function.*

*Proof:* The proof of Lemma 3 is given in Appendix G. □

### B.2  RESULTS ON RANDOM MATRICES

The deterministic approximation of the gradient flow relies on the convergence of the resolvent for the covariance matrix of $\boldsymbol{Z}$, which is defined as

$$\mathbf{Q}(z) = \left(\boldsymbol{Z}\boldsymbol{Z}^\top - z\boldsymbol{I}_p\right)^{-1}, \tag{33}$$

where $z \in \mathbb{C}$ such that $\mathbf{Q}(z)$ is well-defined. The co-resolvent is defined as $\underline{\mathbf{Q}}(z) = (\boldsymbol{Z}^\top \boldsymbol{Z} - z\boldsymbol{I}_n)^{-1}$. The convergence of the resolvent is a fundamental topic in RMT (Bai et al., 2010), as it characterizes the spectral distribution of the random matrix $\boldsymbol{Z}$. Recall $c = \lim_{p,n\to\infty} p/n$. Define $E_- = (1 - \sqrt{c})^2$, $E_+ = (1 + \sqrt{c})^2$, and the set $\mathcal{M} = \{0\} \cup [E_-, E_+]$. The following lemma describes the asymptotic behavior of the resolvents.

**Lemma 4.** *(Convergence of the Resolvents) Assume Assumptions 1 and 2 hold and let $\boldsymbol{u} \in \mathbb{R}^p$ and $\boldsymbol{v} \in \mathbb{R}^n$ be deterministic vectors with $\|\boldsymbol{u}\|_2 = \|\boldsymbol{v}\|_2 = 1$. Let $\Gamma$ be a compact set in $\mathbb{C}$ such that $\mathrm{dist}(\Gamma, \mathcal{M}) > 0$. Then, we have,*

$$\sup_{z\in\Gamma} \left|\boldsymbol{u}^\top \mathbf{Q}(z)\boldsymbol{u} - m(z)\right| \xrightarrow[p,n\to\infty]{a.s.} 0, \tag{34}$$

$$\sup_{z\in\Gamma} \left|\boldsymbol{v}^\top \underline{\mathbf{Q}}(z)\boldsymbol{v} - \underline{m}(z)\right| \xrightarrow[p,n\to\infty]{a.s.} 0, \tag{35}$$

$$\sup_{z \in \Gamma} \left| \boldsymbol{u}^\top \boldsymbol{Q}(z) \boldsymbol{Z} \boldsymbol{v} \right| \xrightarrow[p,n \to \infty]{a.s.} 0, \tag{36}$$

*where $m(z)$ is the unique solution to*

$$zcm^2(z) - (1 - c - z)m(z) + 1, \tag{37}$$

*such that $m(z) \in \mathbb{C}_+ \equiv \{z : \Im(z) > 0\}$ for $z \in \mathbb{C}_+$, and $\underline{m}(z) = cm(z) + (c-1)/z$.*

*Proof:* When $\Gamma$ is a singleton, the proofs for (34) and (35) can be found in (Bai et al., 2010; Couillet & Liao, 2022). The extension to a general compact set $\Gamma$ follows from an $\varepsilon$-net argument and a similar discussion as in (Bai & Silverstein, 1998, Eq. 3.23). The proof for (36) is given in Appendix H □

In fact, it can be shown that $m(z)$ is the Stieltjes transform (Bai et al., 2010) of the MP distribution. Specifically, we have the integral representations

$$m(z) = \int_{\mathbb{R}} \frac{\mu(\mathrm{d}x)}{x - z}, \quad \underline{m}(z) = \int_{\mathbb{R}} \frac{\underline{\mu}(\mathrm{d}x)}{x - z}, \tag{38}$$

where

$$\mu(\mathrm{d}x) = \left(1 - c^{-1}\right)^+ \delta(0) + \frac{\sqrt{(x - E_-)^+(E_+ - x)^+}}{2\pi cx} \mathrm{d}x,$$

$$\underline{\mu}(\mathrm{d}x) = (1 - c)^+ \delta(0) + \frac{\sqrt{(x - E_-)^+(E_+ - x)^+}}{2\pi x} \mathrm{d}x. \tag{39}$$

## C  PROOF OF THE UNIT NORM CONSTRAINT IN GRADIENT FLOW

By the evolution in (4) and taking the derivative, we can obtain

$$\frac{\mathrm{d} \|\boldsymbol{u}_t\|_2^2}{\mathrm{d}t} = 2 \left\langle \boldsymbol{u}_t, \frac{\mathrm{d}\boldsymbol{u}_t}{\mathrm{d}t} \right\rangle = -2 \left\langle \boldsymbol{u}_t, \mathrm{proj}_{\boldsymbol{u}_t}(\nabla_{\boldsymbol{u}} \mathcal{H}(\boldsymbol{u}_t, \boldsymbol{v}_t)) \right\rangle$$

$$- 2 \left\langle \boldsymbol{u}_t, (\boldsymbol{I}_p - \boldsymbol{u}_t \boldsymbol{u}_t^\top) \nabla_{\boldsymbol{u}} \mathcal{H}(\boldsymbol{u}_t, \boldsymbol{v}_t) \right\rangle = 2(\|\boldsymbol{u}_t\|_2^2 - 1)\boldsymbol{u}_t^\top \nabla_{\boldsymbol{u}} \mathcal{H}(\boldsymbol{u}_t, \boldsymbol{v}_t). \tag{40}$$

For ease of notations, we define $f_u(t) = \|\boldsymbol{u}_t\|^2 - 1$ and $g_u(t) = \boldsymbol{u}_t^\top \nabla_{\boldsymbol{u}} \mathcal{H}(\boldsymbol{u}_t, \boldsymbol{v}_t)$. Thus, we have $f_u(t) = 2 \int_0^t f_u(\alpha)g_u(\alpha)\mathrm{d}\alpha$, which implies

$$f_u(t) = C \exp \left( 2 \int_0^t g_u(\alpha)\mathrm{d}\alpha \right), \tag{41}$$

for some constant $C$. Since $f_u(0) = 0$, we have $C = 0$ and $\|\boldsymbol{u}_t\|_2^2 = 1$, for any $t \geqslant 0$. A similar argument on $\|\boldsymbol{v}_t\|_2^2$ yields the same result. Therefore, we have proved that the unit norm constraint is satisfied throughout the flow. □

## D  PROOF OF THEOREM 1

In this section, we will prove Theorem 1. The proof consists of four parts.

1. We first construct a system of differential equations for the characteristic functions (Stieltjes transforms) associated with (4), which are more tractable.

2. We establish the existence and uniqueness of the solution with a specific integral representation.

3. Using Lemma 4, we develop a deterministic approximation for the original system and obtain a closed-form solution.

4. Leveraging the integral representation and the uniqueness property from Step 2, we show that the solution to the original system converges almost surely to that of the approximate system.

### D.1 Derivation of a Tractable System of Differential Equations

We start by expanding the gradient flow in (4) to

$$
\frac{\mathrm{d}}{\mathrm{d}t}\begin{bmatrix}\boldsymbol{u}_t \\ \boldsymbol{v}_t\end{bmatrix} = \begin{bmatrix} \boldsymbol{X}\boldsymbol{v}_t - \boldsymbol{u}_t\boldsymbol{u}_t^\top \boldsymbol{X}\boldsymbol{v}_t \\ \boldsymbol{X}^\top \boldsymbol{u}_t - \boldsymbol{v}_t\boldsymbol{v}_t^\top \boldsymbol{X}^\top \boldsymbol{u}_t \end{bmatrix}
$$

$$
= \begin{bmatrix} \boldsymbol{Z}\boldsymbol{v}_t + \lambda\boldsymbol{u}^*\boldsymbol{v}^{*\top}\boldsymbol{v}_t - \boldsymbol{u}_t\boldsymbol{u}_t^\top \boldsymbol{Z}\boldsymbol{v}_t - \lambda\boldsymbol{u}_t\boldsymbol{u}_t^\top \boldsymbol{u}^*\boldsymbol{v}^{*\top}\boldsymbol{v}_t \\ \boldsymbol{Z}^\top \boldsymbol{u}_t + \lambda\boldsymbol{v}^*\boldsymbol{u}^{*\top}\boldsymbol{u}_t - \boldsymbol{v}_t\boldsymbol{v}_t^\top \boldsymbol{Z}^\top \boldsymbol{u}_t - \lambda\boldsymbol{v}_t\boldsymbol{v}_t^\top \boldsymbol{v}^*\boldsymbol{u}^{*\top}\boldsymbol{u}_t \end{bmatrix}
$$

$$
= \begin{bmatrix} \boldsymbol{Z}\boldsymbol{v}_t + \lambda\boldsymbol{u}^* q_v(t) - f_{u,v,\lambda}(t)\boldsymbol{u}_t \\ \boldsymbol{Z}^\top \boldsymbol{u}_t + \lambda\boldsymbol{v}^* q_u(t) - f_{u,v,\lambda}(t)\boldsymbol{v}_t \end{bmatrix}, \tag{42}
$$

where $f_{u,v,\lambda}(t) = g_{u,v}(t) + \lambda q_u(t)q_v(t)$ and $g_{u,v}(t) = \langle \boldsymbol{u}_t, \boldsymbol{Z}\boldsymbol{v}_t\rangle$. Directly solving the above system is challenging due to its highly coupled nature. To this end, we define the inner products $(q_u, q_v)$ and the resolvent-related functions as follows

$$
C_u(z) = \langle \boldsymbol{u}^*, \mathbf{Q}(z)\boldsymbol{u}^*\rangle, \quad C_v(z) = \langle \boldsymbol{v}^*, \mathbf{Q}(z)\boldsymbol{v}^*\rangle, \quad D_{u,v}(z) = \langle \boldsymbol{u}^*, \mathbf{Q}(z)\boldsymbol{Z}\boldsymbol{v}^*\rangle,
$$

$$
G_u(t,z) = \langle \boldsymbol{u}_t, \mathbf{Q}(z)\boldsymbol{u}^*\rangle, \quad G_v(t,z) = \langle \boldsymbol{v}_t, \underline{\mathbf{Q}}(z)\boldsymbol{v}^*\rangle, \quad \Xi_u(t,z) = \langle \boldsymbol{u}_t, \mathbf{Q}(z)\boldsymbol{Z}\boldsymbol{v}^*\rangle,
$$

$$
\Xi_v(t,z) = \langle \boldsymbol{u}^*, \mathbf{Q}(z)\boldsymbol{Z}\boldsymbol{v}_t\rangle, \quad \Upsilon_u(t,z) = \langle \boldsymbol{u}_t, \mathbf{Q}(z)\boldsymbol{u}_t\rangle, \quad \Upsilon_v(t,z) = \langle \boldsymbol{v}_t, \underline{\mathbf{Q}}(z)\boldsymbol{v}_t\rangle,
$$

$$
\Pi_{u,v}(t,z) = \langle \boldsymbol{u}_t, \mathbf{Q}(z)\boldsymbol{Z}\boldsymbol{v}_t\rangle. \tag{43}
$$

The quantities defined in (43) can be viewed as the characteristic functions (or Stieltjes transforms) associated with the inner products. To illustrate this, consider $G_u(t,z)$ as an example. By selecting a positively oriented contour $\Gamma$ that encloses all the eigenvalues of $\boldsymbol{Z}\boldsymbol{Z}^\top$, and applying Cauchy's integral formula, we obtain

$$
-\frac{1}{2\pi\jmath}\oint_\Gamma G_u(t,z) = q_u(t). \tag{44}
$$

To simplify notation, we define the symbolic variables set $\mathcal{S}_{ode} = \{G_u, G_v, \Xi_u, \Xi_v, \Upsilon_u, \Upsilon_v, \Pi_{u,v}\}$. Using (4), we can derive the differential equations for each $S \in \mathcal{S}_{ode}$. Formally, we introduce the following system of differential equations for the complex-valued functions $G_u, G_v, \Xi_u, \Xi_v, \Pi_{u,v}, \Upsilon_u, \Upsilon_v : \mathbb{R}_+ \times \mathbb{C} \mapsto \mathbb{C}$

$$
\frac{\partial}{\partial t}G_u(t,z) = \Xi_v(t,z) + \lambda q_v(t)C_u(z) - f_{u,v,\lambda}(t)G_u(t,z), \tag{45a}
$$

$$
\frac{\partial}{\partial t}G_v(t,z) = \Xi_u(t,z) + \lambda q_u(t)C_v(z) - f_{u,v,\lambda}(t)G_v(t,z), \tag{45b}
$$

$$
\frac{\partial}{\partial t}\Xi_u(t,z) = q_v(t) + zG_v(t,z) + \lambda q_v(t)D_{u,v}(z) - f_{u,v,\lambda}(t)\Xi_u(t,z), \tag{45c}
$$

$$
\frac{\partial}{\partial t}\Xi_v(t,z) = q_u(t) + zG_u(t,z) + \lambda q_u(t)D_{u,v}(z) - f_{u,v,\lambda}(t)\Xi_v(t,z), \tag{45d}
$$

$$
\frac{\partial}{\partial t}\Pi_{u,v}(t,z) = 2 + z[\Upsilon_u(t,z) + \Upsilon_v(t,z)] + \lambda[q_v(t)\Xi_v(t,z) + q_u(t)\Xi_u(t,z)]
$$
$$
- 2f_{u,v,\lambda}(t)\Pi_{u,v}(t,z), \tag{45e}
$$

$$
\frac{1}{2}\cdot\frac{\partial}{\partial t}\Upsilon_u(t,z) = \Pi_{u,v}(t,z) + \lambda q_v(t)G_u(t,z) - f_{u,v,\lambda}(t)\Upsilon_u(t,z), \tag{45f}
$$

$$
\frac{1}{2}\cdot\frac{\partial}{\partial t}\Upsilon_v(t,z) = \Pi_{u,v}(t,z) + \lambda q_u(t)G_v(t,z) - f_{u,v,\lambda}(t)\Upsilon_v(t,z), \tag{45g}
$$

subjected to the initial conditions

$$
S(0,z) = B_S(z), \quad \forall S \in \mathcal{S}_{ode}. \tag{46}
$$

There exists a bounded interval $I \subset \mathbb{R}$ such that for any given $t \in \mathbb{R}_+$ and $S \in \mathcal{S}_{ode}$, $z \mapsto S(t,z)$ is analytical on $\mathbb{C}\backslash I$. Additionally, the following constraints hold for all $t \in \mathbb{R}_+$ and any positively oriented contour $\Gamma$ enclosing $I$

$$
f_{u,v,\lambda}(t) = g_{u,v}(t) + \lambda q_u(t)q_v(t),
$$

$$
q_u(t) = -\frac{1}{2\pi\jmath}\oint_\Gamma G_u(t,z)\mathrm{d}z, \quad q_v(z) = -\frac{1}{2\pi\jmath}\oint_\Gamma G_v(t,z)\mathrm{d}z,
$$

$$
g_{u,v}(t) = -\frac{1}{2\pi\jmath}\oint_\Gamma \Pi_{u,v}(t,z)\mathrm{d}z, \quad -\frac{1}{2\pi\jmath}\oint_\Gamma \Upsilon_u(t,z)\mathrm{d}z = -\frac{1}{2\pi\jmath}\oint_\Gamma \Upsilon_v(t,z)\mathrm{d}z = 1. \tag{47}
$$

### D.2 Existence and Uniqueness of the Solution

A fundamental question is the structure of the solution to the system of differential equations (45). In the following theorem, we show that if the initial conditions $B_S(z)$ for $S \in \mathcal{S}_{ode}$, together with the functions $C_u(z), C_v(z)$, and $D_{u,v}(z)$, admit certain integral representations (which include (43) as a special case), then the solution to (45) exists and is unique within a certain family.

Before delving into the details, we introduce the definition of a "good transition kernel". Specifically, a function $\mu : \mathbb{R}_+ \times \mathcal{B}(\mathbb{R}) \mapsto \mathbb{R}$ is called a good transition kernel if and only if

- For each $t \in \mathbb{R}_+$, $\mu(t, \cdot)$ is a signed measure on $\mathbb{R}_+$ with bounded total variation.
- The support is uniformly bounded: there exist $E < \infty$ such that

$$\sup_{t \geqslant 0} \sup\{\text{supp}(\mu(t, \cdot))\} \leqslant E. \tag{48}$$

- For every Borel set $A \in \mathcal{B}(\mathbb{R})$, the mapping $\mu(\cdot, A)$ is continuous in the sense that

$$\lim_{\delta \to 0} \|\mu(t, \cdot) - \mu(t + \delta, \cdot)\|_{TV} = 0. \tag{49}$$

Define the family of functions $\mathbb{S}_{\mathcal{K}}$ as

$$\mathbb{S}_{\mathcal{K}} = \left\{ f : \mathbb{R}_+ \times \mathbb{C} \mapsto \mathbb{C} \middle| f(t, z) = \int_{\mathbb{R}_+} \frac{\mu(t, \mathrm{d}x)}{x - z}, \mu \text{ is a good transition kernel} \right\}. \tag{50}$$

**Theorem 3.** *Assume that functions $C_u, C_v$, and $D_{u,v}$ admit the following integral representations*

$$C_u(z) = \int_{\mathbb{R}} \frac{\mu_{C_u}(\mathrm{d}x)}{x - z}, \quad C_v(z) = \int_{\mathbb{R}} \frac{\mu_{C_v}(\mathrm{d}x)}{x - z}, \quad D_{u,v}(z) = \int_{\mathbb{R}} \frac{\mu_{D_{u,v}}(\mathrm{d}x)}{x - z}, \tag{51}$$

*where $\mu_{C,u}, \mu_{C,v}$, and $\mu_{D,u,v}$ are non-negative finite measures supported over $\mathbb{R}_+$. Moreover, their support is bounded, i.e, $\text{supp}(\mu_{C_u}), \text{supp}(\mu_{C_v})$, and $\text{supp}(\mu_{D_{u,v}}) \subset [0, E]$ for some $E < \infty$ and $z \in \mathbb{C} \backslash [0, E]$. Further, assume the initial conditions $B_S$, for $S \in \mathcal{S}_{ode}$ also have the integral representations of the form*

$$B_S(z) = \int_{\mathbb{R}} \frac{\nu_S(\mathrm{d}x)}{x - z}, \tag{52}$$

*where $\nu_S$s are finite signed measures supported in $[0, E]$ and satisfy $\nu_{\Upsilon_u}(\mathbb{R}_+) = \nu_{\Upsilon_v}(\mathbb{R}_+) = 1$. Then, the system of equations (45) admits a unique solution $(\overline{G}_u, \overline{G}_v, \overline{\Xi}_u, \overline{\Xi}_v, \overline{\Pi}_{u,v}, \overline{\Upsilon}_u, \overline{\Upsilon}_v) \in \mathbb{S}_{\mathcal{K}}^7$.*

*Proof:* We first establish the existence of local solutions via Picard iteration by inductively constructing the corresponding integral representations and proving the convergence of the underlying measures. Choose the initial functions $S^0 \in \mathbb{S}_{\mathcal{K}}$ for all $S \in \mathcal{S}_{ode}$ represented by good transition kernels $\mu_S^0$s, such that

$$S^0(t, z) = \int_{\mathbb{R}} \frac{\mu_S^0(t, \mathrm{d}x)}{x - z}, \quad \forall S \in \mathcal{S}_{ode}, \tag{53}$$

with $\mu_{\Upsilon_u}^0(t, \mathbb{R}_+) = \mu_{\Upsilon_v}^0(t, \mathbb{R}_+) = 1$ for all $t \geqslant 0$ and $\mu_S^0(0, \cdot) = \nu_S(\cdot)$ for all $S \in \mathcal{S}_{ode}$. Suppose the induction hypothesis holds for $k$. We then show that 1) $S^{k+1} \in \mathbb{S}_{\mathcal{K}}$; 2) the support of the underline measures $\mu_S^{k+1}$s are bounded by $E$, and 3) $\mu_{\Upsilon_u}^{k+1}(t, \mathbb{R}_+) = \mu_{\Upsilon_v}^{k+1}(t, \mathbb{R}_+) = 1$ under Picard's iteration scheme. In particular, the iterate $G_u$ is given by

$$G_u^{k+1}(t, z) = B_{G_u}(z) + \int_0^t \Xi_v^k(a, z) + \lambda q_v^k(a) C_u(z) - f_{u,v,\lambda}^k(a) G_u^k(a, z) \mathrm{d}a. \tag{54}$$

By the induction hypothesis, we can obtain

$$G_u^{k+1}(t, z) = \int_{\mathbb{R}_+} \frac{\nu_{G_u}(\mathrm{d}x)}{x - z} + \int_0^t \int_{\mathbb{R}_+} \frac{\mu_{\Xi_v}^k(a, \mathrm{d}x)\mathrm{d}a}{x - z} + \lambda \int_0^t q_v^k(a)\mathrm{d}a \int_{\mathbb{R}_+} \frac{\mu_{C_u}(\mathrm{d}x)}{x - z}$$
$$- \int_0^t \int_{\mathbb{R}_+} \frac{f_{u,v,\lambda}^k(a)\mu_{G_u}^k(a, \mathrm{d}x)\mathrm{d}a}{x - z}, \tag{55}$$

where $q_u^k(t) = \mu_{G_u}^k(t, \mathbb{R})$, $q_v^k(t) = \mu_{G_v}^k(t, \mathbb{R})$, $g_{u,v}^k(t) = \mu_{\Pi_{u,v}}^k(t, \mathbb{R})$, and $f_{u,v,\lambda}^k(a) = g_{u,v}^k(a) + \lambda q_u^k(a)q_v^k(a)$. By Fubini's theorem, we can construct the following iterative scheme for $\mu_{G_u}$

$$\mu_{G_u}^{k+1}(t, A) = \nu_{G_u}(A) + \int_0^t \mu_{\Xi_v}^k(a, A) + \lambda q_v^k(a)\mu_{C_u}(A) - f_{u,v,\lambda}^k(a)\mu_{G_u}^k(a, A)\mathrm{d}a, \qquad (56)$$

for any Borel set $A \in \mathcal{B}(\mathbb{R})$. The total variation for $\mu_{G_u}^{k+1}(t, \cdot)$ can be bounded as

$$\left\|\mu_{G_u}^{k+1}(t, \cdot)\right\|_{TV} \leqslant \left\|\nu_{G_u}\right\|_{TV} + \int_0^t \left\|\mu_{G_u}^k(a, \cdot)\right\|_{TV} + |q_v^k(a)| \left\|\mu_{C_u}\right\|_{TV}$$
$$+ |f_{u,v,\lambda}^k(a)| \left\|\mu_{G_u}^k(a, \cdot)\right\|_{TV} \mathrm{d}a, \qquad (57)$$

which is finite. The continuity condition (49) follows directly from the definition of the iteration scheme. Since $\mu_S^k(a, \cdot)$ and $\nu_{G_u}(\cdot)$ are supported over $[0, E]$, the measure $\mu_{G_u}^{k+1}(t, \cdot)$ also supported over $[0, E]$. Therefore, $\mu_{G_u}^{k+1}$ is a good transition kernel and $G_u^{k+1} \in \mathbb{S}_{\mathcal{K}}$. By construction, the relation $S^{k+1} \in \mathbb{S}_{\mathcal{K}}$ for the remaining terms $S \in \mathcal{S}_{ode}$ can be shown in a similar manner, and we directly give the iteration scheme for the transition kernels as follows

$$\mu_{G_u}^{k+1}(t, A) = \nu_{G_u}(A) + \int_0^t \mu_{\Xi_v}^k(a, A) + \lambda q_v^k(a)\mu_{C_u}(A) - f_{u,v,\lambda}^k(a)\mu_{G_u}^k(a, A)\mathrm{d}a, \qquad (58a)$$

$$\mu_{G_v}^{k+1}(t, A) = \nu_{G_v}(A) + \int_0^t \mu_{\Xi_u}^k(a, A) + \lambda q_u^k(a)\mu_{C_v}(A) - f_{u,v,\lambda}^k(a)\mu_{G_v}^k(a, A)\mathrm{d}a, \qquad (58b)$$

$$\mu_{\Xi_u}^{k+1}(t, A) = \nu_{\Xi_u}(A) + \int_0^t \mathrm{d}a \int_A x\mu_{G_v}^k(a, \mathrm{d}x) + \lambda q_v^k(a)\mu_{D_{u,v}}(\mathrm{d}x) - f_{u,v,\lambda}^k(a)\mu_{\Xi_u}^k(a, \mathrm{d}x), \qquad (58c)$$

$$\mu_{\Xi_v}^{k+1}(t, A) = \nu_{\Xi_v}(A) + \int_0^t \mathrm{d}a \int_A x\mu_{G_u}^k(a, \mathrm{d}x) + \lambda q_u^k(a)\mu_{D_{u,v}}(\mathrm{d}x) - f_{u,v,\lambda}^k(a)\mu_{\Xi_v}^k(a, \mathrm{d}x), \qquad (58d)$$

$$\mu_{\Pi_{u,v}}^{k+1}(t, A) = \nu_{\Pi_{u,v}}(A) + \int_0^t \mathrm{d}a \int_A x\mu_{\Upsilon_u}^k(a, \mathrm{d}x) + x\mu_{\Upsilon_v}^k(a, \mathrm{d}x) + \lambda q_v^k(a)\mu_{\Xi_v}^k(a, \mathrm{d}x)$$
$$+ \lambda q_u^k(a)\mu_{\Xi_u}^k(a, \mathrm{d}x) - 2f_{u,v}^k(a)\mu_{\Pi_{u,v}}^k(a, \mathrm{d}x), \qquad (58e)$$

$$\mu_{\Upsilon_u}^{k+1}(t, A) = \nu_{\Upsilon_u}(A) + 2\int_0^t \mathrm{d}a \int_A \mu_{\Pi_{u,v}}^k(a, \mathrm{d}x) + \lambda q_v^k(a)\mu_{G_u}^k(a, \mathrm{d}x) - f_{u,v,\lambda}^k(a)\mu_{\Upsilon_u}^k(a, \mathrm{d}x), \qquad (58f)$$

$$\mu_{\Upsilon_v}^{k+1}(t, A) = \nu_{\Upsilon_v}(A) + 2\int_0^t \mathrm{d}a \int_A \mu_{\Pi_{u,v}}^k(a, \mathrm{d}x) + \lambda q_u^k(a)\mu_{G_u}^k(a, \mathrm{d}x) - f_{u,v,\lambda}^k(a)\mu_{\Upsilon_v}^k(a, \mathrm{d}x). \qquad (58g)$$

Obviously, for any $\mu_S^{k+1}$, $S \in \mathcal{S}_{ode}$, the support of the measure $\mu_S^{k+1}(t, \cdot)$ is contained in $[0, E]$. Here, to show that $\mu_{\Xi_v}^{k+1}$ is a good transition kernel (the argument for $\mu_{\Xi_u}^{k+1}$ and $\mu_{\Pi_{u,v}}^{k+1}$ is analogous), we use the fact $x \leqslant E$ and

$$\left\|x\mu_{G_v}^k(t, \cdot)\right\|_{TV} = \sup_A \left|\int_A x\mu_{G_v}^k(t, \mathrm{d}x)\right| \leqslant E \sup_A \left|\int_A \mu_{G_v}^k(t, \mathrm{d}x)\right| = E \left\|\mu_{G_v}^k(t, \cdot)\right\|_{TV}. \quad (59)$$

By the induction hypothesis $\mu_{\Upsilon_u}^k(t, \mathbb{R}_+) = \mu_{\Upsilon_v}^k(t, \mathbb{R}_+) = 1$, we have

$$\mu_{\Upsilon_u}^{k+1}(t, \mathbb{R}_+) = \nu_{\Upsilon_u}(\mathbb{R}_+) + 2\int_0^t \mu_{\Pi_{u,v}}^k(a, \mathbb{R}_+) + \lambda q_v^k(a)\mu_{G_u}^k(a, \mathbb{R}_+) - f_{u,v,\lambda}^k(a)\mu_{\Upsilon_u}^k(a, \mathbb{R}_+)\mathrm{d}a$$

$$= 1 + 2\int_0^t g_{u,v}^k(a) + \lambda q_v^k(a)q_u^k(a) - f_{u,v,\lambda}^k(a)\mathrm{d}a = 1. \qquad (60)$$

The relation $\mu_{\Upsilon_v}^{k+1}(t, \mathbb{R}_+) = 1$ for any $t$ follows analogously. This completes the inductive step, and we have thus constructed sequences $\{S^k\}_{k \geqslant 0}$ in $\mathbb{S}_{\mathcal{K}}$.

Next, we will prove the existence of the local solutions. By the continuity of the kernels, we can choose a sufficiently small positive number $T > 0$ such that for any $k \geqslant 0$, the following holds

$$\sup_{0 \leqslant t \leqslant T} \max \left\{ \left\| \mu_S^k(t, \cdot) - \nu_S(\cdot) \right\|_{TV}, \left\| \mu_S^k(t, \cdot) \right\| \right\}_{TV} \leqslant U, \ \ \forall S \in \mathcal{S}_{ode},$$

$$\sup_{0 \leqslant t \leqslant T} |f_{u,v,\lambda}^k(t)| \leqslant U, \tag{61}$$

where $U > 0$ is a constant. In fact, the bounds (61) are proved by induction. Assume it holds for some $k \geqslant 0$. Then for $k + 1$ and any Borel set $A \subset \mathbb{R}$, we have

$$\begin{aligned}
\left| \mu_{G_u}^{k+1}(T, A) - \nu_{G_u}(A) \right| &\leqslant \int_0^T \left| \mu_{\Xi_v}^k(a, A) \right| + |q_v^k(a)| |\mu_{C_u}(A)| \\
&\quad + \left| f_{u,v,\lambda}^k(a) \right| \left| \mu_{G_u}^k(a, A) - \nu_{G_u}(A) \right| + \left| f_{u,v,\lambda}^k(a) \right| |\nu_{G_u}(A)| \, da \\
&\leqslant T \left( U + U \left\| \mu_{C_u} \right\|_{TV} + U^2 + U \left\| \nu_{G_u} \right\|_{TV} \right). \tag{62}
\end{aligned}$$

Now, choose $T$ such that $T \leqslant (1 + \|\mu_{C_u}\|_{TV} + U + \|\nu_{G_u}\|_{TV})^{-1}$ (note that since $\mu_S^0(t, \cdot)$s are good transition kernels, such a $T$ exists). Then, for this choice of $T$, we obtain $\left| \mu_{G_u}^{k+1}(T, A) - \nu_S(A) \right| \leqslant U$ for any Borel set $A$. The bounds for the other measures follow similarly and the details are omitted.

We now prove the convergence of the measures on the interval $[0, T]$. To this end, define

$$M_k(t) = \max_{S \in \mathcal{S}_{ode}} \left\{ \left\| \mu_S^{k+1}(t, \cdot) - \mu_S^k(t, \cdot) \right\|_{TV} \right\}. \tag{63}$$

Hence, for each $k$ and $a \in [0, T]$, the following estimates hold

$$\begin{aligned}
\left| g_{u,v}^k(a) - g_{u,v}^{k-1}(a) \right| &= \left| \mu_{\Pi_{u,v}}^k(a, \mathbb{R}) - \mu_{\Pi_{u,v}}^{k-1}(a, \mathbb{R}) \right| \leqslant M_{k-1}(a), \\
\left| f_{u,v,\lambda}^k(a) - f_{u,v,\lambda}^{k-1}(a) \right| &\leqslant (2\lambda U + 1) M_{k-1}(a). \tag{64}
\end{aligned}$$

Consider the difference of successive iterates for $\mu_{G_u}$

$$\begin{aligned}
\left| \mu_{G_u}^{k+1}(t, A) - \mu_{G_u}^k(t, A) \right| &= \left| \int_0^t \mu_{\Xi_v}^k(a, A) - \mu_{\Xi_v}^{k-1}(a, A) \right. \\
&\quad + q_v^k(a) \mu_{C_u}(A) - q_v^{k-1}(a) \mu_{C_u}(A) \\
&\quad \left. - \left( f_{u,v,\lambda}^k(a) \mu_{G_u}^k(a, A) - f_{u,v,\lambda}^{k-1}(a) \mu_{G_u}^{k-1}(a, A) \right) da \right| \\
&\leqslant \int_0^t \left[ 1 + |\mu_{C_u}|(A) + U(2\lambda U + 1) + U \right] M_{k-1}(a). \tag{65}
\end{aligned}$$

Taking the supreme over all Borel sets $A$, we obtain for all $t \leqslant T$

$$\left\| \mu_{G_u}^{k+1}(t, \cdot) - \mu_{G_u}^k(t, \cdot) \right\|_{TV} \leqslant C \int_0^t M_{k-1}(a), \tag{66}$$

where the constant $C = 1 + \|\mu_{C_u}\|_{TV} + U(2\lambda U + 1) + U$. By analyzing the remaining terms $S \in \mathcal{S}_{ode}$, we can get similar inequalities with different constants $C$. Combining the estimates for all $S \in \mathcal{S}_{ode}$ and using the definition of $M_k(t)$, we obtain

$$M_k(t) \leqslant \widetilde{C} \int_0^t M_{k-1}(a), \tag{67}$$

where the constant $\widetilde{C}$ is independent of $k$. Since $M_0(a) \leqslant 2U$, an induction argument shows that for every $k \geqslant 1$

$$\sup_{0 \leqslant t \leqslant T} M_k(t) \leqslant \frac{2U(\widetilde{C}T)^k}{k!}. \tag{68}$$

In particular, we have $\sum_{k\geqslant 1}\sup_{0\leqslant t\leqslant T}M_k(t) < \infty$. Therefore, for each $t \in [0,T]$, the sequence of measures $\{\mu_S^k(t,\cdot)\}$ converges in total variation to a limit measure $\mu_S^\infty(t,\cdot)$, for every $S \in \mathcal{S}_{ode}$. Passing to the limit $k \to \infty$ in the iterative scheme (58) and using the integral representation, one can verify that the functions defined by

$$\overline{S}(t,z) = \int_{\mathbb{R}} \frac{\mu_S^\infty(t,\mathrm{d}x)}{x-z}, \ \ t \in [0,T], \ z \in \mathbb{C}\backslash[0,E], \tag{69}$$

satisfy the system (45) for all $S \in \mathcal{S}_{ode}$. The convergence of total variation implies that each $\mu_S^\infty(t,\cdot)$ is finite and supported within $[0,E]$. Moreover, since the convergence is uniform in $t$, i.e., $\sup_{t\leqslant T}\left\|\mu_S^k(t,\cdot) - \mu_S^\infty(t,\cdot)\right\|_{TV} \to 0$ as $k \to \infty$, it follows that $\mu_S^\infty(t,\cdot)$ is continuous in the sense that $\lim_{\delta\to 0}\|\mu_S^\infty(t,\cdot) - \mu_S^\infty(t+\delta,\cdot)\|_{TV} = 0$ for all $t \in [0,T]$ and $S \in \mathcal{S}_{ode}$. This implies $\mu_S^\infty$, $S \in \mathcal{S}_{ode}$ are good transition kernels.

To prove uniqueness, we argue by contradiction. Suppose there exist two distinct solutions $S_1$ and $S_2$ belong to $\mathbb{S}_\mathcal{K}$ with underlying measures $\mu_{S,1}$ and $\mu_{S,2}$ for each $S \in \mathcal{S}_{ode}$. Denote

$$M(t) = \sup_{S\in\mathcal{S}_{ode}} \left\{\|\mu_{S,1}(t,\cdot) - \mu_{S,2}(t,\cdot)\|_{TV}\right\}. \tag{70}$$

By the inversion formula of the Stieltjes transform (Hachem et al., 2007), for $i = 1, 2$ and any interval $[a,b] \subset \mathbb{R}$, we have

$$\mu_{S,i}(t,[a,b]) + \frac{\mu_{S,i}(t,\{a,b\})}{2} = \frac{1}{\pi}\lim_{\beta\downarrow 0}\int_a^b \Im\left\{S_i(t,\alpha+\jmath\beta)\right\}\mathrm{d}\alpha. \tag{71}$$

By considering the difference between the two equations associated with $S_1$ and $S_2$, and using the inversion formula (71) and together with a standard $\pi$-$\lambda$ argument (Le Gall, 2016), we conclude that $M(t)$ satisfies

$$M(t) \leqslant C\int_0^t M(a)\mathrm{d}a, \ \ \forall t \in [0,T], \tag{72}$$

for some constant $C > 0$. Since $\|\mu_{S,i}(t,\cdot)\|_{TV}$, $i = 1, 2, S \in \mathcal{S}_{ode}$ is bounded, it follows from Grönwall's lemma (Lemma 1) that $M(t) = 0$ for every $t \in [0,T]$, which establishes local uniqueness. We have thus demonstrated the existence and uniqueness of the solution on the interval $[0,T]$. Using a standard extension argument for solutions of differential equations (Hartman, 2002), the solution can be extended to all $t > 0$, and we omit the details. Therefore, we have completed the proof. $\qquad\square$

It can be verified that the solution to the system of differential equations (42) exists uniquely. Substituting this solution $(\boldsymbol{u}_t, \boldsymbol{v}_t)$ into (43) yields $S(t,z)$ for each $S \in \mathcal{S}_{ode}$. Moreover, the functions $B_S, C_u, C_v$ and $D_{u,v}$ satisfy the conditions of Theorem 3 with probability one. Consequently, the solution $\{S(t,z), S \in \mathcal{S}_{ode}\}$ coincides with the solution $\{\overline{S}(t,z), S \in \mathcal{S}_{ode}\}$ given in Theorem 3. For example, let the SVD of $\boldsymbol{Z}$ be given by $\boldsymbol{Z} = \boldsymbol{U}\boldsymbol{\Lambda}\boldsymbol{V}^\top = \sum_j \sigma_j \boldsymbol{u}_j \boldsymbol{v}_j^\top$. Then, we have

$$C_u(z) = \langle \boldsymbol{u}^*, \mathbf{Q}(z)\boldsymbol{u}^*\rangle = \sum_{j=1}^p \frac{[\boldsymbol{U}^\top\boldsymbol{u}^*]_j^2}{\sigma_j^2 - z} = \int_{\mathbb{R}} \frac{F_{C_u}(\mathrm{d}x)}{x-z}, \tag{73}$$

where the distribution function is given by $F_{C_u}(x) = \sum_{j=1}^p[\boldsymbol{U}^\top\boldsymbol{u}^*]_j^2\mathbb{1}\{\sigma_j^2 \leqslant x\}$. The integral representations for the other terms $S \in \mathcal{S}_{ode}$ can be verified similarly and details are omitted for brevity.

We note that the agreement between these two solutions indicates that the solution of (45) within $\mathbb{S}_\mathcal{K}$ accurately captures the learning dynamics of matrix denoising.

### D.3 CLOSED-FORM SOLUTION TO THE APPROXIMATED SYSTEM

Although Theorem 3 characterizes the solution structure of (45), obtaining a closed-form solution is still infeasible. This is due to the fact that the functions $C_u$, $C_v$, and $D_{u,v}$, as well as the initial conditions $B_S$, are random and lack explicit expressions.

However, in the high-dimensional setting, Lemma 4 implies these quantities will converge to deterministic limits. Specifically, as $p, n \to \infty$, for any $z \in \mathbb{C} \backslash \mathcal{M}$, we have

$$C_u(z) \xrightarrow[p,n\to\infty]{a.s.} m(z), \quad C_v(z) \xrightarrow[p,n\to\infty]{a.s.} \underline{m}(z), \quad D_{u,v}(z) \xrightarrow[p,n\to\infty]{a.s.} 0, \tag{74}$$

and

$$[B_{G_u}(z), B_{G_v}(z), B_{\Upsilon_u}(z), B_{\Upsilon_v}(z), B_{\Xi_u}(z), B_{\Xi_v}(z), B_{\Pi_{u,v}}(z)]$$
$$\xrightarrow{a.s.} [\alpha_u m(z), \alpha_v \underline{m}(z), m(z), \underline{m}(z), 0, 0, 0], \tag{75}$$

where $\alpha_u = \langle \boldsymbol{u}_0, \boldsymbol{u} \rangle$, $\alpha_v = \langle \boldsymbol{v}_0, \boldsymbol{v} \rangle$, and $m(z)$ and $\underline{m}(z)$ are defined in Lemma 4. Heuristically, we may substitute the random functions $B_S$, $C_u$, $C_v$, and $D_{u,v}$ with their deterministic limits and consider the following asymptotic system of differential equations

$$\frac{\partial}{\partial t} G_u(t, z) = \Xi_v(t, z) + \lambda q_v(t) m(z) - f_{u,v,\lambda}(t) G_u(t, z), \tag{76a}$$

$$\frac{\partial}{\partial t} G_v(t, z) = \Xi_u(t, z) + \lambda q_u(t) \underline{m}(z) - f_{u,v,\lambda}(t) G_v(t, z), \tag{76b}$$

$$\frac{\partial}{\partial t} \Xi_u(t, z) = q_v(t) + z G_v(t, z) - f_{u,v,\lambda}(t) \Xi_u(t, z), \tag{76c}$$

$$\frac{\partial}{\partial t} \Xi_v(t, z) = q_u(t) + z G_u(t, z) - f_{u,v,\lambda}(t) \Xi_v(t, z), \tag{76d}$$

$$\frac{\partial}{\partial t} \Pi_{u,v}(t, z) = 2 + z[\Upsilon_u(t, z) + \Upsilon_v(t, z)] + \lambda[q_v(t)\Xi_v(t, z) + q_u(t)\Xi_u(t, z)]$$
$$- 2f_{u,v,\lambda}(t)\Pi_{u,v}(t, z), \tag{76e}$$

$$\frac{1}{2} \cdot \frac{\partial}{\partial t} \Upsilon_u(t, z) = \Pi_{u,v}(t, z) + \lambda q_v(t) G_u(t, z) - f_{u,v,\lambda}(t) \Upsilon_u(t, z), \tag{76f}$$

$$\frac{1}{2} \cdot \frac{\partial}{\partial t} \Upsilon_v(t, z) = \Pi_{u,v}(t, z) + \lambda q_u(t) G_v(t, z) - f_{u,v,\lambda}(t) \Upsilon_v(t, z), \tag{76g}$$

with initial conditions given in (75) and subject to constraints (47). We note that, according to Lemma 4 and (75), the initial conditions $B_S$ and the functions $C_u$, $C_v$, and $D_{u,v}$ (where $\mu_{D_{u,v}}(t, \cdot) = 0$) satisfy the assumptions required by Theorem 3. Consequently, the system admits a unique solution within the family $\mathbb{S}_{\mathcal{K}}$. The remainder of this section is devoted to deriving the closed-form expressions for the components in (76). Note that our final objective is to solve for $q_u$ and $q_v$, since they characterize the learning dynamics.

We begin by simplifying the system of differential equations using the method of variation of parameters. To this end, we define the auxiliary function $F_{u,v,\lambda}(t) = \int_0^t f_{u,v,\lambda}(a) \mathrm{d}a$ and

$$G_h(t, z) = \widehat{G}_h(t, z) \exp\left(-F_{u,v,\lambda}(t)\right), \quad \Upsilon_h(t, z) = \widehat{\Upsilon}_h(t, z) \exp\left(-2F_{u,v,\lambda}(t)\right),$$
$$\Xi_h(t, z) = \widehat{\Xi}_h(t, z) \exp\left(-F_{u,v,\lambda}(t)\right), \quad q_h = \widehat{q}_h(t) \exp\left(-F_{u,v,\lambda}(t)\right), \quad h \in \{u, v\},$$
$$\Pi_{u,v}(t, z) = \widehat{\Pi}_{u,v}(t, z) \exp\left(-2F_{u,v,\lambda}(t)\right), \quad g_{u,v}(t) = \widehat{g}_{u,v}(t) \exp(-2F_{u,v}(t)). \tag{77}$$

To solve for $q_u$ and $q_v$, it suffices to analyze $\widehat{q}_u$, $\widehat{q}_v$, and $F_{u,v,\lambda}$. Substituting (77) into (76), we obtain the following system of differential equations

$$\frac{\partial}{\partial t} \widehat{G}_u(t, z) = \lambda \widehat{q}_v(t) m(z) + \widehat{\Xi}_v(t, z), \tag{78a}$$

$$\frac{\partial}{\partial t} \widehat{G}_v(t, z) = \lambda \widehat{q}_u(t) \underline{m}(z) + \widehat{\Xi}_u(t, z), \tag{78b}$$

$$\frac{\partial}{\partial t} \widehat{\Xi}_u(t, z) = \widehat{q}_v(t) + z \widehat{G}_v(t, z), \tag{78c}$$

$$\frac{\partial}{\partial t} \widehat{\Xi}_v(t, z) = \widehat{q}_u(t) + z \widehat{G}_u(t, z), \tag{78d}$$

$$\frac{\partial}{\partial t} \widehat{\Upsilon}_u(t, z) = 2\widehat{\Pi}_{u,v}(t, z) + 2\lambda \widehat{q}_v(t) \widehat{G}_u(t, z), \tag{78e}$$

$$\frac{\partial}{\partial t} \widehat{\Upsilon}_v(t, z) = 2\widehat{\Pi}_{u,v}(t, z) + 2\lambda \widehat{q}_u(t) \widehat{G}_v(t, z), \tag{78f}$$

$$\frac{\partial}{\partial t}\widehat{\Pi}_{u,v}(t,z) = e^{2F_{u,v,\lambda}(t)} + z\widehat{\Upsilon}_u(t,z) + z\widehat{\Upsilon}_v(t,z)$$
$$+ \lambda[\widehat{q}_v(t)\widehat{\Xi}_v(t,z) + \widehat{q}_u(t)\widehat{\Xi}_u(t,z)]. \tag{78g}$$

Next, we analyze the equations (78) in the Laplace domain. Here, for a function $f$, if there exists a constant $a$ with $\sup_{t\geqslant 0}|f(t)e^{-at}|$ being finite, its Laplace transform is defined as $\mathcal{L}[f](s) = \int_0^\infty f(t)e^{-st}\mathrm{d}t$ for $\Re(s) > a$. Treating $z$ as a constant and taking the Laplace transform with respect to $t$ on both sides of (78), the following linear system of equations is obtained

$$s\mathcal{L}[\widehat{G}_u](s,z) - \alpha_u m(z) = \lambda m(z)\mathcal{L}[\widehat{q}_v](s) + \mathcal{L}[\widehat{\Xi}_v](s,z), \tag{79a}$$

$$s\mathcal{L}[\widehat{G}_v](s,z) - \alpha_v \underline{m}(z) = \lambda\underline{m}(z)\mathcal{L}[\widehat{q}_u](s) + \mathcal{L}[\widehat{\Xi}_u](s,z), \tag{79b}$$

$$s\mathcal{L}[\widehat{\Xi}_u](s,z) = \mathcal{L}[\widehat{q}_v](s) + z\mathcal{L}[\widehat{G}_v](s,z), \tag{79c}$$

$$s\mathcal{L}[\widehat{\Xi}_v](s,z) = \mathcal{L}[\widehat{q}_u](s) + z\mathcal{L}[\widehat{G}_u](s,z), \tag{79d}$$

$$s\mathcal{L}[\widehat{\Upsilon}_u](s,z) - m(z) = 2\mathcal{L}[\widehat{\Pi}_{u,v}](s,z) + 2\lambda\mathcal{L}[\widehat{q}_v\widehat{G}_u](s,z), \tag{79e}$$

$$s\mathcal{L}[\widehat{\Upsilon}_v](s,z) - \underline{m}(z) = 2\mathcal{L}[\widehat{\Pi}_{u,v}](s,z) + 2\lambda\mathcal{L}[\widehat{q}_u\widehat{G}_v](s,z), \tag{79f}$$

$$s\mathcal{L}[\widehat{\Pi}_{u,v}](s,z) = \mathcal{L}[e^{2F_{u,v,\lambda}}](s) + z\mathcal{L}[\widehat{\Upsilon}_u](s,z) + z\mathcal{L}[\widehat{\Upsilon}_v](s,z)$$
$$+ \lambda[\mathcal{L}[\widehat{q}_v\widehat{\Xi}_v](s,z) + \mathcal{L}[\widehat{q}_u\widehat{\Xi}_u](s,z)]. \tag{79g}$$

We first handle (79a)-(79d). Solving $\mathcal{L}[\widehat{G}_u]$ and $\mathcal{L}[\widehat{G}_v]$, we have

$$\mathcal{L}[\widehat{G}_u](s,z) = \frac{\alpha_u s m(z) + \lambda s m(z)\mathcal{L}[\widehat{q}_v](s) + \mathcal{L}[\widehat{q}_u](s)}{s^2 - z}, \tag{80}$$

$$\mathcal{L}[\widehat{G}_v](s,z) = \frac{\alpha_v s\underline{m}(z) + \lambda s\underline{m}(z)\mathcal{L}[\widehat{q}_u](s) + \mathcal{L}[\widehat{q}_v](s)}{s^2 - z}. \tag{81}$$

Choose $s$ such that $s^2 \notin \mathcal{M}$. Let $\Gamma$ be a contour that encloses $s^2$ and does not intersect $\mathcal{M}$. Then, by performing a contour integral with respect to $z$ on both sides of (80) and (81), we obtain

$$0 = \alpha_u s m(s^2) + \lambda s m(s^2)\mathcal{L}[\widehat{q}_v](s) + \mathcal{L}[\widehat{q}_u](s), \tag{82}$$

$$0 = \alpha_v s\underline{m}(s^2) + \lambda s\underline{m}(s^2)\mathcal{L}[\widehat{q}_u](s) + \mathcal{L}[\widehat{q}_v](s). \tag{83}$$

Solving the above system of equations with respect to $\mathcal{L}[\widehat{q}_v]$ and $\mathcal{L}[\widehat{q}_u]$, we have

$$\mathcal{L}[\widehat{q}_u](s) = \frac{\alpha_v \lambda s^2 m(s^2)\underline{m}(s^2) - \alpha_u s m(s^2)}{1 - \lambda^2 s^2 m(s^2)\underline{m}(s^2)}, \tag{84}$$

$$\mathcal{L}[\widehat{q}_v](s) = \frac{\alpha_u \lambda s^2 m(s^2)\underline{m}(s^2) - \alpha_v s\underline{m}(s^2)}{1 - \lambda^2 s^2 m(s^2)\underline{m}(s^2)}. \tag{85}$$

We solve $\widehat{q}_u(t)$ first. According to the fundamental equation (37), we have $zm(z) + zm(z)\underline{m}(z) = -1$ and $z\underline{m}(z) + zcm(z)\underline{m}(z) = -1$. These identities yield

$$\mathcal{L}[\widehat{q}_u](s) = \frac{\alpha_v \lambda s^2 \underline{m}(s^2) - \alpha_u s}{\lambda^2} \frac{m(s^2)}{\frac{1}{\lambda^2} + 1 + s^2 m(s^2)}$$

$$\overset{(a)}{=} \frac{1 + \lambda^2}{\lambda^2} \frac{(\alpha_v \lambda s^2 \underline{m}(s^2) - \alpha_u s)(cm(s^2) + \frac{\lambda^2}{1+\lambda^2})}{(1 + \frac{c}{\lambda^2})(1 + \lambda^2) - s^2}$$

$$= \frac{\alpha_v(1 + \lambda^2)}{\lambda} \frac{s^2 cm(s^2)\underline{m}(s^2)}{\vartheta_\lambda - s^2} + \alpha_v\lambda\frac{s^2\underline{m}(s^2)}{\vartheta_\lambda - s^2} - \frac{\alpha_u c(1 + \lambda^2)}{\lambda^2}\frac{sm(s^2)}{\vartheta_\lambda - s^2} - \frac{\alpha_u s}{\vartheta_\lambda - s^2}$$

$$= -\frac{\alpha_v s^2 \underline{m}(s^2)}{\lambda(\vartheta_\lambda - s^2)} - \frac{\alpha_u c(1 + \lambda^2)}{\lambda^2}\frac{sm(s^2)}{\vartheta_\lambda - s^2} - \frac{\alpha_u s}{\vartheta_\lambda - s^2} - \frac{\alpha_v(1 + \lambda^2)}{\lambda(\vartheta_\lambda - s^2)}$$

$$= Q_{u,1} + Q_{u,2} + Q_{u,3} + Q_{u,4}, \tag{86}$$

where step $(a)$ follows from the identity $(zm(z) + 1 + \frac{1}{\lambda^2})(cm(z) + \frac{\lambda^2}{1+\lambda^2}) = m(z)(1 + \frac{c}{\lambda^2} - \frac{z}{1+\lambda^2})$. Next, we calculate the inverse Laplace transform for each term $Q_{u,j}$ separately. Beginning with $Q_{u,1}$, we have

$$\mathcal{L}^{-1}[Q_{u,1}] = -\frac{\alpha_v}{\lambda}\mathcal{L}^{-1}\left[\frac{s^2\underline{m}(s^2)}{\vartheta_\lambda - s^2}\right] = -\frac{\alpha_v}{\lambda}\mathcal{L}^{-1}\left[\int_{\mathbb{R}}\frac{s^2\underline{\mu}(\mathrm{d}x)}{(\vartheta_\lambda - s^2)(x - s^2)}\right]$$

$$\stackrel{(a)}{=} -\frac{\alpha_v}{\lambda}\mathcal{L}^{-1}\left[\frac{1}{s^2-\vartheta_\lambda}\right] - \frac{\alpha_v}{\lambda}\int_{\mathbb{R}}\mathcal{L}^{-1}\left[\frac{x}{(s^2-\vartheta_\lambda)(s^2-x)}\right]\underline{\mu}(\mathrm{d}x)$$

$$= -\frac{\alpha_v}{\lambda}\ell_{3,\vartheta_\lambda}(t) - \frac{\alpha_v}{\lambda}\int_{\mathbb{R}}x(\ell_{3,\vartheta_\lambda}*\ell_{3,x})(t)\underline{\mu}(\mathrm{d}x), \tag{87}$$

where step $(a)$ follows from Fubini's theorem. The inverse Laplace transform of $Q_{u,2}$ is given by

$$\mathcal{L}^{-1}[Q_{u,2}] = -\frac{\alpha_u c(1+\lambda^2)}{\lambda^2}\mathcal{L}^{-1}\left[\frac{sm(s^2)}{\vartheta_\lambda-s^2}\right] = -\frac{\alpha_u c(1+\lambda^2)}{\lambda^2}\mathcal{L}^{-1}\left[\int_{\mathbb{R}}\frac{s\mu(\mathrm{d}x)}{(\vartheta_\lambda-s^2)(x-s^2)}\right]$$

$$= -\frac{\alpha_u c(1+\lambda^2)}{\lambda^2}\int_{\mathbb{R}}(\ell_{1,\vartheta_\lambda}*\ell_{3,x})(t)\mu(\mathrm{d}x). \tag{88}$$

The inverse Laplace transforms of $Q_{2,u}$ and $Q_{3,u}$ can be obtained directly from standard tables: $\mathcal{L}^{-1}[Q_{u,3}] = \alpha_u\ell_{1,\vartheta_\lambda}(t)$ and $\mathcal{L}^{-1}[Q_{u,4}] = \frac{\alpha_v(1+\lambda^2)}{\lambda}\ell_{3,x}(t)$. Consequently, summing all the inverse transform results yields (8). The derivation for $\widehat{q}_v(t)$ is analogous, and we omit it for brevity. Note that by analyzing (79a)-(79d), explicit expressions for $\widehat{G}_h$ and $\widehat{\Xi}_h$, $h \in \{u,v\}$ can also be derived. However, these intermediate quantities are not directly relevant to the objects. Instead of solving them explicitly, we use their Laplace transforms.

In the following, we solve (79e)-(79g) to derive the explicit expression for $e^{F_{u,v,\lambda}(t)}$. Solving for $\mathcal{L}[\widehat{\Pi}_{u,v}]$, we have $\mathcal{L}[\widehat{\Pi}_{u,v}](s,z) = \Pi_1 + \Pi_2 + \Pi_3 + \Pi_4$, where

$$\Pi_1 = \frac{s\mathcal{L}[e^{2F_{u,v}}](s)}{s^2-4z}, \quad \Pi_2 = \frac{zm(z)+z\underline{m}(z)}{s^2-4z},$$

$$\Pi_3 = \frac{2\lambda z[\mathcal{L}[\widehat{q}_v\widehat{G}_u](s,z) + \mathcal{L}[\widehat{q}_u\widehat{G}_v](s,z)]}{s^2-4z},$$

$$\Pi_4 = \frac{\lambda s[\mathcal{L}[\widehat{q}_v\widehat{\Xi}_v](s,z) + \mathcal{L}[\widehat{q}_u\widehat{\Xi}_u](s,z)]}{s^2-4z}. \tag{89}$$

Let both $s^2/4, s^2 \notin \mathcal{M}$ and choose a contour $\Gamma$ that encloses the MP support $\mathcal{M}$ but excludes the points $s^2/4$ and $s^2$. We then analyze the contour integral of each term in (89). Clearly, $\frac{1}{2\pi\jmath}\oint_\Gamma \Pi_1 \mathrm{d}z = 0$ since $\Pi_1$ is analytic inside and on $\Gamma$. For term $\Pi_2$, we have

$$\frac{1}{2\pi\jmath}\oint_\Gamma \Pi_2\mathrm{d}z = \frac{1}{2\pi\jmath}\oint_\Gamma\frac{zm(z)+z\underline{m}(z)}{s^2-4z}\mathrm{d}z = \frac{1}{2\pi\jmath}\oint_\Gamma\frac{z}{s^2-4z}\int_{\mathbb{R}}\frac{[\mu+\underline{\mu}](\mathrm{d}x)}{x-z}\mathrm{d}z$$

$$= -\int_{\mathbb{R}}[\mu+\underline{\mu}](\mathrm{d}x)\frac{x}{s^2-4x}. \tag{90}$$

We now evaluate the integral of $\Pi_3$. By the definition of the Laplace transform and Fubini's theorem, we have

$$\frac{1}{2\pi\jmath}\oint_\Gamma \Pi_3\mathrm{d}z = \frac{1}{2\pi\jmath}\oint_\Gamma \mathrm{d}z\frac{2\lambda z\left[\mathcal{L}[\widehat{q}_v\widehat{G}_u](s,z) + \widehat{q}_u\widehat{G}_v](s,z)\right]}{s^2-4z}$$

$$= \frac{1}{2\pi\jmath}\oint_\Gamma \mathrm{d}z\int_0^\infty\frac{2\lambda z\left[\widehat{q}_v(t)\widehat{G}_u(t,z) + \widehat{q}_u(t)\widehat{G}_v(t,z)\right]}{s^2-4z}e^{-st}\mathrm{d}t$$

$$= 2\lambda\int_0^\infty\widehat{q}_v(t)\overline{\Pi}_{3,u}(t)e^{-st}\mathrm{d}t + 2\lambda\int_0^\infty\widehat{q}_u(t)\overline{\Pi}_{3,v}(t)e^{-st}\mathrm{d}t, \tag{91}$$

where $\overline{\Pi}_{3,u}(t) = \frac{1}{2\pi\jmath}\oint_\Gamma \mathrm{d}z\frac{z\widehat{G}_u(t,z)}{s^2-4z}$ and $\overline{\Pi}_{3,v}(t) = \frac{1}{2\pi\jmath}\oint_\Gamma \mathrm{d}z\frac{z\widehat{G}_v(t,z)}{s^2-4z}$. Next, we solve $\overline{\Pi}_{3,h}(t)$, $h \in \{u,v\}$. Due to their similarity, we will only focus on $\overline{\Pi}_{3,u}$. It is more convenient to handle its Laplace transform, which is given by

$$\mathcal{L}[\overline{\Pi}_{3,u}](r) = \mathcal{L}\left[\frac{1}{2\pi\jmath}\oint\frac{z\widehat{G}_u(t,z)}{s^2-4z}\mathrm{d}z\right](r) = \frac{1}{2\pi\jmath}\oint\frac{z\mathcal{L}[\widehat{G}_u](r,z)}{s^2-4z}\mathrm{d}z$$

$$\stackrel{(a)}{=} \frac{1}{2\pi\jmath}\oint_\Gamma\frac{z}{s^2-4z}\left(\frac{\alpha_u rm(z) + \lambda rm(z)\mathcal{L}[\widehat{q}_v](r) + \mathcal{L}[\widehat{q}_u](r)}{r^2-z}\right)\mathrm{d}z.$$

$$= \frac{\alpha_u r + \lambda \mathcal{L}[\widehat{q}_v](r) r}{2\pi\jmath} \oint_\Gamma \mathrm{d}z \int_\mathbb{R} \frac{z\mu(\mathrm{d}x)}{(s^2 - 4z)(r^2 - z)(x - z)}$$

$$= -\int_\mathbb{R} \frac{x(\alpha_u r + \lambda \mathcal{L}[\widehat{q}_v](r) r)\mu(\mathrm{d}x)}{(s^2 - 4x)(r^2 - x)}, \tag{92}$$

where we enforce $r^2 \notin \mathcal{M}$ and $r^2$ lies outside the contour $\Gamma$. In step $(a)$, we use the identity (80). Taking the inverse Laplace transform (with respect to the variable $r$) and applying Fubini's theorem, we get

$$\overline{\Pi}_{3,u} = -\int_\mathbb{R} \frac{\alpha_u x}{s^2 - 4x} \mathcal{L}^{-1}\left[\frac{r}{r^2 - x}\right]\mu(\mathrm{d}x) - \int_\mathbb{R} \frac{\lambda x}{s^2 - 4x} \mathcal{L}^{-1}\left[\frac{\mathcal{L}[\widehat{q}_v](r) r}{r^2 - x}\right]\mu(\mathrm{d}x)$$

$$\stackrel{(a)}{=} -\int_\mathbb{R} \frac{\alpha_u x \ell_{1,x}(t)}{s^2 - 4x}\mu(\mathrm{d}x) - \int_\mathbb{R} \frac{\lambda x (\widehat{q}_v * \ell_{1,x})(t)}{s^2 - 4x}\mu(\mathrm{d}x), \tag{93}$$

where step $(a)$ follows from the convolution property of the Laplace transform $\mathcal{L}(f*g) = \mathcal{L}(f)\mathcal{L}(g)$. Similarly, we can obtain

$$\overline{\Pi}_{3,v} = -\int_\mathbb{R} \frac{\alpha_v x \ell_{1,x}(t)}{s^2 - 4x}\mu(\mathrm{d}x) - \int_\mathbb{R} \frac{\lambda x (\widehat{q}_u * \ell_{1,x})(t)}{s^2 - 4x}\mu(\mathrm{d}x). \tag{94}$$

Substituting (93) and (94) into (91), we obtain

$$\frac{1}{2\pi\jmath} \oint_\Gamma \Pi_3 \mathrm{d}z = -2\lambda \int_0^\infty \int_\mathbb{R} \frac{\alpha_u x \widehat{q}_v(t)\ell_{1,x}(t) + \lambda x \widehat{q}_v(t)(\widehat{q}_v * \ell_{1,x})(t)}{s^2 - 4x}\mu(\mathrm{d}x)e^{-st}\mathrm{d}t$$

$$- 2\lambda \int_0^\infty \int_\mathbb{R} \frac{\alpha_v x \widehat{q}_u(t)\ell_{1,x}(t) + \lambda x \widehat{q}_u(t)(\widehat{q}_u * \ell_{1,x})(t)}{s^2 - 4x}\mu(\mathrm{d}x)e^{-st}\mathrm{d}t$$

$$= -2\lambda \int_\mathbb{R} \alpha_u x \mathcal{L}[\widehat{q}_v \cdot \ell_{1,x}](s)\mathcal{L}[\ell_{2,x}](s) + \lambda x \mathcal{L}[\widehat{q}_v \cdot (\widehat{q}_v * \ell_{1,x})](s)\mathcal{L}[\ell_{2,x}](s)\mu(\mathrm{d}x)$$

$$- 2\lambda \int_\mathbb{R} \alpha_v x \mathcal{L}[\widehat{q}_u \cdot \ell_{1,x}](s)\mathcal{L}[\ell_{2,x}](s) + \lambda x \mathcal{L}[\widehat{q}_u \cdot (\widehat{q}_u * \ell_{1,x})](s)\mathcal{L}[\ell_{2,x}](s)\mu(\mathrm{d}x)$$

$$\stackrel{(a)}{=} -2\lambda \int_\mathbb{R} \alpha_u x \mathcal{L}[(\widehat{q}_v \cdot \ell_{1,x}) * \ell_{2,x}](s) + \lambda x \mathcal{L}\left\{[\widehat{q}_v \cdot (\widehat{q}_v * \ell_{1,x})] * \ell_{2,x}\right\}(s)\mu(\mathrm{d}x)$$

$$- 2\lambda \int_\mathbb{R} \alpha_v x \mathcal{L}[(\widehat{q}_u \cdot \ell_{1,x}) * \ell_{2,x}](s) + \lambda x \mathcal{L}\left\{[\widehat{q}_u \cdot (\widehat{q}_u * \ell_{1,x})] * \ell_{2,x}\right\}(s)\mu(\mathrm{d}x), \tag{95}$$

where in step $(a)$ we apply the convolution property. The contour integral for $\Pi_4$ can be evaluated using a similar method to that for $\Pi_3$, utilizing the Laplace transforms of $\widehat{\Xi}_u$ and $\widehat{\Xi}_v$. We provide the result directly as follows

$$\frac{1}{2\pi\jmath} \oint_\Gamma \Pi_4 \mathrm{d}z = -\lambda \int_\mathbb{R} \alpha_u x \mathcal{L}[(\widehat{q}_v \cdot \ell_{3,x}) * \ell_{4,x}](s) + \lambda x \mathcal{L}\{[\widehat{q}_v \cdot (\widehat{q}_v * \ell_{3,x})] * \ell_{4,x}\}(s)\mu(\mathrm{d}x)$$

$$- \lambda \int_\mathbb{R} \alpha_v x \mathcal{L}[(\widehat{q}_u \cdot \ell_{3,x}) * \ell_{4,x}](s) + \lambda x \mathcal{L}\{[\widehat{q}_u \cdot (\widehat{q}_u * \ell_{3,x})] * \ell_{4,x}\}(s)\mu(\mathrm{d}x). \tag{96}$$

Since $\frac{1}{2\pi\jmath} \oint_\Gamma \Pi_{u,v}(t, z)\mathrm{d}z = -g_{u,v}(t)$, we have $\frac{1}{2\pi\jmath} \oint_\Gamma \mathcal{L}[\widehat{\Pi}_{u,v}](s, z) = -\mathcal{L}[\widehat{g}_{u,v}](s)$. By taking the inverse Laplace transform and using (90), (95), and (96), (11) is obtained.

Finally, we establish the relation between $\widehat{g}_{u,v}$ and $e^{F_{u,v,\lambda}}$. Choose $\Gamma$ that encloses $\mathcal{M}$ and let $|s|$ be large enough such that $s^2$, $s^2/4$ lies outside $\Gamma$. Using (79e), we obtain

$$-s\mathcal{L}[e^{2F_{u,v,\lambda}}] = \frac{1}{2\pi\jmath} \oint_\Gamma m(z)\mathrm{d}z - 2\mathcal{L}[\widehat{g}_{u,v}] + \frac{1}{2\pi\jmath} \oint_\lambda 2\lambda\mathcal{L}[\widehat{q}_v\widehat{G}_u](s, z)\mathrm{d}z, \tag{97}$$

which yields

$$e^{2F_{u,v}(t)} = 1 + 2\int_0^t \widehat{g}_{u,v}(a)\mathrm{d}a$$

$$+ \int_0^t \mathrm{d}a \int_\mathbb{R} \mu(\mathrm{d}x)\left\{2\alpha_u\lambda\widehat{q}_v(a)\ell_{1,x}(a) + 2\lambda^2\widehat{q}_v(a)(\widehat{q}_v * \ell_{1,x})(a)\right\}. \tag{98}$$

Thus, we have derived the closed-form expressions for $e^{F_{u,v,\lambda}}$ and $g_{u,v}$. Substituting these results into (79e)-(79g), the closed-form expressions for $(\widehat{\Upsilon}_u, \widehat{\Upsilon}_v, \widehat{\Pi}_{u,v})$ can be obtained. By verifying that these solutions satisfy (76a)-(76g), we conclude that the system of equations (76) has been solved.

### D.4 ASYMPTOTIC EQUIVALENCE OF THE SOLUTIONS

Let $\{S^{(1)}, S \in \mathcal{S}_{ode}\}$ denote the solution to (45) where the initial conditions $B_S$ and functions $C_u$, $C_v$, $D_{u,v}$ are given in (43) with $(\boldsymbol{u}_t, \boldsymbol{v}_t) = (\boldsymbol{u}_0, \boldsymbol{v}_0)$. Denote the underlying measures for $S^{(1)}$ as $\mu_S^{(1)}$, for every $S \in \mathcal{S}_{ode}$. Similarly, define $\{S^{(2)}, S \in \mathcal{S}_{ode}\}$ as the solution to (76) with $S^{(2)} \in \mathbb{S}_{\mathcal{K}}$ for all $S \in \mathcal{S}_{ode}$ with underlying measures $\mu_S^{(2)}$. Let $\Gamma$ be a positive contour enclosing and all eigenvalues $\boldsymbol{Z}\boldsymbol{Z}^\top$ and $\mathcal{M}$. According to (Bai & Silverstein, 1998, Theorem 1.1), the largest and smallest non-zero eigenvalues of $\boldsymbol{Z}\boldsymbol{Z}^\top$ converge almost surely to $E_+$ and $E_-$, respectively. There-fore, the deterministic contour $\Gamma$ is well-defined with probability one. We note the identification $q_h^{(1)}(t) = q_h(t)$, $q_h^{(2)}(t) = \widetilde{q}_h(t)$ for $h \in \{u, v\}$. Define

$$M(t) = \sup_{z \in \Gamma} \max_{S \in \mathcal{S}_{ode}} \left\{ \left| S^{(1)}(t, z) - S^{(2)}(t, z) \right| \right\}, \tag{99}$$

$$L = \sup_{z \in \Gamma} \max \left\{ \left| B_{G_u}(z) - \alpha_u m(z) \right|, \left| B_{G_v}(z) - \alpha_v \underline{m}(z) \right|, \left| B_{\Upsilon_u}(z) - m(z) \right|, \right.$$

$$\left| B_{\Upsilon_v}(z) - \underline{m}(z) \right|, \left| B_{\Xi_u}(z) \right|, \left| B_{\Xi_v}(z) \right|, \left| B_{\Pi_{u,v}}(z) \right|,$$

$$\left. \left| C_u(z) - m(z) \right|, \left| C_v(z) - \underline{m}(z) \right|, \left| D_{u,v}(z) \right| \right\}. \tag{100}$$

In what follows, we show that $M(t) \to 0$ almost surely. By definition, we have $|q_v^{(1)}(t) - q_v^{(2)}(t)|$, $|q_u^{(1)}(t) - q_u^{(2)}(t)|$, $|g_{u,v}^{(1)}(t) - g_{u,v}^{(2)}(t)| \leqslant |\Gamma|/(2\pi)M(t)$, where $|\Gamma|$ denotes the length of $\Gamma$. Further-more,

$$\left| f_{u,v,\lambda}^{(1)} - f_{u,v,\lambda}^{(2)} \right| \leqslant (1 + \lambda |q_v^{(1)}| + \lambda |q_u^{(2)}|) \frac{|\Gamma| M(t)}{2\pi} \overset{(a)}{\leqslant} (1 + 2\lambda) \frac{|\Gamma|}{2\pi} M(t), \tag{101}$$

where the variables $t$ are omitted. In step $(a)$, the bound $|q_v^{(1)}| \leqslant 1$ follows from the existence and uniqueness of the solution to (42) and its agreement to the integral representation in Theorem 3, as discussed in Section D.2. The inequality $|q_u^{(2)}| \leqslant 1$ can be derived similarly. Now, by considering the difference of the equations related to $\Xi_u$, we obtain

$$\left| \Xi_u^{(1)}(t, z) - \Xi_u^{(2)}(t, z) \right| \leqslant |B_{\Xi_u}(z)| + \int_0^t \mathrm{d}a |q_v^{(1)}(a) - q_v^{(2)}(a)| + |z| \left| G_v^{(1)}(a, z) - G_v^{(2)}(a, z) \right|$$

$$+ \lambda \left| q_v^{(1)}(a) \right| |D_{u,v}(z)| + \left| f_{u,v,\lambda}^{(1)}(a) \right| \left| \Xi_u^{(1)}(a, z) - \Xi_u^{(2)}(a, z) \right|$$

$$+ \left| f_{u,v,\lambda}^{(1)}(a) - f_{u,v,\lambda}^{(2)}(a) \right| \left| \Xi_u^{(2)}(a, z) \right|. \tag{102}$$

Due to the continuity of the good transition kernels and the finite total variation of the measures, for any fixed $T > 0$, there exists a constant $U$ such that $\sup_{0 \leqslant t \leqslant T} |f_{u,v,\lambda}(t)| \leqslant U$. Furthermore, for every $j \in \{1, 2\}$, $S \in \mathcal{S}_{ode}$ and $z \in \Gamma$, we have

$$\sup_{0 \leqslant t \leqslant T} |S^{(j)}(t, z)| \leqslant \sup_{0 \leqslant t \leqslant T} \int_{\mathbb{R}} \frac{|\mu_S^{(j)}|(t, \mathrm{d}x)}{|x - z|} \leqslant \sup_{0 \leqslant t \leqslant T} d^{-1} \left\| \mu_S^{(j)}(t, \cdot) \right\|_{TV} \leqslant U, \tag{103}$$

almost surely, where $d$ denotes the distance between the (common) support of the measures and $\Gamma$. Taking the supremum over $z$ on both sides of (102), we get, for every $t \in [0, T]$,

$$\sup_{z \in \Gamma} \left| \Xi_u^{(1)}(t, z) - \Xi_u^{(2)}(t, z) \right| \leqslant L + \int_0^t \left( \frac{|\Gamma|}{2\pi} + \sup_{z \in \Gamma} |z| + \frac{(1 + 2\lambda)U|\Gamma|}{2\pi} + U \right) M(a) + \lambda L \mathrm{d}a$$

$$\leqslant C_1 + C_2 \int_0^t M(a) \mathrm{d}a, \tag{104}$$

almost surely, where $C_1 = (1 + \lambda T)L$ and $C_2 = \frac{|\Gamma|}{2\pi} + \sup_{z \in \Gamma} |z| + \frac{(1+2\lambda)U|\Gamma|}{2\pi} + U$. By Lemma 4, it follows that $C_1 \to 0$ almost surely as $p, n \to \infty$. In contrast, the constant $C_2$ depends only on the fixed deterministic contour $\Gamma$ and the bound $U$, and thus remains upper bounded. This argument extends to all other components $S^{(1)}, S^{(2)} \in \mathcal{S}_{ode}$, leading to the composite bound

$$M(t) \leqslant \widetilde{C}_1 + \widetilde{C}_2 \int_0^t M(a) \mathrm{d}a, \tag{105}$$

almost surely. where $\widetilde{C}_1$ are upper bounded by polynomials in $L$ and $\widetilde{C}_2$ is a bounded constant. According to Grönwall's lemma (Lemma 1), we obtain

$$\sup_{0 \leqslant t \leqslant T} M(t) \leqslant \widetilde{C}_1 \exp(\widetilde{C}_2 T). \tag{106}$$

Again by Lemma 4, we have $\sup_{0 \leqslant t \leqslant T} M(t) \to 0$ almost surely, as $p, n \to \infty$. This implies, in particular, that for every $h \in \{u, v\}$,

$$\sup_{0 \leqslant t \leqslant T} |q_h^{(1)}(t) - q_h^{(2)}(t)| \leqslant \sup_{0 \leqslant t \leqslant T} M(t) \to 0, \tag{107}$$

almost surely. This completes the proof. $\qquad\square$

## E  PROOF OF THE BOUND (18)

In this section, we analyze the gap between GD and gradient flow. Define

$$\Delta_t = \max \left\{ \left\| \boldsymbol{u}_t - \widetilde{\boldsymbol{u}}_{\lfloor \frac{t}{\eta} \rfloor} \right\|_2, \left\| \boldsymbol{v}_t - \widetilde{\boldsymbol{v}}_{\lfloor \frac{t}{\eta} \rfloor} \right\|_2 \right\}. \tag{108}$$

Using the fact that $\widetilde{\boldsymbol{u}}_k$ is orthogonal to $(\boldsymbol{I}_p - \widetilde{\boldsymbol{u}}_k \widetilde{\boldsymbol{u}}_k^\top) \boldsymbol{X} \widetilde{\boldsymbol{v}}_k$ for any $k$, we have

$$\left\| \widetilde{\boldsymbol{u}}_k + \eta (\boldsymbol{I}_p - \widetilde{\boldsymbol{u}}_k \widetilde{\boldsymbol{u}}_k^\top) \boldsymbol{X} \widetilde{\boldsymbol{v}}_k \right\|_2^{-1} = \left( 1 + \eta^2 \left\| (\boldsymbol{I}_p - \widetilde{\boldsymbol{u}}_k \widetilde{\boldsymbol{u}}_k^\top) \boldsymbol{X} \widetilde{\boldsymbol{v}}_k \right\|_2^2 \right)^{-\frac{1}{2}}$$

$$= 1 - G_{\eta,k}^u \eta^2 \left\| (\boldsymbol{I}_p - \widetilde{\boldsymbol{u}}_k \widetilde{\boldsymbol{u}}_k^\top) \boldsymbol{X} \widetilde{\boldsymbol{v}}_k \right\|_2^2, \tag{109}$$

where $G_{\eta,k}^u$ is a uniformly bounded term. In particular, we have $|G_{\eta,k}^u| \leqslant \sup_{x>0} \{ \frac{\sqrt{1+x}-1}{x\sqrt{1+x}} \}$. By (109), we can approximate $\widetilde{\boldsymbol{u}}_{k+1}$ by

$$\widetilde{\boldsymbol{u}}_{k+1} = \frac{\widetilde{\boldsymbol{u}}_k + \eta(\boldsymbol{I}_p - \widetilde{\boldsymbol{u}}_k \widetilde{\boldsymbol{u}}_k^\top) \boldsymbol{X} \widetilde{\boldsymbol{v}}_k}{\left\| \widetilde{\boldsymbol{u}}_k + \eta(\boldsymbol{I}_p - \widetilde{\boldsymbol{u}}_k \widetilde{\boldsymbol{u}}_k^\top) \boldsymbol{X} \widetilde{\boldsymbol{v}}_k \right\|_2} = \widetilde{\boldsymbol{u}}_k + \eta(\boldsymbol{I}_p - \widetilde{\boldsymbol{u}}_k \widetilde{\boldsymbol{u}}_k^\top) \boldsymbol{X} \widetilde{\boldsymbol{v}}_k + \eta^2 \boldsymbol{\delta}_{\eta,k}^u, \tag{110}$$

where the norm of the vector $\boldsymbol{\delta}_{\eta,k}^u$ is uniformly bounded in $k$, i.e., $\sup_{k \geqslant 1} \|\boldsymbol{\delta}_{\eta,k}^u\|_2 \leqslant C \|\boldsymbol{X}\|_2^2 (1 + \eta \|\boldsymbol{X}\|_2)$ for some constant $C > 0$. A similar argument on $\widetilde{\boldsymbol{v}}_k$ yields

$$\widetilde{\boldsymbol{v}}_{k+1} = \widetilde{\boldsymbol{v}}_k + \eta(\boldsymbol{I}_n - \widetilde{\boldsymbol{v}}_k \widetilde{\boldsymbol{v}}_k^\top) \boldsymbol{X}^\top \widetilde{\boldsymbol{u}}_k + \eta^2 \boldsymbol{\delta}_{\eta,k}^v, \tag{111}$$

where the norm $\boldsymbol{\delta}_{\eta,k}^v$ is also bounded by $C \|\boldsymbol{X}\|_2^2 (1 + \eta \|\boldsymbol{X}\|_2)$ uniformly in $k$. Therefore, by the gradient flow (42) and identity (110), we have

$$\boldsymbol{u}_{(k+1)\eta} - \widetilde{\boldsymbol{u}}_{k+1} = \boldsymbol{u}_{k\eta} - \widetilde{\boldsymbol{u}}_k + \eta^2 \boldsymbol{\delta}_{\eta,k}^u$$

$$+ \int_{k\eta}^{(k+1)\eta} \left[ \left( \boldsymbol{I}_p - \boldsymbol{u}_\alpha \boldsymbol{u}_\alpha^\top \right) \boldsymbol{X} \boldsymbol{v}_\alpha - (\boldsymbol{I}_p - \widetilde{\boldsymbol{u}}_k \widetilde{\boldsymbol{u}}_k^\top) \boldsymbol{X} \widetilde{\boldsymbol{v}}_k \right] \mathrm{d}\alpha.$$

As a result, by the inequality

$$\left\| (\boldsymbol{I}_p - \boldsymbol{u}_1 \boldsymbol{u}_1^\top) \boldsymbol{X} \boldsymbol{v}_1 - (\boldsymbol{I}_p - \boldsymbol{u}_2 \boldsymbol{u}_2^\top) \boldsymbol{X} \boldsymbol{v}_2 \right\|_2 \leqslant \|\boldsymbol{X}(\boldsymbol{v}_1 - \boldsymbol{v}_2)\|_2$$

$$+ \left\| (\boldsymbol{u}_1 - \boldsymbol{u}_2) \boldsymbol{u}_1^\top \boldsymbol{X} \boldsymbol{v}_1 \right\|_2 + \left\| \boldsymbol{u}_2 \left( \boldsymbol{u}_1^\top - \boldsymbol{u}_2^\top \right) \boldsymbol{X} \boldsymbol{v}_1 \right\|_2$$

$$+ \left\| \boldsymbol{u}_2 \boldsymbol{u}_2^\top \boldsymbol{X} (\boldsymbol{v}_1 - \boldsymbol{v}_2) \right\|_2 \leqslant 2 \|\boldsymbol{X}\|_2 \left[ \|\boldsymbol{u}_1 - \boldsymbol{u}_2\|_2 + \|\boldsymbol{v}_1 - \boldsymbol{v}_2\|_2 \right], \tag{112}$$

for any unit norm vectors $\boldsymbol{u}_i$ and $\boldsymbol{v}_i$, $i \in \{1, 2\}$, we have

$$\left\| \boldsymbol{u}_{(k+1)\eta} - \widetilde{\boldsymbol{u}}_{k+1} \right\|_2 \leqslant \|\boldsymbol{u}_{k\eta} - \widetilde{\boldsymbol{u}}_k\|_2 + \eta^2 \left\| \boldsymbol{\delta}_{\eta,k}^u \right\|_2$$

$$+ 2 \|\boldsymbol{X}\|_2 \int_{k\eta}^{(k+1)\eta} \|\boldsymbol{u}_\alpha - \widetilde{\boldsymbol{u}}_k\|_2 + \|\boldsymbol{v}_\alpha - \widetilde{\boldsymbol{v}}_k\|_2 \, \mathrm{d}\alpha$$

$$\leqslant \Delta_{k\eta} + C \|\boldsymbol{X}\|_2^2 (1 + \eta \|\boldsymbol{X}\|_2) \eta^2 + 4 \|\boldsymbol{X}\|_2 \int_{k\eta}^{(k+1)\eta} \Delta_\alpha \mathrm{d}\alpha. \tag{113}$$

The same argument on $\|\boldsymbol{v}_{(k+1)\eta} - \widetilde{\boldsymbol{v}}_{k+1}\|_2$ together with (113) yields the inequality $\Delta_{(k+1)\eta} \leqslant \Delta_{k\eta} + C \|\boldsymbol{X}\|_2^2 (1 + \eta \|\boldsymbol{X}\|_2)\eta^2 + 4 \|\boldsymbol{X}\|_2 \int_{k\eta}^{(k+1)\eta} \Delta_\alpha \mathrm{d}\alpha$, which immediately gives

$$\Delta_{k\eta} \leqslant C \|\boldsymbol{X}\|_2^2 (1 + \eta \|\boldsymbol{X}\|_2)k\eta^2 + 4 \|\boldsymbol{X}\|_2 \int_0^{k\eta} \Delta_\alpha \mathrm{d}\alpha, \tag{114}$$

for any non negative integers $k$. Then, for any $t > 0$ and $k = \lfloor \frac{t}{\eta} \rfloor$, we have $\boldsymbol{u}_t - \widetilde{\boldsymbol{u}}_k = \boldsymbol{u}_t - \boldsymbol{u}_{k\eta} + \boldsymbol{u}_{k\eta} - \widetilde{\boldsymbol{u}}_k$ and

$$\begin{aligned}
\|\boldsymbol{u}_t - \widetilde{\boldsymbol{u}}_k\|_2 &\leqslant \Delta_{k\eta} + \int_{k\eta}^t \left\|\left(\boldsymbol{I}_p - \boldsymbol{u}_\alpha \boldsymbol{u}_\alpha^\top\right) \boldsymbol{X} \boldsymbol{v}_\alpha - (\boldsymbol{I}_p - \widetilde{\boldsymbol{u}}_k \widetilde{\boldsymbol{u}}_k^\top)\boldsymbol{X}\widetilde{\boldsymbol{v}}_k\right\|_2 \mathrm{d}\alpha \\
&\quad + (t - k\eta) \left\|(\boldsymbol{I}_p - \widetilde{\boldsymbol{u}}_k \widetilde{\boldsymbol{u}}_k^\top)\boldsymbol{X}\widetilde{\boldsymbol{v}}_k\right\|_2 \\
&\leqslant 4 \|\boldsymbol{X}\|_2 \int_0^t \Delta_\alpha \mathrm{d}\alpha + C \|\boldsymbol{X}\|_2^2 (1 + \eta \|\boldsymbol{X}\|_2)k\eta^2 + \eta \|\boldsymbol{X}\|_2.
\end{aligned} \tag{115}$$

Using the same method as in (115) to bound the term $\|\boldsymbol{v}_t - \widetilde{\boldsymbol{v}}_k\|_2$, we can obtain

$$\Delta_t \leqslant 4 \|\boldsymbol{X}\|_2 \int_0^t \Delta_\alpha \mathrm{d}\alpha + C \|\boldsymbol{X}\|_2^2 (1 + \eta \|\boldsymbol{X}\|_2)k\eta^2 + \eta \|\boldsymbol{X}\|_2. \tag{116}$$

By Grönwall's Lemma, we have

$$\sup_{0 \leqslant t \leqslant T} \Delta_t \leqslant \left[C \|\boldsymbol{X}\|_2^2 (1 + \eta \|\boldsymbol{X}\|_2)T\eta + \eta \|\boldsymbol{X}\|_2\right] \exp\left(4 \|\boldsymbol{X}\|_2 T\right). \tag{117}$$

Therefore, we have derived the bound for $\Delta_t$ with given $T$ and $\eta > 0$.

Next, we study the asymptotic case. In fact, by (Yin et al., 1988, Lemma 3.1), we have $\lim_{p,n \to \infty} \|\boldsymbol{Z}\|_2 = (1 + \sqrt{c})^2$ almost surely, which implies $\|\boldsymbol{X}\|_2 \leqslant \lambda\|\boldsymbol{u}^*(\boldsymbol{v}^*)^\top\|_2 + \|\boldsymbol{Z}\|_2 \leqslant \lambda + (1 + \sqrt{c})^2$ almost surely, as $p, n \to \infty$. Therefore, we have $\mathbb{P}(\limsup_{p,n} \sup_{0 \leqslant t \leqslant T} \Delta_t \leqslant C(\eta + \eta^2)) = 1$ for a constant $C > 0$ which only depends on $T, \lambda$, and $c$. Therefore, we have completed the proof of (18). $\qquad\square$

## F  PROOF OF THEOREM 2

### F.1  CASE I: $\lambda > c^{1/4}$

We begin by analyzing the asymptotic order of $\widehat{q}_u(t)$ as $t \to \infty$. We express (8) in the form $\widehat{q}_u(t) = Z_1(t) + Z_2(t) + Z_3(t) + Z_4(t)$ and evaluate each term separately. Through direct calculation, the following is obtained

$$\begin{aligned}
\ell_{3,x}(t) * \ell_{3,\vartheta_\lambda}(t) = &-\frac{e^{-\sqrt{\vartheta_\lambda}t}}{2\sqrt{\vartheta_\lambda}(\vartheta_\lambda - x)} + \frac{e^{\sqrt{\vartheta_\lambda}t}}{2\sqrt{\vartheta_\lambda}(\vartheta_\lambda - x)} \\
&-\frac{e^{-\sqrt{x}t}}{2\sqrt{x}(x - \vartheta_\lambda)} + \frac{e^{\sqrt{x}t}}{2\sqrt{x}(x - \vartheta_\lambda)},
\end{aligned} \tag{118}$$

Since $\lambda^2 > \sqrt{c}$, it follows that $\vartheta_\lambda > (1 + \sqrt{c})^2 = E_+$. Consequently, we have

$$\begin{aligned}
Z_1(t) &= -\frac{\alpha_v}{\lambda} \frac{e^{\sqrt{\vartheta_\lambda}t}}{2\sqrt{\vartheta_\lambda}} \int_{\mathbb{R}} \frac{x}{\vartheta_\lambda - x}\mu(\mathrm{d}x) - \frac{\alpha_v}{2\lambda} \int_{\mathbb{R}} \frac{\sqrt{x}e^{\sqrt{x}t}}{x - \vartheta_\lambda}\mu(\mathrm{d}x) + o(1) \\
&\overset{(a)}{=} -\frac{e^{\sqrt{\vartheta_\lambda}t}\alpha_v c}{2\sqrt{\vartheta_\lambda}\lambda^3} - \frac{\alpha_v}{2\lambda} \int_{E_-}^{E_+} \frac{\sqrt{x}e^{\sqrt{x}t}}{x - \vartheta_\lambda} \frac{\sqrt{(x - E_-)(E_+ - x)}}{2\pi x}\mathrm{d}x + o(1) \\
&\overset{(b)}{=} -\frac{e^{\sqrt{\vartheta_\lambda}t}\alpha_v c}{2\sqrt{\vartheta_\lambda}\lambda^3} + \frac{\alpha_v\sqrt{E_+ - E_-}(E^+)^{1/4}}{2\lambda\sqrt{2\pi}(\vartheta_\lambda - E^+)} \frac{e^{\sqrt{E_+}t}}{t^{3/2}} + o(e^{\sqrt{E_+}t}t^{-3/2}),
\end{aligned} \tag{119}$$

where step $(a)$ uses properties of the Stieltjes transform of the MP law (Couillet & Liao, 2022)

$$\vartheta_\lambda m(\vartheta_\lambda) = -1 - \frac{1}{\max(c\lambda^{-2}, \lambda^2)}, \tag{120}$$

and step $(b)$ follows from Lemma 3. For the term $Z_2(t)$, we have

$$Z_2(t) = \alpha_v \lambda \frac{e^{\sqrt{\vartheta_\lambda}t} - e^{-\sqrt{\vartheta_\lambda}t}}{2\sqrt{\vartheta_\lambda}} = \frac{\alpha_v \lambda e^{\sqrt{\vartheta_\lambda}t}}{2\sqrt{\vartheta_\lambda}} + o(1). \tag{121}$$

Similar to the evaluation of $Z_1(t)$, and using the identity

$$\ell_{1,\vartheta_\lambda}(t) * \ell_{3,x}(t) = \frac{e^{-\sqrt{\vartheta_\lambda}t}}{2(\vartheta_\lambda - x)} + \frac{e^{\sqrt{\vartheta_\lambda}t}}{2(\vartheta_\lambda - x)} - \frac{e^{-\sqrt{x}t}}{2(\vartheta_\lambda - x)} - \frac{e^{\sqrt{x}t}}{2(\vartheta_\lambda - x)}, \tag{122}$$

we can get

$$Z_3(t) = -\frac{\alpha_u c(1 + \lambda^2)}{2\lambda^2}\left[e^{\sqrt{\vartheta_\lambda}t}\int_{\mathbb{R}}\frac{\mu(\mathrm{d}x)}{\vartheta_\lambda - x} - \int_{E_-}^{E_+}\frac{e^{\sqrt{x}t}\sqrt{(E_+ - x)(x - E_-)}}{(\vartheta_\lambda - x)2\pi c x}\mathrm{d}x\right] + O(1)$$

$$= -\frac{\alpha_u c(\lambda^2 + 1)e^{\sqrt{\vartheta_\lambda}t}}{2(\lambda^2 + c)\lambda^2} + \frac{\alpha_u(1 + \lambda^2)\sqrt{E_+ - E_-}}{2\lambda^2(\vartheta_\lambda - E_+)\sqrt{2\pi}(E^+)^{1/4}}\frac{e^{\sqrt{E_+}t}}{t^{3/2}} + o(e^{\sqrt{E_+}t}t^{-\frac{3}{2}}). \tag{123}$$

By definition, $Z_4(t) = \alpha_u e^{\sqrt{\vartheta_\lambda}t}/2 + o(1)$. Combining these results, the following estimation is obtained

$$\widehat{q}_u(t) = \frac{(\lambda^4 - c)}{2\lambda^2\sqrt{\lambda^2 + c}}\left[\frac{\alpha_v}{\sqrt{1 + \lambda^2}} + \frac{\alpha_u}{\sqrt{\lambda^2 + c}}\right]e^{\sqrt{\vartheta_\lambda}t}$$

$$+ \frac{\sqrt{E_+ - E_-}}{2\sqrt{2\pi}(\vartheta_\lambda - E_+)}\left[\frac{\alpha_v(E^+)^{1/4}}{\lambda} + \frac{\alpha_u(1 + \lambda^2)}{\lambda^2(E^+)^{1/4}}\right]\frac{e^{\sqrt{E_+}t}}{t^{3/2}} + o(e^{\sqrt{E^+}t^{-3/2}}t^{-3/2})$$

$$= A_u e^{\sqrt{\vartheta_\lambda}t} + B_u e^{\sqrt{E^+}t}t^{-3/2} + o(e^{\sqrt{E^+}t}t^{-3/2}). \tag{124}$$

Similar to the evaluation for $\widehat{q}_u(t)$, we can get

$$\widehat{q}_v(t) = A_v e^{\sqrt{\vartheta_\lambda}t} + B_v e^{\sqrt{E^+}t}t^{-3/2} + o(e^{\sqrt{E^+}t}t^{-3/2}), \tag{125}$$

where

$$A_v = \frac{(\lambda^4 - c)}{2\lambda^2\sqrt{\lambda^2 + 1}}\left[\frac{\alpha_v}{\sqrt{1 + \lambda^2}} + \frac{\alpha_u}{\sqrt{\lambda^2 + c}}\right], \tag{126}$$

$$B_v = \frac{\sqrt{E_+ - E_-}}{2\sqrt{2\pi}(\vartheta_\lambda - E_+)}\left[\frac{\alpha_u(E_+)^{1/4}}{\lambda} + \frac{\alpha_v(c + \lambda^2)}{c\lambda^2(E^+)^{1/4}}\right]. \tag{127}$$

Next, we will analyze the order of the denominator $\widehat{p}(t)$ for the following two cases.

### F.1.1 $A_u \neq 0$

Under this condition, the solutions have the asymptotic forms $\widehat{q}_u(t) = A_u e^{\sqrt{\vartheta_\lambda}t} + o(e^{\sqrt{\vartheta_\lambda}t})$ and $\widehat{q}_v(t) = A_v e^{\sqrt{\vartheta_\lambda}t} + o(e^{\sqrt{\vartheta_\lambda}t})$. To determine the asymptotic order of $\widehat{p}(t)$, we first evaluate $\widehat{g}_{u,v}(t)$. Writing (11) as $\widehat{g}_{u,v}(t) = \sum_{j=1}^{5} G_j(t)$, we focus on the evaluations for $G_1$ and $G_2$ as the remaining terms follow similar estimations.

Given $\vartheta_\lambda > E_+$, we observe $e^{-2\sqrt{\vartheta_\lambda}t}G_1(t) \to 0$ as $t \to \infty$. Therefore, the term $G_1(t)$ does not contribute to the leading order asymptotics. We then analyze $e^{-2\sqrt{\vartheta_\lambda}t}G_2(t)$. By definition, we have

$$e^{-2\sqrt{\vartheta_\lambda}t}G_2(t) = 2\lambda\alpha_u\int_{\mathbb{R}}e^{-2\sqrt{\vartheta_\lambda}t}x(\widehat{q}_v \cdot \ell_{1,x}) * \ell_{2,x}\mu(\mathrm{d}x)$$

$$+ 2\lambda^2\int_{\mathbb{R}}e^{-2\sqrt{\vartheta_\lambda}t}x[\widehat{q}_v \cdot (\widehat{q}_v * \ell_{1,x})] * \ell_{2,x}\mu(\mathrm{d}x) = G_{2,1}(t) + G_{2,2}(t). \tag{128}$$

To analyze the limits, we apply the Final Value Theorem (Lemma 2). Since $e^{-\sqrt{\vartheta_\lambda}t}\ell_{1,x}(t) \to 0$ and $\mathcal{L}[\ell_{2,x}](2\sqrt{\vartheta_\lambda}) = 1/4(\vartheta_\lambda - x)$, Lemma 2 implies $G_{2,1}(t) \to 0$ as $t \to \infty$. Applying Lemma 2 again yields

$$\lim_{t\to\infty}e^{-\sqrt{\vartheta_\lambda}t}(\widehat{q}_v * \ell_{1,x})(t) = \frac{\sqrt{\vartheta_\lambda}A_v}{\vartheta_\lambda - x}, \tag{129}$$

which implies

$$\lim_{t \to \infty} e^{-2\sqrt{\vartheta_\lambda} t} \{[\widehat{q}_v \cdot (\widehat{q}_v * \ell_{1,x})] * \ell_{2,x}\}(t) = \frac{\sqrt{\vartheta_\lambda} A_v^2}{4(\vartheta_\lambda - x)^2}. \tag{130}$$

By the dominated convergence theorem and the properties of the Stieltjes transform, we have

$$\lim_{t \to \infty} G_{2,2}(t) = 2\lambda^2 \cdot \frac{\sqrt{\vartheta_\lambda} A_v^2}{4} \int_{\mathbb{R}} \frac{x}{(\vartheta_\lambda - x)^2} \mu(\mathrm{d}x) = \frac{\lambda^2 \sqrt{\vartheta_\lambda} A_v^2}{2(\lambda^4 - c)}. \tag{131}$$

The asymptotic orders of $G_3$, $G_4$, and $G_5$ can be evaluated similarly using Lemma 2. Gathering all the results gives

$$\lim_{t \to \infty} e^{-2\sqrt{\vartheta_\lambda} t} \widehat{g}_{u,v}(t) = \frac{\lambda^2 \sqrt{\vartheta_\lambda}(A_v^2 + cA_u^2)}{\lambda^4 - c}. \tag{132}$$

As a result, we can get

$$\lim_{t \to \infty} e^{-2\sqrt{\vartheta_\lambda} t} \widehat{p}(t) = \frac{\lambda^2(A_v^2 + cA_u^2)}{\lambda^4 - c} + \frac{\lambda^2(\lambda^2 + 1)A_v^2}{\vartheta_\lambda \lambda^2}$$

$$= \frac{\lambda^4 - c}{4\lambda^2} \left[ \frac{\alpha_v}{\sqrt{1 + \lambda^2}} + \frac{\alpha_u}{\sqrt{\lambda^2 + c}} \right]^2, \tag{133}$$

by the observation $\int_0^t f(a)\mathrm{d}a = (f * 1)(t)$. Therefore, provided that $A_u \neq 0$, the limit values of $\widetilde{q}_u(t)$ and $\widetilde{q}_v(t)$ are given by $\lim_{t \to \infty} \widetilde{q}_u(t) = \mathcal{I} \cdot \sqrt{\frac{1 - c\lambda^{-4}}{1 + c\lambda^{-2}}}$ and $\lim_{t \to \infty} \widetilde{q}_v(t) = \mathcal{I} \cdot \sqrt{\frac{1 - c\lambda^{-4}}{1 + \lambda^{-2}}}$, respectively.

### F.1.2 $A_u = 0$

According to Lemma 3, we have

$$G_1(t) = \frac{1 + c^{-1}}{8\pi} \int_{E_-}^{E_+} \frac{e^{2\sqrt{x}t}\sqrt{(E_+ - x)(x - E_-)}}{\sqrt{x}} \mathrm{d}x$$

$$= \frac{(1 + c^{-1})\sqrt{E_+ - E_-}(E_+)^{1/4}}{16\sqrt{\pi}} \frac{e^{2\sqrt{E_+}t}}{t^{3/2}} + o(e^{2\sqrt{E_+}t} t^{-3/2}). \tag{134}$$

Thus, there holds

$$\widehat{p}(t) \geqslant 2 \int_0^t G_1(a)\mathrm{d}a \geqslant K \frac{e^{2\sqrt{E_+}t}}{t^{3/2}}, \tag{135}$$

for $t$ sufficiently large and $K > 0$ is a constant. Therefore, we have

$$\widetilde{q}_u(t) \leqslant \frac{(B_u + o(1))e^{\sqrt{E_+}t} t^{-3/2}}{\sqrt{K} e^{\sqrt{E_+}t} t^{-3/4}}, \tag{136}$$

which implies $\lim_{t \to \infty} \widetilde{q}_u(t) = 0$. The relation $\lim_{t \to \infty} \widetilde{q}_v(t) = 0$ can be obtained similarly.

### F.2 Case II: $\lambda < c^{1/4}$

By applying the same method used in (124) and (125), we obtain $\widehat{q}_h(t) = B_h e^{\sqrt{E^+}t} t^{-3/2} + o(e^{\sqrt{E^+}t} t^{-3/2})$, $h \in \{u, v\}$. This result aligns with the case discussed in Section F.1.2, which implies $\lim_{t \to \infty} \widetilde{q}_h(t) = 0$, $h \in \{u, v\}$.

### F.3 Case III: $\lambda = c^{1/4}$

In this case, we have $\vartheta_\lambda = E_+$, which is precisely the right endpoint of the support of the MP distribution. Fortunately, the Stieltjes transform $m(x)$ is right continuous at $x = E_+$ (Couillet & Liao, 2022). Using Lemma 3 and employing estimates similar to those in (124) and (125), we can obtain

$$\widehat{q}_h(t) = (C_h + o(1))e^{\sqrt{E_+}t} t^{-1/2}, \quad h \in \{u, v\}, \tag{137}$$

where $C_u$ and $C_v$ are constants. However, we need to make a more precise estimate for $\widehat{p}(t)$ in the denominator, as the upper bound in (136) increases to infinity. To this end, the following lemma is useful.

**Lemma 5.** *Assume $b > a > 0$ and $f \in \mathcal{C}([a,b])$ such that $f(b) \neq 0$. Then, for any sufficiently large $t$, it holds that*

$$K_L e^{2\sqrt{b}t} t^{-1/2} \leqslant \int_a^b f(x)\sqrt{b-x}e^{2\sqrt{x}t}\left[\int_0^t \frac{e^{\sqrt{b}y}}{\sqrt{y}}e^{-\sqrt{x}y}\mathrm{d}y\right]^2 \mathrm{d}x \leqslant K_U e^{2\sqrt{b}t} t^{-1/2}, \quad (138)$$

*where $K_U \geqslant K_L > 0$ are two constants.*

*Proof:* It suffices to consider $f(x) = 1$ and restrict the integration to $[b - \varepsilon, b]$ for small enough $\varepsilon$, since $e^{\sqrt{x}t}$ concentrates near $x = b$. For given $\delta > 0$, we choose $\varepsilon > 0$ such that $\sup_{u \in [0,\varepsilon]} |\delta_u| \leqslant \delta$, where $\delta_u$ is defined in (146). By changing the integration variables via $u = b - x$, $\alpha = ut/\sqrt{b}$, and $\beta = y/t$, we obtain

$$e^{-2\sqrt{b}t}t^{1/2}\int_{b-\varepsilon}^b \sqrt{b-x}e^{2\sqrt{x}t}\left[\int_0^t \frac{e^{\sqrt{b}y}}{\sqrt{y}}e^{-\sqrt{x}y}\mathrm{d}y\right]^2 \mathrm{d}x$$

$$= t^{1/2}\int_0^\varepsilon \sqrt{u}e^{-\frac{(1+\delta_u)ut}{\sqrt{b}}}\left[\int_0^t y^{-\frac{1}{2}}e^{\frac{(1+\delta_u)uy}{2\sqrt{b}}}\mathrm{d}y\right]^2 \mathrm{d}u$$

$$= b^{\frac{3}{4}}\int_0^{\frac{\varepsilon t}{\sqrt{b}}} \sqrt{\alpha}e^{-(1+\delta_{\alpha,t})\alpha}\left[\int_0^1 \beta^{-\frac{1}{2}}e^{\frac{(1+\delta_{\alpha,t})\alpha\beta}{2}}\mathrm{d}\beta\right]^2 \mathrm{d}\alpha = b^{\frac{3}{4}}I. \quad (139)$$

We next estimate the inner integral $\int_0^1 \beta^{-\frac{1}{2}}e^{\frac{(1+\delta_{\alpha,t})\alpha\beta}{2}}\mathrm{d}\beta$ and derive bounds for $I$. By splitting the integration interval at some $\tau \in (0,1)$, we have

$$\int_0^1 \beta^{-\frac{1}{2}}e^{\frac{(1+\delta_{\alpha,t})\alpha\beta}{2}}\mathrm{d}\beta = \left(\int_0^\tau + \int_\tau^1\right)\beta^{-\frac{1}{2}}e^{\frac{(1+\delta_{\alpha,t})\alpha\beta}{2}}\mathrm{d}\beta$$

$$\leqslant 2\sqrt{\tau}e^{\frac{(1+\delta_{\alpha,t})\alpha\tau}{2}} + \frac{2e^{\frac{(1+\delta_{\alpha,t})\alpha}{2}}}{\sqrt{\tau}\alpha(1+\delta_{\alpha,t})}. \quad (140)$$

Therefore, by $|\delta_{\alpha,t}| \leqslant \delta$ (147) and (140), we derive an upper bound

$$I \leqslant I_1 + 4\int_1^\infty \sqrt{\alpha}\tau e^{(1-\delta)\alpha(\tau-1)} + \frac{1}{\tau\alpha^{3/2}(1-\delta)^2} + \frac{2e^{\frac{(1-\delta)\alpha(\tau-1)}{2}}}{\alpha^{1/2}(1-\delta)}\mathrm{d}\alpha := I_U, \quad (141)$$

where $I_1$ is some absolute constant. It can be verified that $I_U < \infty$ for any $\tau \in (0,1)$.

For the lower bound, again using $|\delta_{\alpha,t}| \leqslant \delta$, we can obtain

$$I \geqslant I_1 + \int_1^{\frac{\varepsilon t}{\sqrt{b}}} \sqrt{\alpha}e^{-(1+\delta)\alpha}\left[\int_0^1 \beta^{-\frac{1}{2}}e^{\frac{(1-\delta)\alpha\beta}{2}}\mathrm{d}\beta\right]^2 \mathrm{d}\alpha$$

$$\geqslant I_1 + 4\int_1^{\frac{\varepsilon t}{\sqrt{b}}} \sqrt{\alpha}e^{-(1+\delta)\alpha}\left[\frac{e^{\frac{(1-\delta)\alpha}{2}}}{(1-\delta)\alpha}\right]^2 \mathrm{d}\alpha \geqslant I_1 + \frac{4}{(1-\delta)^2}\int_1^{\frac{\varepsilon t}{\sqrt{b}}} \frac{e^{-2\delta\alpha}}{\alpha^{3/2}}\mathrm{d}\alpha. \quad (142)$$

Taking $t \to \infty$, we have $\liminf_{t \to \infty} I \geqslant I_L$ for some constant $I_L > 0$. Therefore, we have completed the proof for Lemma 5. $\qquad\square$

To determine the asymptotic order for $\widehat{p}(t)$, we first give a lower bound for $\widehat{g}_{u,v}(t)$. In particular, we will evaluate the term $\int_{\mathbb{R}}[\widehat{q}_v \cdot (\widehat{q}_v * \ell_{1,x})] * \ell_{2,x}\mu(\mathrm{d}x)$. Noting that the orders of $\ell_{1,x}$ and $\ell_{2,x}$ are dominated by $e^{\sqrt{x}t}/2$ and $e^{2\sqrt{x}}\sqrt{x}/4$, respectively, it suffices to study the simplified integral $\int_{\mathbb{R}} \sqrt{x}[\widehat{q}_v \cdot (\widehat{q}_v * e^{\sqrt{x}t})] * e^{2\sqrt{x}t}\mu(\mathrm{d}x)$. Rewriting this expression yields

$$I_{v,v} = \int_{\mathbb{R}} \sqrt{x}[\widehat{q}_v \cdot (\widehat{q}_v * e^{\sqrt{x}t})] * e^{2\sqrt{x}t}\mu(\mathrm{d}x)$$

$$= \int_{\mathbb{R}} \sqrt{x}\int_0^t \widehat{q}_v(a)\int_0^a \widehat{q}_v(b)e^{\sqrt{x}(a-b)}\mathrm{d}b e^{2\sqrt{x}(t-a)}\mathrm{d}a$$

$$= \frac{1}{2}\int_{E_-}^{E_+} \sqrt{x}e^{2\sqrt{x}t}\left[\int_0^t \widehat{q}_v(a)e^{-\sqrt{x}a}\mathrm{d}a\right]^2 \frac{\sqrt{(x-E_-)(E_+-x)}}{2\pi cx}\mathrm{d}x$$

$$= \int_{E_-}^{E_+} f(x) e^{2\sqrt{x}t} \left[ \int_0^t \widehat{q}_v(a) e^{-\sqrt{x}a} \mathrm{d}a \right]^2 \sqrt{E_+ - x} \mathrm{d}x, \tag{143}$$

where $f(x) = \sqrt{(x - E_-)x}/(4\pi cx)$. According to (137), we know that there exists constant $A > 0$ such that $\inf_{t \geqslant A} |\widehat{q}_v(t)/(e^{\sqrt{E_+}t}t^{-1/2})| \geqslant C_v/2$. Thus, we have

$$e^{-2\sqrt{E_+}t}t^{\frac{1}{2}} I_{v,v}$$

$$\geqslant \frac{e^{-2\sqrt{E_+}t}t^{\frac{1}{2}} C_v^2}{4} \int_{E_-}^{E_+} f(x) e^{2\sqrt{x}t} \left[ \int_A^t \frac{e^{(\sqrt{E_+} - \sqrt{x})a}}{\sqrt{a}} \mathrm{d}a \right]^2 \sqrt{E_+ - x} \mathrm{d}x$$

$$\stackrel{(a)}{=} \frac{e^{-2\sqrt{E_+}t}t^{\frac{1}{2}} C_v^2}{4} \int_{E_-}^{E_+} f(x) e^{2\sqrt{x}t} \left[ \int_0^t \frac{e^{(\sqrt{E_+} - \sqrt{x})a}}{\sqrt{a}} \mathrm{d}a \right]^2 \sqrt{E_+ - x} \mathrm{d}x + o(1), \tag{144}$$

where step $(a)$ is due to $\int_0^A \frac{e^{(\sqrt{E_+} - \sqrt{x})a}}{\sqrt{a}} \mathrm{d}a \leqslant 2e^{(\sqrt{E_+} - \sqrt{x})A} \sqrt{A}$ contributes only lower order terms. By applying Lemma 5, we have $\liminf_{t \to \infty} e^{-2\sqrt{E_+}t}t^{\frac{1}{2}} I_{v,v} \geqslant K_L$ for some constant $K_L$. A symmetric argument establishes the corresponding upper bound, yielding $\limsup_{t \to \infty} e^{-2\sqrt{E_+}t}t^{\frac{1}{2}} I_{v,v} \leqslant K_U$ for some constant $K_U > 0$.

At this stage, we can give the upper bound for $\widetilde{q}_h(t)$, $h \in \{u, v\}$. Combining with (137), we obtain

$$\widetilde{q}_h(t) \leqslant \frac{\widehat{q}_h(t)}{\sqrt{\widehat{p}(t)}} \leqslant \frac{(C_h + o(1)) e^{\sqrt{E_+}t} t^{-1/2}}{\sqrt{2\lambda^2 \int_0^t I_{u,v} \mathrm{d}t}} \stackrel{(a)}{\leqslant} \frac{\widetilde{C}_h e^{\sqrt{E_+}t} t^{-1/2}}{e^{\sqrt{E_+}t} t^{-1/4}}, h \in \{u, v\}, \tag{145}$$

where $\widetilde{C}_h$ is a constant and step $(a)$ follows from the fact that the ratio $\int_0^t I_{u,v} \mathrm{d}t/(e^{2\sqrt{E_+}t}t^{-1/2})$ converges to a positive value as $t \to \infty$. Therefore, we have completed the proof for Theorem 2. $\square$

## G   PROOF OF LEMMA 3

According to Taylor's lemma, we have

$$\sqrt{b - u} = \sqrt{b} \left[ 1 - \frac{(1 + \delta_u)u}{2b} \right], \tag{146}$$

for small $u$. By the continuity of $f$, for any given $\delta > 0$, we can choose a sufficiently small $\varepsilon$ such that

$$\sup_{x,y \in [a,b], |x-y| \leqslant \varepsilon} |f(x) - f(y)| \leqslant \delta, \quad \sup_{0 \leqslant u \leqslant \varepsilon} |\delta_u| \leqslant \delta. \tag{147}$$

Define $M = \sup_{x \in [a,b]} |f(x)|$, $A(t) = \int_a^b f(x) \exp[\sqrt{x}t](b - x)^\alpha \mathrm{d}x$, and $B(t) = f(b) \exp[\sqrt{b}t] \Gamma(\alpha + 1) \left( \frac{2\sqrt{b}}{t} \right)^{\alpha+1}$. By calculating $|A(t)/B(t) - 1|$, we have

$$\left| \frac{A(t)}{B(t)} - 1 \right| = \left| \int_0^{b-a} f(b - u) \exp[\sqrt{b - u}t] u^\alpha \mathrm{d}u \middle/ B(t) - 1 \right|$$

$$\leqslant \left| \frac{f(b) \int_0^\varepsilon \exp[\sqrt{b - u}t] u^\alpha \mathrm{d}u}{B(t)} - 1 \right| + \frac{\delta \int_0^\varepsilon \exp[\sqrt{b - u}t] u^\alpha \mathrm{d}u}{B(t)}$$

$$+ \frac{M \exp[\sqrt{b - \varepsilon}t] \int_\varepsilon^{b-a} u^\alpha \mathrm{d}u}{B(t)} = X_1(t) + X_2(t) + X_3(t). \tag{148}$$

It is clear that $\lim_{t \to \infty} X_3(t) = 0$. Next, we evaluate $X_1(t)$. By plugging (146) into (148), and using $\Gamma(\alpha + 1) = \int_0^\infty \exp[-y] y^\alpha \mathrm{d}y$, we obtain

$$\frac{f(b) \exp[\sqrt{b}t] \int_0^\varepsilon \exp\left[ -\frac{(1+\delta_u)ut}{2\sqrt{b}} \right] u^\alpha \mathrm{d}u}{B(t)} \stackrel{(a)}{=} \frac{\int_0^{\frac{t\varepsilon}{2\sqrt{b}}} \exp\left[ -(1 + \delta_u)y \right] y^\alpha \mathrm{d}y}{\int_0^\infty \exp[-y] y^\alpha \mathrm{d}y}$$

$$\leqslant \frac{\int_0^\infty \exp\left[-(1-\delta)y\right] y^\alpha \mathrm{d}y}{\int_0^\infty \exp[-y]y^\alpha \mathrm{d}y} \leqslant \frac{1}{(1-\delta)^{\alpha+1}}, \tag{149}$$

where in step $(a)$ we change the variable $y = ut/(2\sqrt{b})$. Similarly, we can derive the lower bound

$$\liminf_{t\to\infty} \frac{f(b)\exp[\sqrt{b}t]\int_0^\varepsilon \exp\left[-\frac{(1+\delta_u)ut}{2\sqrt{b}}\right] u^\alpha \mathrm{d}u}{B(t)} \geqslant \frac{1}{(1+\delta)^{\alpha+1}}. \tag{150}$$

Combining this with the corresponding upper bound implies

$$\limsup_{t\to\infty} X_1(t) \leqslant \max\left\{(1-\delta)^{-\alpha-1} - 1, 1 - (1+\delta)^{-\alpha-1}\right\}. \tag{151}$$

An analogous analysis yields

$$\limsup_{t\to\infty} X_3(t) \leqslant \frac{\delta}{|f(b)|(1-\delta)^{\alpha+1}}. \tag{152}$$

Since $\delta > 0$ can be chosen arbitrarily small, it follows from (151) and (152) that

$$\lim_{t\to\infty} \left|\frac{A(t)}{B(t)} - 1\right| = 0, \tag{153}$$

which completes the proof. $\qquad\square$

## H  PROOF OF LEMMA 4

To simplify notation, let $C$ denote a constant and $C_a$ denote a constant that depends on $a$. Their values may depend on the context. By a standard truncation argument (Bai & Silverstein, 1998), it suffices to assume that the elements of the random matrix $\boldsymbol{Z}$ are bounded, i.e., $|\sqrt{n}Z_{ij}| \leqslant C$, for some absolute constant $C$, as this truncation does not alter the asymptotic locations of the eigenvalues.

We first prove that $\boldsymbol{u}^\top \mathbf{Q}(z)\boldsymbol{Z}\boldsymbol{v} - \boldsymbol{u}^\top \mathbb{E}[\mathbf{Q}(z)\boldsymbol{Z}]\boldsymbol{v}$ converges to 0 almost surely for given $z \in \mathbb{C}_+$. To this end, we control the moment by the martingale difference argument and the Burkholder inequality (Bai & Silverstein, 1998, Lemma 2.1). Write $\boldsymbol{Z} = [\boldsymbol{z}_1, \ldots, \boldsymbol{z}_n]$, $\boldsymbol{v} = [v_1, \ldots, v_n]^\top$, $\boldsymbol{Z}_j = [\boldsymbol{z}_j, \ldots, \boldsymbol{z}_{j-1}, \boldsymbol{z}_{j+1}, \ldots, \boldsymbol{z}_n]$, $\boldsymbol{v}_j = [v_1, \ldots, v_{j-1}, v_{j+1}, \ldots, v_n]^\top$, and $\mathbf{Q}_j(z) = (\boldsymbol{Z}_j\boldsymbol{Z}_j - z\boldsymbol{I}_p)^{-1}$. Define the quantities

$$\alpha_j(z) = \frac{1}{n}\operatorname{Tr}\mathbf{Q}_j(z), \quad \eta_j(z) = \frac{1}{1+\alpha_j(z)}, \quad \beta_j(z) = \frac{1}{1+\boldsymbol{z}_j^\top \mathbf{Q}_j(z)\boldsymbol{z}_j},$$

$$\Delta_j(z) = \alpha_j(z) - \boldsymbol{z}_j^\top \mathbf{Q}_j(z)\boldsymbol{z}_j. \tag{154}$$

Let $\mathbb{E}_j(\cdot) = \mathbb{E}(\cdot|\boldsymbol{z}_1, \ldots, \boldsymbol{z}_j)$ and $\mathbb{E}_0(\cdot) = \mathbb{E}(\cdot)$. By taking the difference between $\boldsymbol{u}^\top\mathbf{Q}(z)\boldsymbol{Z}\boldsymbol{v}$ and $\mathbb{E}\boldsymbol{u}^\top\mathbf{Q}(z)\boldsymbol{Z}\boldsymbol{v}$, we have

$$\boldsymbol{u}^\top\mathbf{Q}(z)\boldsymbol{Z}\boldsymbol{v} - \boldsymbol{u}^\top\mathbb{E}[\mathbf{Q}(z)\boldsymbol{Z}]\boldsymbol{v} = \sum_{j=1}^n [\mathbb{E}_j - \mathbb{E}_{j-1}]\boldsymbol{u}^\top\mathbf{Q}(z)\boldsymbol{Z}\boldsymbol{v}$$

$$\sum_{j=1}^n [\mathbb{E}_j - \mathbb{E}_{j-1}]\left(\boldsymbol{u}^\top\mathbf{Q}(z)\boldsymbol{Z}\boldsymbol{v} - \boldsymbol{u}^\top\mathbf{Q}(z)\boldsymbol{Z}_j\boldsymbol{v}_j\right) + \left(\boldsymbol{u}^\top\mathbf{Q}(z)\boldsymbol{Z}_j\boldsymbol{v}_j - \boldsymbol{u}^\top\mathbf{Q}_j(z)\boldsymbol{Z}_j\boldsymbol{v}_j\right)$$

$$= W_1 + W_2. \tag{155}$$

We first evaluate $W_1$. By the Woodbury matrix identity, we have

$$W_1 = \sum_{j=1}^n [\mathbb{E}_j - \mathbb{E}_{j-1}]\,\boldsymbol{u}^\top\mathbf{Q}(z)\boldsymbol{z}_j v_j = \sum_{j=1}^n [\mathbb{E}_j - \mathbb{E}_{j-1}]\frac{\boldsymbol{u}^\top\mathbf{Q}_j(z)\boldsymbol{z}_j v_j}{1+\boldsymbol{z}_j^\top\mathbf{Q}_j(z)\boldsymbol{z}_j}$$

$$= \sum_{j=1}^n [\mathbb{E}_j - \mathbb{E}_{j-1}]\beta_j\eta_j\boldsymbol{u}^\top\mathbf{Q}_j(z)\boldsymbol{z}_j v_j\Delta_j + \mathbb{E}_j\eta_j\boldsymbol{u}^\top\mathbf{Q}_j(z)\boldsymbol{z}_j v_j = \sum_{j=1}^n W_{1,1j} + W_{1,2j}, \tag{156}$$

since the identity $\beta_j = \eta_j + \beta_j \eta_j \Delta_j$ and the independence between $\boldsymbol{z}_j$ and $\mathbf{Q}_j$. By applying the inequality $\mathbb{E}_{j-1}|[\mathbb{E}_j - \mathbb{E}_{j-1}]x|^2 \leqslant 2\mathbb{E}_{j-1}|x|^2$, the Cauchy-Schwarz inequality $\mathbb{E}_j|xy| \leqslant \mathbb{E}_j^{\frac{1}{2}}|x|^2 \mathbb{E}_j^{\frac{1}{2}}|y|^2$, and the trace lemma (Bai & Silverstein, 1998, Lemma 2.7), we have

$$\sum_{j=1}^n \mathbb{E}_{j-1}|W_{1,1j}|^2 \leqslant \frac{2|z|^4}{\Im^4(z)} \sum_{j=1}^n v_j^2 \mathbb{E}_{j-1}^{\frac{1}{2}}|\boldsymbol{u}^\top \mathbf{Q}_j(z)\boldsymbol{z}_j|^4 \mathbb{E}_{j-1}^{\frac{1}{2}}|\Delta_j|^4 \leqslant \frac{C|z|^4}{n^2\Im^8(z)}, \tag{157}$$

where we use the bounds (Zhuang et al., 2025; Hachem et al., 2007) $|\eta_j(z)| \leqslant |z|/\Im(z)$, $|\beta_j(z)| \leqslant |z|/\Im(z)$, and the moment bound $\mathbb{E}_{j-1}|\Delta_j|^{2k} \leqslant \frac{C_k}{n^k}$. On the other hand, we have

$$\sum_{j=1}^n \mathbb{E}|W_{1,1j}|^{2k} \leqslant \frac{C_k|z|^{4k}}{\Im^{4k}(z)} \sum_{j=1}^n v_j^{2k} \mathbb{E}|\boldsymbol{u}^\top \mathbf{Q}_j(z)\boldsymbol{z}_j|^{2k}|\Delta_j|^{2k} \leqslant \sum_{j=1}^n v_j^{2k} \frac{C_k|z|^{4k}}{n^{2k}\Im^{8k}(z)}. \tag{158}$$

The evaluation for $\sum_j \mathbb{E}_{j-1}|W_{1,2j}|^2$ can be given by

$$\sum_{j=1}^n \mathbb{E}_{j-1}|W_{1,2j}|^2 \leqslant \frac{|z|^2}{\Im^2(z)} \sum_{j=1}^n v_j^2 \mathbb{E}_{j-1}\left|\boldsymbol{u}^\top \mathbf{Q}_j(z)\boldsymbol{z}_j\right|^2 \leqslant \frac{C|z|^2 \sum_{j=1}^n v_j^2}{n\Im^4(z)}. \tag{159}$$

Similar to (158), we can get $\sum_j \mathbb{E}|W_{2,2j}|^{2k} \leqslant C_k|z|^{2k}/(\Im^{4k}(z)n^k)$. Hence, according to the Burkholder inequality, we have

$$\mathbb{E}|W_1|^{2k} \leqslant C_k \sum_{i=1,2}\left[\mathbb{E}\left[\sum_{j=1}^n \mathbb{E}_{j-1}|W_{1,ij}|^2\right]^k + \sum_{j=1}^n \mathbb{E}|W_{1,ij}|^{2k}\right] \leqslant \frac{C_k(|z|^{4k} + |z|^{2k})}{n^k(\Im^{8k}(z) + \Im^{4k}(z))}. \tag{160}$$

For $k > 1$, the RHS of (160) is summable, which implies $W_1 \to 0$ almost surely by the Borel–Cantelli Lemma.

We then evaluate $W_2$, which can be given by

$$-W_2 = \sum_{j=1}^n [\mathbb{E}_j - \mathbb{E}_{j-1}]\beta_j \boldsymbol{u}^\top \mathbf{Q}_j(z)\boldsymbol{z}_j \boldsymbol{z}_j^\top \mathbf{Q}_j(z)\boldsymbol{Z}_j \boldsymbol{v}_j$$

$$= \sum_{j=1}^n [\mathbb{E}_j - \mathbb{E}_{j-1}]\eta_j \boldsymbol{u}^\top \mathbf{Q}_j(z)\boldsymbol{z}_j \boldsymbol{z}_j^\top \mathbf{Q}_j(z)\boldsymbol{Z}_j \boldsymbol{v}_j + \eta_j \beta_j \Delta_j \boldsymbol{u}^\top \mathbf{Q}_j(z)\boldsymbol{z}_j \boldsymbol{z}_j^\top \mathbf{Q}_j(z)\boldsymbol{Z}_j \boldsymbol{v}_j$$

$$= \sum_{j=1}^n W_{2,1j} + W_{2,2j}. \tag{161}$$

The term $W_{2,1j}$ can be written as

$$W_{2,1j} = \mathbb{E}_j \eta_j \left(\boldsymbol{u}^\top \mathbf{Q}_j(z)\boldsymbol{z}_j \boldsymbol{z}_j^\top \mathbf{Q}_j(z)\boldsymbol{Z}_j \boldsymbol{v}_j - \frac{1}{n}\boldsymbol{u}^\top \mathbf{Q}_j^2(z)\boldsymbol{Z}_j \boldsymbol{v}_j\right). \tag{162}$$

By the trace lemma and $\|\boldsymbol{v}_j\| \leqslant \|\boldsymbol{v}\| = 1$, we have

$$\mathbb{E}_{j-1}|W_{2,1j}|^2 \leqslant \frac{C|z|^2}{n^2\Im^6(z)}, \quad \mathbb{E}|W_{2,1j}|^{2k} \leqslant \frac{C_k|z|^{2k}}{n^{2k}\Im^{6k}(z)}, \tag{163}$$

which implies $W_{2,1j} \to 0$ almost surely. The relation $W_{2,2j} \to 0$ can be shown in a similar manner, and we omit the details.

Next, we prove that $\mathbb{E}\boldsymbol{u}^\top \mathbf{Q}(z)\boldsymbol{Z}\boldsymbol{v} = O(\frac{1}{\sqrt{n}})$. We can show that

$$\boldsymbol{u}^\top \mathbf{Q}(z)\boldsymbol{Z}\boldsymbol{v} = \sum_{j=1}^n \boldsymbol{u}^\top \mathbf{Q}(z)\boldsymbol{z}_j v_j = \sum_{j=1}^n \beta_j(z)\boldsymbol{u}^\top \mathbf{Q}_j(z)\boldsymbol{z}_j v_j. \tag{164}$$

Since $\mathbf{Q}_j(z)$ and $\boldsymbol{z}_j$ are independent, we have

$$\mathbb{E}\boldsymbol{u}^\top \mathbf{Q}(z)\boldsymbol{Z}\boldsymbol{v} = \mathbb{E}\sum_{j=1}^{n} v_j\left(\boldsymbol{u}^\top \mathbf{Q}_j(z)\boldsymbol{z}_j\right)(\beta_j - \eta_j)$$

$$= \sum_{j=1}^{n} v_j \mathbb{E}\boldsymbol{u}^\top \mathbf{Q}_j(z)\boldsymbol{z}_j \beta_j \eta_j \Delta_j. \tag{165}$$

Using the Cauchy-Schwarz inequality $\mathbb{E}|xy| \leqslant \mathbb{E}^{\frac{1}{2}}|x|^2 \mathbb{E}^{\frac{1}{2}}|y|^2$, we have

$$\left|\mathbb{E}\boldsymbol{u}^\top \mathbf{Q}(z)\boldsymbol{Z}\boldsymbol{v}\right| \leqslant \frac{|z|^2 |v_j|}{\Im^2(z)}\sum_{j=1}^{n} \mathbb{E}^{\frac{1}{2}}\left|\boldsymbol{u}^\top \mathbf{Q}_j(z)\boldsymbol{z}_j\right|^2 \mathbb{E}^{\frac{1}{2}}|\Delta_j|^2$$

$$\leqslant \sum_{j=1}^{n} \frac{C|z|^2}{n\Im^4(z)}|v_j| \leqslant \frac{C|z|^2}{\sqrt{n}\Im^4(z)}. \tag{166}$$

This completes the proof for the case $z \in \mathbb{C}_+$. The same approach can be extended to analyze the cases where $\Im(z) < 0$, $z < 0$, and $z > E_+$. Furthermore, for any $\epsilon > 0$ and $k > 0$, it can be shown that,

$$\mathbb{P}\left(\left|\boldsymbol{u}^\top \mathbf{Q}(z)\boldsymbol{Z}\boldsymbol{v}\right| \geqslant \epsilon\right) \leqslant C_{d,z}\epsilon^{-d}n^{-k}, \tag{167}$$

for all $d$ sufficiently large, where $C_{d,z}$ is a constant depending on $d$ and $\mathrm{dist}(z, \mathcal{M})$.

Next, we prove the convergence for a general compact set $\Gamma$. Since $\Gamma$ is compact, we can choose an $n^{-\frac{1}{2}}$-net $\mathcal{N}_n \subset \Gamma$ such that for every $z \in \Gamma$ there exists $z' \in \mathcal{N}_n$ with $|z - z'| \leqslant n^{-\frac{1}{2}}$, and for any two distinct points $x, y \in \mathcal{N}_n$ satisfy $|x - y| \geqslant n^{-\frac{1}{2}}$. Then, we have

$$\left|\boldsymbol{u}^\top \mathbf{Q}(z)\boldsymbol{Z}\boldsymbol{v} - \boldsymbol{u}^\top \mathbf{Q}(z')\boldsymbol{Z}\boldsymbol{v}\right| \leqslant \frac{\|\boldsymbol{Z}\|_2 |z - z'|}{\mathrm{dist}(\Lambda, z)\,\mathrm{dist}(\Lambda, z')}, \tag{168}$$

where $\Lambda = \{\lambda_j\}_{1 \leqslant j \leqslant p}$ are the eigenvalues of $\boldsymbol{Z}\boldsymbol{Z}^\top$. Inequality (168) gives

$$\sup_{z \in \Gamma}\left|\boldsymbol{u}^\top \mathbf{Q}(z)\boldsymbol{Z}\boldsymbol{v}\right| \leqslant \sup_{z \in \Gamma}\left|\boldsymbol{u}^\top \mathbf{Q}(z)\boldsymbol{Z}\boldsymbol{v} - \boldsymbol{u}^\top \mathbf{Q}(z')\boldsymbol{Z}\boldsymbol{v}\right| + \sup_{z' \in \mathcal{N}_n}\left|\boldsymbol{u}^\top \mathbf{Q}(z')\boldsymbol{Z}\boldsymbol{v}\right|$$

$$\leqslant \frac{\|\boldsymbol{Z}\|_2 \, n^{-\frac{1}{2}}}{\mathrm{dist}(\Lambda, \Gamma)\,\mathrm{dist}(\Lambda, \mathcal{N}_n)} + \sup_{z' \in \mathcal{N}_n}\left|\boldsymbol{u}^\top \mathbf{Q}(z')\boldsymbol{Z}\boldsymbol{v}\right| = X_1 + X_2. \tag{169}$$

According to the no-eigenvalue property (Bai & Silverstein, 1998, Theorem 1.1), we have $\mathrm{dist}(\Lambda, \Gamma) \to \mathrm{dist}(\mathcal{M}, \Lambda)$ almost surely as $p, n \to \infty$ and $\|\boldsymbol{Z}\|_2$ is almost surely bounded. Hence, $X_1 \to 0$ almost surely, as $p, n \to \infty$. By (167), for any $\epsilon > 0$ and $k > 2$, we can choose a $d$ sufficiently large such that

$$\mathbb{P}(X_2 \geqslant \epsilon) = \mathbb{P}\left(\bigcup_{z' \in \mathcal{N}_n}\left|\boldsymbol{u}^\top \mathbf{Q}(z')\boldsymbol{Z}\boldsymbol{v}\right| \geqslant \epsilon\right) \leqslant \sum_{z \in \mathcal{N}_n}\mathbb{P}\left(\left|\boldsymbol{u}^\top \mathbf{Q}(z)\boldsymbol{Z}\boldsymbol{v}\right| \geqslant \epsilon\right)$$

$$\leqslant \frac{\mathrm{card}(\mathcal{N}_n)C_{d,\Gamma}}{\epsilon^d n^k} \overset{(a)}{\leqslant} \frac{4(\sup_{x \in \Gamma}|x| + 1)^2 C_{d,\Gamma}}{\epsilon^d n^{k-1}}, \tag{170}$$

where $C_{d,\Gamma}$ is a constant depends on $d$ and $\mathrm{dist}(\Gamma, \mathcal{M})$. Step $(a)$ follows from the fact that, by the definition of $n^{-\frac{1}{2}}$-net, the disks of radius $1/(2\sqrt{n})$ centered at points in $\mathcal{N}_n$ are mutually disjoint. Since $k > 2$, the RHS of the above inequality is summable, which implies $X_2 \to 0$ almost surely by the Borel–Cantelli Lemma. Therefore, we have completed the proof. $\square$

# I    ADDITIONAL EXPERIMENTS

In this section, we provide supplemental experiments. The simulation settings, specifically the GD algorithm, are consistent with those described in Section 4.

**Learning Dynamics with Additional** $(p, n, \lambda)$ **Settings:** In Figure 4, we plot the learning dynamics with a broader range of triple $(p, n, \lambda)$. The settings here are consistent with those in Figure 1, and

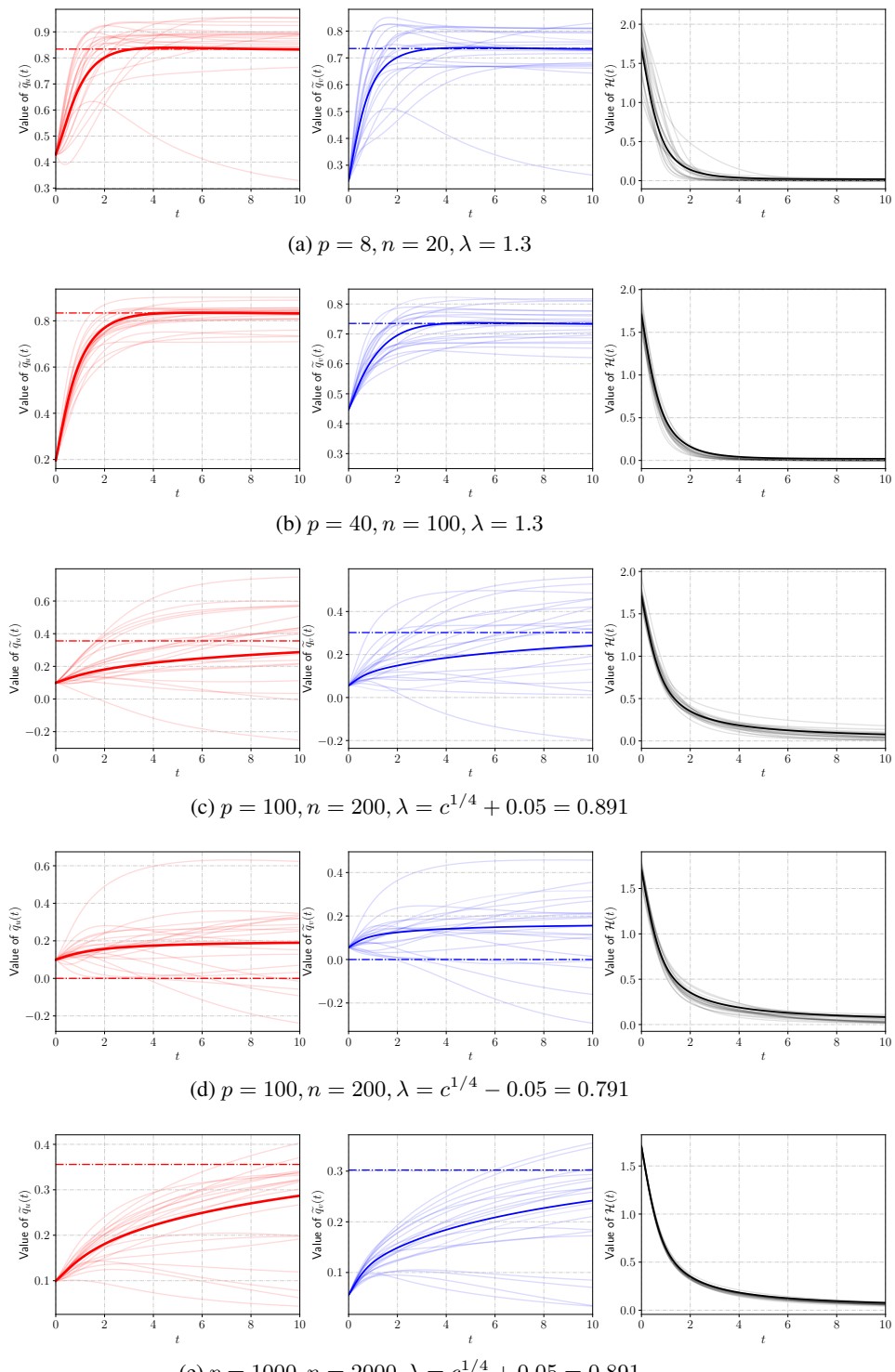

(a) $p = 8, n = 20, \lambda = 1.3$

(b) $p = 40, n = 100, \lambda = 1.3$

(c) $p = 100, n = 200, \lambda = c^{1/4} + 0.05 = 0.891$

(d) $p = 100, n = 200, \lambda = c^{1/4} - 0.05 = 0.791$

(e) $p = 1000, n = 2000, \lambda = c^{1/4} + 0.05 = 0.891$

Figure 4: Theoretical and Empirical Learning Dynamics with Different Values of $p, n, \lambda$.

the legends are omitted for brevity. From Figures 4a and 4b, we observe that as the dimensions increase, the deterministic approximations $\widetilde{q}_u$ and $\widetilde{q}_v$ become more accurate. Furthermore, it can be

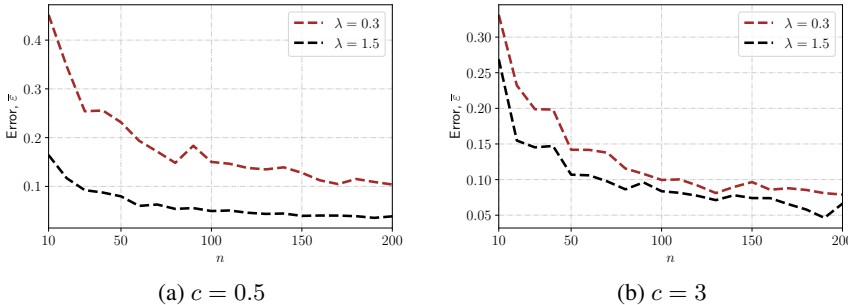

Figure 5: Approximation Error in Low Dimensions.

noted that, with high SNR, Theorem 1 is accurate even in low dimensions. However, the convergence of the deterministic approximation with low SNR is notably slower than that with high SNR.

In Figures 4c-4e, the parameters $\alpha_u$, $\alpha_v$, and $c$ are set to be the same. In Figure 4c and Figure 4e, the theoretical learning dynamics, i.e., the solid curves, are identical. It is shown that the convergence speed for the deterministic approximation is very slow, even for SNRs above the critical threshold. A comparison between Figure 4c and Figure 4d reveals that, despite the systems being in distinct phases, the learning dynamics curves closely resemble each other when their SNRs are similar. This suggests that signal detection may still be feasible even when the SNR is below the critical threshold.

**Approximation Error:** Figure 5 illustrates the error of the deterministic approximation in low dimensions. The initial points are set to satisfy $\alpha_u = \alpha_v = 0.447$. In Figure 5a, the ratio is set to $c = \frac{p}{n} = 0.5$, and in Figure 5b, the ratio is set to $c = \frac{p}{n} = 3$. The error is defined as $\bar{\varepsilon} = \sup_{1 \leqslant t \leqslant 10} \{|\widetilde{q}_u(t) - q_u(t)|, |\widetilde{q}_v(t) - q_v(t)|\}$. Here, the $y$-axis represents the average error $\bar{\varepsilon}$ computed over 50 independent trials. It can be observed that the error decreases as the dimensions $p$ and $n$ increase at the same pace. Additionally, the error is smaller when the SNR is high, which is consistent with the observation in Figures 1 and 4.

