# OpenReview forum: "Gradient Descent Dynamics of Rank-One Matrix Denoising"
_ICLR.cc/2026/Conference — ICLR 2026 Poster_

### Official Review · Reviewer_he9F · 2025-10-28

**Soundness:** 2
**Presentation:** 2
**Contribution:** 3
**Rating:** 4
**Confidence:** 3

**Summary:**

This paper studies the rank-one Jonstone's spiked model in matrix denoising problem within random matrix theorem (RMT) scenerio. The main contributions are two theorems. In theorem 1, they derive a closed-form deterministic approximation for the inner products between the learned vectors and the ground truth when ratio $\frac{p}{n}\to c>$. In theorem 2, they show that $c^{1/4}$ is the critical threshold for the gradient flow estimation.

**Strengths:**

- Use random matrix theory (RMT) derive a closed-form solution for the learning dynamics of matrix denoising problem. By RMT, naturally extend the matrix denoising problem to high dimensional scenario.
- The theorems and derivations are solid, and for example, the complex but precise expression in Theorem 1 is derived.
- The problem and assumptions are stated clearly.
- Provide a more comprehensive understanding of the dynamics of gradient-based learning in high-dimensional matrix problems

**Weaknesses:**

- Lack of explanation of the application of matrix denoising in the random matrix scenario, i.e. $\frac{p}{n}\to c$ with $p, n \to \infty$.
- The statement about computational complexity is too vague, like "the  complexity is affordable" on line 220 and "We note that $\hat{t}(\alpha_u,\alpha_v)$ can be efficiently computed by standard numerical methods." on line 244.
- On line 244, need more details of the so called "standard numerical methods".
- On line 42, "Extensive research has shown that" lacks reference.
- The theoretical work on which the article is based is very classic, and there is a lack of reference from recent new theoretical work.
- Lacks explanation of Riemannian gradient operator on line 144.
- In experiments, only one set of $(p,n)$ is tested, need more testing sets.
- In experients, the critical threshold is about 0.93, but the SNR is set to 0.3 and 1.5, which are far away from the threshold. More attempt on SNR need to be tested, especially the SNR near the threshold.- The ground truth is too trival.

**Questions:**

In Remark 1, the assumption of knowing the SNR $\lambda$ is practical in reality?In experiment, what will the results change if $p$ and $n$ are not that large, like $p=20$, i.e. beyond the RMT scenario.

---

> ### Author Response · Authors · 2025-11-21
> **Response to Reviewer he9F (1/2)**
>
> **We sincerely appreciate your time in reviewing the manuscript and your thoughtful feedback. We reply to your comments in the below.**
>
> > **W1:** Lack of explanation of the application of matrix denoising in the random matrix scenario, i.e. $\frac{p}{n} \to c$ with $p, n \to \infty$.
>
> **R-W1:** We would like to thank the reviewer for this comment. The reviewer is correct that our analysis is under on the framework of random matrix theory (RMT), which is asymptotic.  In fact, we consider high-dimensional setting, where the data dimension $p$ is not significantly smaller than the sample size $n$. We would like to clarify that, although the theoretical analysis describes the asymptotic (i.e., $p, n \to \infty$) performance of gradient descent (GD) based matrix denoising, the result is also accurate in finite dimensions (for example, when $n < 50$).  To demonstrate this, we have included additional experiments regarding the approximation accuracy. Due to the page limitation, we show the results in Appendix I and we kindly refer the reviewer to Figure 5 (lines 1967-1973) in the revised manuscript.
>
>
> > **W2:** The statement about computational complexity is too vague, like "the complexity is affordable" on line 220 and "We note that $\widehat{t}(\alpha_u, \alpha_v)$ can be efficiently computed by standard numerical methods." on line 244.
>
> **R-W2:** We would like to thank the reviewer for pointing out this. To make the statement more precise, we have provided the concrete computational complexity. Furthermore, following the same suggestion as in comment **W3**, we have provided the details for the numerical methods and added discussion on the complexity. We kindly refer the reviewer to lines 233-238 and lines 258-259.
>
>
>
> > **W3:** On line 244, need more details of the so called "standard numerical methods".
>
> **R-W3:** We would like to thank the reviewer for this comment. Following your suggestion, we have included additional details regarding the numerical methods to avoid confusion. We kindly refer the reviewer to lines 233-238.
>
> > **W4:** On line 42, "Extensive research has shown that" lacks reference.
>
> **R-W4:** Thanks for this comment. We have added the relevant works and references. Please refer to lines 044-045.
>
> > **W5:** The theoretical work on which the article is based is very classic, and there is a lack of reference from recent new theoretical work.
>
> **R-W5:** We would like to thank the reviewer for raising this important point. The reviewer is correct that our theoretical results are built upon classical theoretical frameworks. Specifically, for the random matrix component, our analysis primarily relies on the Bai-Silverstein method. As for the gradient flow analysis, it draws on classical methods from the theory of ordinary differential equations. It is worth noticing that although these mathematical tools are classical, they remain highly powerful and continue to be widely applied in contemporary research. To better illustrate this point, we have included additional recent references or works that utilize and extend these methods. Please refer to lines 044-045 and lines 087-089.
>
> > **W6:** Lacks explanation of Riemannian gradient operator on line 144.
>
> **R-W6:** We thank the reviewer for raising this important point. Following your suggestion, we have provided a detailed explanation of the Riemannian gradient flow. Specifically, we define the projection onto the tangent space and include relevant references to avoid confusion. Furthermore, we provide the proof on how the Riemannian gradient flow (4) preserves the unit norm constraint in Appendix C. We kindly refer the reviewer to lines 149-157 for more details.

---

> > ### Author Response · Authors · 2025-11-21
> > **Response to Reviewer he9F (2/2)**
> >
> > > **W7:** In experiments, only one set of $(p, n)$ is tested, need more testing sets.
> >
> > **R-W7:** We agree with the reviewer that testing a wider range of $(p, n)$ pairs would better demonstrate the accuracy of the analysis. To address this, we have conducted additional experiments (Figures 4 and 5) in Appendix I, which include various $(p, n)$ pairs. The results show that, with high SNR, the theoretical results are accurate even in small dimensions. We kindly refer the reviewer to lines 1886-1973 for detailed discussions.
> >
> > > **W8:** In experients, the critical threshold is about 0.93, but the SNR is set to 0.3 and 1.5, which are far away from the threshold. More attempt on SNR need to be tested, especially the SNR near the threshold.
> >
> > **R-W8:** We would like to thank the reviewer for this comment and we agree that the behavior of the learning dynamics near the critical threshold is particularly important and offers valuable physical insights. Following this suggestion, we have added new experiments (Figure 5 in Appendix I), where we examine the behavior of SNR near the critical threshold.  It is shown that even when the SNR is high ($\lambda > c^{1/4}$), as the SNR approaches the critical point, the convergence of the deterministic approximation slows down. When the SNR is slightly lower than the critical point, the dynamics suggest that there is still an opportunity to detect the signal. We kindly refer the reviewer to lines 1835-1912 for more details.
> >
> > > **W9:** The ground truth is too trival.
> >
> > **R-W9:**  We would like to thank the reviewer for this insightful comment. We agree with the reviewer that our ground truth $ ( \boldsymbol{u}^* , \boldsymbol{v}^* )  $ are two basis vectors and relative simple. Following the reviewer's comment, we have adopted a more complex ground truth structure (see lines 427-429) in Figure 1, where the components of $ \boldsymbol{u}^* $ and $ \boldsymbol{v}^* $ are sampled from Gaussian and Bernoulli distribution, respectively. The updated results in Figure 1 demonstrate that our theoretical analysis remains accurate with more complex ground truth signals.
> >
> > Moreover, it is worth noting that the specific method of generating the ground truth does not affect the theoretical analysis. This is because in our framework, we only assume that the ground truth signals $ \boldsymbol{u}^* $ and $\boldsymbol{v}^* $ have unit norm and  no prior information is required (i.e.,  non-Bayesian setting).
> >
> > > **Q1:** In Remark 1, the assumption of knowing the SNR $\lambda$ is practical in reality?
> >
> > **R-Q1:** We would like to thank the reviewer for this comment. In fact, our analysis indicates that the SNR affects the learning dynamics and different SNR levels correspond to different early stopping criteria.  In practice, knowing the exact SNR may be too restrictive. However, in many cases, we can determine the range of the SNR. For example, when using sensors or radar to sense the environment, we can estimate the range of the SNR. In such scenarios, the method in Remark 1 can also be extended. To clarify this point, we have added the relevant discussion in the revised manuscript and we refer the reviewer to lines 261-264.
> >
> > > **Q2:** In experiment, what will the results change if $p$ and are $n$ not that large, like, $p = 20$, i.e., beyond the RMT scenario.
> >
> > **R-Q2:** We would like to thank the reviewer for this insightful comment. The reviewer is correct that our analysis relies on the mathematical tools from RMT. The applicability of these asymptotic approximations to finite, lower-dimensional settings is indeed a central and valid concern in RMT. To illustrate the accuracy of the asymptotic analysis in finite dimensions, we have included numerical experiments with small $(p, n)$ pairs in Figures 4 and 5. It can be observed that even when the dimensions are not large, the analysis in Theorem 1 remains accurate.
> >
> > **Thank you once again for your utter generosity with your precious time and insightful comments.**

---

> > > ### Comment · Reviewer_he9F · 2025-11-24
> > > **Response to Rebuttal**
> > >
> > > Dear Authors:
> > >
> > > Thank you for your detailed responses, and for updating the manuscripts to reflect them. This rebuttal clears a lot of my confusions, and have made the contribution of your work much easier to follow.
> > >
> > > In light of it, I would raise my score to 6 and lean towards acceptance of this work.

---

> > > > ### Author Response · Authors · 2025-11-24
> > > > **Message to Reviewer he9F**
> > > >
> > > > Dear Reviewer he9F,
> > > >
> > > > Thank you for recognizing our contributions and for raising the score. We sincerely appreciate your invaluable comments, and all your time and efforts in helping us improve this work.

---

### Official Review · Reviewer_RZPg · 2025-10-29

**Soundness:** 4
**Presentation:** 3
**Contribution:** 4
**Rating:** 8
**Confidence:** 4

**Summary:**

The authors address the problem of rank-one matrix denoising, focusing on the gradient descent dynamics underlying this process. The main contribution of the paper lies in two theoretical results. The first theorem establishes a deterministic approximation for q_u and q_v, two widely used metrics that measure the alignment between the ground truth and the estimated components. The second theorem characterizes the asymptotic behavior of these deterministic approximations.

**Strengths:**

The paper is well written, well structured, and clearly explained. The theoretical analysis is rigorous, and the overall presentation is easy to follow. I find the work interesting and relevant to the study of optimization dynamics in low-rank estimation problems.

**Weaknesses:**

My only minor concern relates to the clarity of the experimental results. In Figure 2, which illustrates the effect of the critical SNR threshold, it is visually difficult to distinguish the different curves, especially in the middle subfigure. I understand that it is challenging to convey a large amount of information within a single figure, but since this result serves as an important validation of the theoretical findings, it should be presented more clearly to enhance its impact.

**Questions:**

N/A

---

> ### Author Response · Authors · 2025-11-21
> **Response to Reviewer RZPg**
>
> **We sincerely appreciate your time reviewing the manuscript and for your thoughtful feedback. We reply to your comments in the below.**
>
> > **W1:** My only minor concern relates to the clarity of the experimental results. In Figure 2, which illustrates the effect of the critical SNR threshold, it is visually difficult to distinguish the different curves, especially in the middle subfigure. I understand that it is challenging to convey a large amount of information within a single figure, but since this result serves as an important validation of the theoretical findings, it should be presented more clearly to enhance its impact.
>
> **R-W1:** We would like to thank the reviewer for raising this comment. Following your suggestion, we have replotted Figure 2 (lines 452-461) to improve the clarity of the experimental results and enhance the distinguishability of the curves.
>
> **Thank you once again for your utter generosity with your precious time and insightful comments.**

---

> > ### Comment · Reviewer_RZPg · 2025-11-27
> >
> > Thank you for the reply. I will keep my current positive score.

---

> > > ### Author Response · Authors · 2025-11-27
> > > **Message to Reviewer RZPg**
> > >
> > > Dear Reviewer RZPg,
> > >
> > > Thank you for recognizing our contributions and for the reply. We sincerely appreciate your invaluable comments, and all your time and efforts in helping us improve this work.

---

### Official Review · Reviewer_YymG · 2025-11-01

**Soundness:** 3
**Presentation:** 3
**Contribution:** 3
**Rating:** 6
**Confidence:** 3

**Summary:**

This paper studies the gradient-flow dynamics of the rectangular rank-one spiked matrix denoising problem in the high-dimensional limit.
Using tools from random matrix theory and Laplace transforms over the Marčenko–Pastur law, the authors derive explicit deterministic equations describing the evolution of the overlaps between the estimated and true directions.
They prove convergence to a deterministic limit, identify a BBP-type phase transition at the expected signal-to-noise threshold, and show that the limiting alignment carries the same sign as the initialization (“signed BBP”).
A kernel-based argument establishes existence and uniqueness of the solution.
Experiments nicely confirm the theoretical predictions.
Overall, the work extends the analysis of Bodin and Macris (2021), which treated the symmetric Wigner case, to the rectangular Wishart setting.

**Strengths:**

The results are technically sound and contribute to the understanding of optimization dynamics in high dimension.Extending the analysis from the symmetric to the rectangular setting is nontrivial and closes a natural gap in the literature. Conceretly:
- Provides explicit, analytic trajectories for gradient flow in the rectangular spiked model.
- The results are rigorous and match empirical observations very well.
- The “signed BBP” effect offers a clear dynamical interpretation of initialization dependence.
- Writing and figures are clear; the paper is enjoyable to read.
- Strengthens the theoretical link between random matrix theory and learning dynamics

**Weaknesses:**

- The work feels somewhat incremental relative to Bodin & Macris (2021); the main novelty lies in adapting the approach to the rectangular case.
- The “signed BBP” result, while nicely explained, is largely an expected property of continuous gradient flow.
- Experiments are limited to confirming the theory in the simplest setting; discrete-time or noisy gradient dynamics are not discussed.
- The discussion of related literature could be more complete and precise.

**Questions:**

I) Clarify the exact novelty relative to previous analyses.
The paper’s results are very close in spirit to Bodin & Macris (2021, arXiv:2105.12257), which already provided deterministic gradient-flow equations and asymptotic limits in the symmetric spiked Wigner setting.
It would help to spell out precisely which technical steps differ in the rectangular case and which parts of the proof had to be redone — for instance, changes in the resolvent structure, contour integration, or Laplace-transform kernel.
Are there specific mathematical obstacles that make the rectangular case substantially more difficult, or is it mainly a matter of replacing the semicircular law with the Marčenko–Pastur one?
A short paragraph clearly highlighting these differences would make the contribution much clearer.

II) Connection to the broader literature surveyed by Macris.
The “Related Work” section of Bodin & Macris (2021) already offers a remarkably complete overview of the theoretical ecosystem surrounding gradient descent, AMP, and high-dimensional inference. Many of those references are directly relevant here.
The Bayesian analyses of the spiked Wigner and tensor models (Korada & Macris 2009; Barbier et al. 2016; Lelarge & Miolane 2018; Lesieur et al. 2017; Perry et al. 2020) provide precise information-theoretic benchmarks in the form of mutual information and MMSE.
The dynamical behavior of AMP and the existence of computational-to-statistical gaps (Barbier et al. 2016; Lesieur et al. 2017) are also well understood.
Given this context, could the authors explain what new qualitative insight is gained from their explicit time-evolution formulas?
For instance, does the analytic expression for the transient trajectories reveal any phenomenon that is not already implicit in the AMP state evolution or in the energy-landscape picture?

III) Relation to the matrix–tensor and Langevin dynamics literature.
Macris notes that recent works (Sarao Mannelli et al., 2019; 2020) analyzed the optimization of mixed matrix–tensor inference problems using integro-differential Cugliandolo–Kurchan (CSHCK) equations — a fully dynamic, spin-glass-inspired formalism.
It would be interesting for the authors to comment on how their much simpler kernel/ODE formulation compares conceptually to those dynamical equations.
Does the present approach capture similar information about the saddle structure or convergence rates, but in a more tractable regime?
Or is it strictly a deterministic “mean-field” limit without the stochastic thermal components appearing in the CSHCK-type equations?

IV) Discrete gradient descent versus continuous gradient flow.
Since the work focuses entirely on continuous-time dynamics, a natural question is whether these results extend (even approximately) to discrete gradient descent with a finite learning rate.
Previous works, such as Lee et al. (2016) and Ge et al. (2017), established convergence for discrete GD under the “strict saddle” property, while Saxe et al. (2013) and Mei & Montanari (2019) studied learning-rate effects in linear and nonlinear models.
Could the authors discuss whether the signed-BBP phenomenon and transient behavior persist under discrete updates?
Even a conjectural statement or some preliminary numerical evidence would be welcome.

V) Connections to the energy landscape and spin-glass literature.
Macris also situates the work within the broader context of non-convex optimization in random energy landscapes (Subag & Zeitouni 2017; Ros et al. 2019; Auffinger et al. 2013).
It would be valuable to comment on whether the present gradient flow can be interpreted as traversing a spin-glass-like energy surface with a small number of global minima and exponentially many saddles.
Does the deterministic flow derived here correspond to the typical trajectory that avoids these saddles in the large-n limit?
Such a discussion would help bridge the current mathematical analysis with the well-developed physical intuition from statistical mechanics.

Overall, the paper is technically clean and the results are credible, but the authors could significantly increase its impact by situating it more deeply within the broad theoretical lineage summarized by Macris — including AMP and Bayesian limits, non-convex low-rank recovery, spin-glass Langevin dynamics, and deterministic gradient-flow analyses.
Clarifying what the present framework adds to that landscape, and where it could go next, would make the work more compelling for the ICLR audience.

---

> ### Author Response · Authors · 2025-11-21
> **Response to Reviewer YymG (1/5)**
>
> **We sincerely appreciate your time in reviewing the manuscript and your thoughtful feedback. We reply to your comments in the below.**
>
> > **W1:** The work feels somewhat incremental relative to Bodin & Macris (2021); the main novelty lies in adapting the approach to the rectangular case.
>
> **R-W1:** We would like to thank the reviewer for your comments. The reviewer is correct that (Bodin & Macris, 2021) is the most closely related work, and as mentioned in your comment **Q1**, our technical approach follows a similar spirit. We would like to clarify that generalizing from the Wigner model to the rectangular matrix model introduces significant challenges. Furthermore, we have also derived several novel theoretical results that are not available in the literature. We kindly refer the reviewer to our response **R-Q1** for the detailed discussion and the revision.
>
> > **W2:** The “signed BBP” result, while nicely explained, is largely an expected property of continuous gradient flow.
>
> **R-W2:** We would like to thank the reviewer for this insightful observation. The reviewer's intuition is correct as the signed BBP transition exhibits a similar behavior for both Wigner and rectangular models. As the reviewer also pointed out in comment **Q5**, this similarity stems from an analogous structure in the energy landscapes.
>
> We would like to emphasize that, despite this qualitative similarity, the specific initial conditions under which the dynamics converge to the saddle points are not obvious. Furthermore, while the property are analogous, the Wigner and rectangular models remain distinct, and the technical extension of the analysis to the rectangular case bring significant challenges. We kindly refer the reviewer to our response **R-Q1**.
>
> > **W3:** Experiments are limited to confirming the theory in the simplest setting; discrete-time or noisy gradient dynamics are not discussed.
>
> **R-W3:** We would like to thank the reviewer for this comment. We agree that analyzing the discrete-time version, i.e., gradient descent (GD), is very crucial as it corresponds to the algorithm implemented in practice. In  the revised manuscript,  we have studied the gap between continuous gradient flow and discrete GD   and run additional numerical experiments.  It is shown that gradient flow closely matches GD, and the theoretical analysis can be extended to GD. We kindly refer the reviewer to our reply **R-Q4** for more details.
> Regarding the case of gradient noise, we believe this is a highly meaningful and insightful research direction, especially given the prevalence of SGD. Following the reviewer's suggestion, we have included a discussion of relevant connections in the revised manuscript (see lines 290-294), while leaving a more detailed investigation for future work.
>
> > **W4:** The discussion of related literature could be more complete and precise.
>
> **R-W4:** We would like to thank the reviewer for pointing out this. We are also grateful for the valuable perspectives from different research areas and the insightful guidance regarding relevant literature that you provided in your comments. For more details, we kindly refer the reviewer to responses **R-Q2**~**R-Q5**.

---

> > ### Author Response · Authors · 2025-11-21
> > **Response to Reviewer YymG (2/5)**
> >
> > > **Q1:** Clarify the exact novelty relative to previous analyses.
> >
> > **R-Q1:** We would like to thank the reviewer for this constructive comment. The reviewer is correct that the spirit of our work is similar to that of (Bodin & Macris, 2021). However, we emphasize that some key technical steps differ from those in (Bodin & Macris, 2021), and compared to the Wigner model, the rectangular model brings novel challenges.
> >
> > - As the reviewer mentioned, the structure of the resolvent becomes more complicated for the rectangular model, and the time evolution involves correlation between the resolvent and the random matrix. We refer the reviewer to Lemma 4 (lines 804-816) and Eq. (43) (lines 877-882) in revised manuscript for more details. Due to the existence of the correlation, the proof is very challenging.
> >
> > - For the rectangular model, the system of Laplace transform equations is more complex. The poles are implicitly defined for several terms. For instance, in Eqs. (84) and (85) (lines 1218-1223), the zeros of the function  $ 1 - \lambda^2 s^2 m(s^2) \underline{m}(s^2) $  are non-obvious. For the Wigner case, we can directly use the explicit expression of the Stieltjes transform  $ m(z) $ . However, this method is not applicable to find the poles for the rectangular case.
> >
> > - When taking the Laplace inversion of high-order terms such as Eq. (89), the poles are of order 2 for the rectangular case (terms like  $ \mathcal{L}[f](s, z) / (s^2 - 4z) $ ). For the Wigner model, the order is of 1 (e.g.,  $ \mathcal{L}[f](s, z) / (s - 2z) $ ). In (Bodin & Macris, 2021), these terms are expressed by the convolution between  $ f(t,z) $  and  $ \mathcal{L}^{-1}[(s - 2z)^{-1}] = e^{2zt} $ , followed by contour integration over  $ z $ . However, in the rectangular model, the case becomes  $ \mathcal{L}^{-1}[(s^2 - 4z)^{-1}] = \sinh(2\sqrt{z}t) / (2\sqrt{z}) $ . The function  $ z^{-\frac{1}{2}} $  introduces a branch cut, significantly complicating the contour integration over  $ z $ . To overcome this, we use a "double Laplace transform" strategy. In particular, we apply an additional Laplace transform to  $ \mathcal{L}(f) $  to form  $ \mathcal{L}[f](r, s; z) $ , which has a decoupled structure and becomes analytically tractable.
> >
> > - We establish the existence and uniqueness of solutions (Theorem 3, lines 940-951) and demonstrate the integral representations of these solutions, which are not established in (Bodin & Macris, 2021). From a rigorousness viewpoint, we believe Theorem 3 is important because  it  ensures that the application of the Gronwall-type argument is valid, i.e., the two solutions are well-defined. Furthermore, by using Theorem 3, the proof for the convergence (lines 1350-1414) could be simplified.
> >
> > In summary, the results in this work are non-trivial and cannot be obtained by simply replacing the semicircular law with the Marčenko-Pastur distribution. To highlight the key technical differences and emphasize the contributions, we have expanded the discussion on resolvent analysis, and provided a more detailed explanation for the distinct pole structures and analytical properties of the differential equations in Remark 6 (lines 295-311).

---

> > > ### Author Response · Authors · 2025-11-21
> > > **Response to Reviewer YymG (3/5)**
> > >
> > > > **Q2:** Connection to the broader literature surveyed by Macris.
> > >
> > >
> > > **R-Q2:** Thank you for this insightful comment and for highlighting the profound body of works. These works provide an information-theoretic analysis and investigate the performance limits of efficient iterative algorithms like approximate message passing (AMP) under the Bayesian framework, where the ground-truth is drawn from a prior. Furthermore, they also determine the critical phase transition threshold below which the detection is impossible. These contributions offer deep insights into the low-rank signal recovery of high-dimensional data and the optimization landscapes of machine learning algorithms. In contrast, our work mainly focuses on a deterministic signal setting. That is, we aim to understand how fundamental algorithms like GD behave when no prior information is available. A natural conjecture is that the critical phase transition threshold  $ \lambda_{\mathrm{th}} $  in our work is higher than that of the Bayes-optimal detection, because we do not utilize prior information.
> > >
> > > We note that a similar angle is discussed in (Lelarge et al., 2017, Section 2.3). The interesting observation is that in the Wigner model, PCA is Bayes-optimal when the prior is Gaussian, but is not optimal under certain priors (e.g., Bernoulli). We conjecture that an analogous conclusion holds for the concerned rectangular model as well.
> > >
> > > Moreover, as the reviewer points out, the state evolution for AMP is defined recursively and involves expectations over general distributions and nonlinear functions. While the aforementioned works provide solid analysis of its fixed point, i.e., the large-time limit, we believe that a precise characterization of its finite-time trajectory is very challenging. In contrast, the trajectory of GD is analytically more tractable in our setting. Our closed-form evaluation also reveals that in scenarios with low SNR, early stopping can improve generalization performance. We believe this finding offers meaningful insights for broader machine learning algorithms, given the widespread use of GD.
> > >
> > > Following the reviewer's suggestions, we have revised the related work section to provide more comprehensive comparison and discussion of the connections between the works mentioned by the reviewer and ours. We have also added discussion regarding the new insights offered by the analysis of phase transition thresholds as compared to AMP. Please kindly refer to the introduction (lines 66-71) and Remark 9 (lines 396-398) for these updates.
> > >
> > > > **Q3:** Relation to the matrix–tensor and Langevin dynamics literature.
> > >
> > > **R-Q3:** We would like to thank the reviewer for this insightful comment regarding the connection to the CSHCK equations.  To the best of our understanding, both the CSHCK equations and our kernel ODEs describe the evolution of key quantities such as the inner product between the estimator and the ground truth in low-rank signal recovery.
> > >
> > > The CSHCK framework considers a more general matrix-tensor mixed model. However, the CSHCK equations involve the full time correlation (see  $ C(t. t') $  in (Sarao Mannelli et al, 2019, Eq. (7)) ) and require solving functional fixed-points. In contrast, as the reviewer mentioned, our ODE formulation is causal,  simpler, and admits a closed-form solution. Both frameworks can reveal phase transition phenomena, which implicitly suggest that GD can escape saddle points in high dimentional senarios. Our analysis thus provides a more tractable perspective on the dynamics.
> > >
> > > We believe the dynamics revealed in this work are not simply a zero-temperature limit of the CSHCK equations, but could potentially be unified with CSHCK under a broader framework. A key difference lies in how the unit-norm constraint is enforced. The prior works use a scalar multiplier (see  $ \mu(t) $  in [Sarao Mannelli et al, 2019, Eq. (7)]), which is uniform across components  $ x_j(t) $ s. A natural extension would consider the interaction among  $ x_j $ s, i.e.,  $ \boldsymbol{\mu}(t) $  becomes a vector. This takes the Riemannian gradient as a special case.
> > >
> > > Following your suggestion, we have added the corresponding discussion regarding the CSHCK framework in Remark 5 and we kindly refer  the reviewer to lines 283-294.

---

> > > > ### Author Response · Authors · 2025-11-21
> > > > **Response to Reviewer YymG (4/5)**
> > > >
> > > > > **Q4:** Discrete gradient descent versus continuous gradient flow.
> > > >
> > > > **R-Q4:** We thank the reviewer for raising this important point. Following your suggestion, we have quantitatively studied the gap between continuous gradient flow and discrete GD (under the Riemmanian optimazation framwork). In particular, we show that, for a given time horizon  $ [0, T] $  and learning rate  $ \eta $ , there exists a constant  $ C $  such that $  \max_{0 \leq k \leq [\frac{T}{\eta}]}   \max(\lVert \widetilde{\boldsymbol{u}} _ k - \boldsymbol{u} _ {k\eta} \rVert, \lVert \widetilde{\boldsymbol{v}}_k - \boldsymbol{v} _ {k\eta} \rVert) \leq C(\eta + \eta^2) $ . This implies that the behavior of GD can be effectively studied via the continuous process when  $ \eta $  is small. The details are provided in Remark 7 and  the proof is given in Appendix D  of the revised manuscript. We kindly refer the reviewer to lines 311-328.
> > > >
> > > > It is worth noticing that the signed BBP transition is theoretically achieved with  $ T \to \infty $ , which requires an infinitesimal learning rate and seems impractical. However, due to the continuity of the dynamics, the system approaches the global optimum (where the BBP occurs) within finite time. This suggests that the phenomenon should also persist under discrete updates for sufficient small learning rate. It is worth noticing that the curves labelled "Sim." in Figure 1 and the additional experiment in Figure 4, which are based on GD, align with the theoretical results. This alignment verifies that the BBP phenomenon is persist under GD. Following your suggestion, we have added the discussion regarding the BBP transition for discrete settings and we kindly refer the reviewer to line 377.
> > > >
> > > > > **Q5:** Connections to the energy landscape and spin-glass literature.
> > > >
> > > > **R-Q5:** We thank the reviewer for raising this comment and providing the perspective regarding the number of saddle points. The reviewer is correct that our loss function for the gradient flow can be viewed as a specific instance of the p-spin glass energy function discussed in these works. In particular,  (Auffinger et al., 2013, Theorem 2.2) demonstrated that for tensor model, the energy landscape contains an exponential number of critical points. However, for the matrix case, as discussed in (Auffinger et al., 2013, Remark 2.3), the number of saddle points grows linearly, since these saddle points correspond to the eigenvectors. In our analysis, we demonstrate that with random initialization on the sphere, the learning trajectory avoids these saddle points and converges to the global minimum in high dimensions. This is because, the learning dynamics exhibit BBP phase transition in large time limit. According to random matrix theory, the phase transition is associated with the top singular vector, which is the global minimum point, while  $ q_u(t), q_v(t) $  go to  $ 0 $  for saddles.
> > > >
> > > > Furthermore, we believe that for more complex tensor models with gradient flow, a similar property may hold. As the reviewer mentioned, the learning dynamics would avoid the exponentially many saddle points and still reach the global minimum. Following the reviewer's suggestion, we have included the discussion on the connection between the learning loss surface and the spin-glass energy landscape. We kindly refer the reviewer to lines 408-412.

---

> > > > > ### Author Response · Authors · 2025-11-21
> > > > > **Response to Reviewer YymG (5/5)**
> > > > >
> > > > > > **Overall:** Overall, the paper is technically clean and the results are credible, but the authors could significantly increase its impact by situating it more deeply within the broad theoretical lineage summarized by Macris — including AMP and Bayesian limits, non-convex low-rank recovery, spin-glass Langevin dynamics, and deterministic gradient-flow analyses. Clarifying what the present framework adds to that landscape, and where it could go next, would make the work more compelling for the ICLR audience.
> > > > >
> > > > > **R-Overall:** We would like to thank the reviewer for the recognition of our contribution and for providing broader perspectives, which not only enhanced insights of the theoretical results but also broadened the authors' knowledge, particularly in the field of statistical physics. We have realized the connections between spin glass systems and matrix signal processing, and have also gained insights into the measure of the saddle points in the energy landscapes of high-dimensional systems. In the future, we will further explore on this direction.
> > > > >
> > > > >
> > > > > **Thank you once again for your utter generosity with your precious time and insightful comments.**

---

### Official Review · Reviewer_DPd9 · 2025-11-04

**Soundness:** 3
**Presentation:** 3
**Contribution:** 2
**Rating:** 6
**Confidence:** 2

**Summary:**

The paper analyzes the statistical properties of gradient flow (as a proxy for gradient descent) for rank-one matrix denoising under a deformed Wishart model (rectangular matrices) with noise that has i.i.d. entries and . It derives a deterministic term for the limit of the inner products between limiting singular vector estimates and ground truth singular vectors in the asymptotic limit of matrix dimensions becoming infinite with fixed relation fraction. As an implication of these results, the authors are able to relatively accurately predict the behavior of gradient flow depending on the problem's signal-to-noise ratio (SNR) threshold akin to the BBP [Baik, Ben Arous, Péché 2005] phase transition. The results also can be used to quantify the dependence of the dynamics on initialization (value of $\alpha_u$/$\alpha_v$) and reasonable stopping times can be theoretically derived (see Remark 1 and Remark 2). From a technical perspective, the results lean on the analysis of [Bodin & Macris 2021], who have showed similar results for the symmetric case. Simulations are presented that substantiate the qualitative accuracy of the asymptotic analysis in the finite sample / matrix dimension case of $p$ and $n$ fixed.

**Strengths:**

The analysis presented in the paper seems to be new and studies a foundational problem in high-dimensional statistics / linear algebra, the behavior of singular value decomposition under the influence of noise in the case of rectangular matrices. The noise model is rather general, which is positive. It is of interest that the gradient flow dynamics more or less matches information theoretical phase transitions that are intrinsic to the problem.
While carefully checking many proofs in the appendix was beyond my abilities in the allocated time-frame as a reviewer, the results are plausible from a perspective of a reviewer who is familiar with tools for analyzing non-asymptotic high-dimensional problems.
Beyond covering the asymmetric case, some assumptions are weaker than in the related paper [Bodin & Macris 2021], such as the finite fourth moment assumption (as a opposed to assuming existence of all moments).

**Weaknesses:**

A fundamental weakness of the work is that it applies only in the high-dimensional limit of $\lim_{p, n \to \infty} p/n = c$, which is in contrast to many analyses of iterative algorithms in machine learning. Related to this issue, it can be pointed out that the title containing "Gradient Descent" is to a certain extent a misnomer as gradient flow, which is less relevant than gradient descent in practice in machine learning, is being analyzed. Thus, a lack of treatment of the discrete-time gradient descent method is a weakness of the paper given the framing of the paper.
A more unified discussion pointing out the differences and similarities between a power method algorithm for computing the leading singular vector pair and the presented algorithm would also have been insightful - I somewhat disagree with the framing that "SVD is intractable" as it is clear that a reasonable algorithm for the problem would involve a partial SVD implemented via randomized techniques [see, e.g., Martinsson, Tropp 2020].
Finally, it can be be pointed out that, while the asymmetric case being more challenging, the analyses / simulations presented are relatively close aligned to the ones of [Bodin, Macris 2021].

**Questions:**

1. In lines 141-145, it is mentioned that $\operatorname{grad](\cdot)$ is a "Riemannian gradient operator, which enforces the unit norm constraint". However, I do not see that the update equation of (4) enforces such a constraint. In some sense, this is statement is incompatible with the framework of Riemannian optimization as the Riemannian gradient lives in the tangent space onto Riemannian manifold and requires a retraction back onto the manifold (here, enforcing unit-norm vectors) to enforce the constraints.
Can you clarify or correct this discussion? In particular, how does your studied gradient flow algorithm enforce the unit norm constraints throughout its flow?

2. What are the limitations of the presented analysis for higher-rank ground-truths? Where does your current analysis fail to go through?

---

> ### Author Response · Authors · 2025-11-21
> **Response to Reviewer DPd9 (1/3)**
>
> **We sincerely appreciate your time in reviewing the manuscript and your thoughtful feedback. We reply to your comments in the below.**
>
> > **W1:** The weakness of this work.
>
> **R-W1:** We would like to thank the reviewer for providing many constructive comments, particularly those regarding the discrete and continuous processes, which addresses the essence of learning dynamics. We will respond to specific comments in the following.
>
> - *Large dimensional assumption:* The reviewer is correct that our analysis focuses on the high-dimensional regime where the data dimension  $ p $  is not too small compared to the sample size  $ n $ . We want to emphasize that the analysis in this work remains accurate in lower dimensions. Furthermore,  the constant  $ c $  can span a wide range (e.g.,  $ c = 0.01 $  or  $ 100 $ ) which covers many practical datasets such as MNIST (Couillet & Liao, 2022), where  $ n \approx 60000 $  and  $ p = 784 $  ( $ c \approx 0.013 $ ). To verify the accuracy of the analysis in low dimensions, we have included experiments with finite-dimensionalities. Please kindly refer to Figure 4 and Figure 5 (lines 1887-1973) in the revised manuscript.
>
> - *Gradient descent vs gradient flow:* We agree with the reviewer that the current work primarily focuses on the continuous gradient flow, instead of gradient descent (GD). Following your suggestion, we have qualitatively analyzed the gap between the discrete update and the continuous differential equation. In particular, we show that for a given time horizon  $ [0, T] $  and learning rate  $ \eta $ , there exists a constant  $ C $  such that $  \max_{0 \leq k \leq [\frac{T}{\eta}]}   \max(\lVert \widetilde{\boldsymbol{u}}_k - \boldsymbol{u} _ {k\eta} \rVert, \lVert \widetilde{\boldsymbol{v}}_k - \boldsymbol{v} _ {k\eta} \rVert) \leq C(\eta + \eta^2) $ . This implies that the behavior of GD can be effectively studied via the continuous process when  $ \eta $  is small. For example, the BBP transition indicates that with a small learning rate, GD with random initialization exhibits the capability of avoiding saddle points. The detailed discussion is provided in Remark 7 of the revised manuscript (lines 311-328), and the proof is given in Appendix E.
>
> - *Power iteration vs gradient algorithm:* The reviewer rightly points out that more efficient algorithms, such as power iteration or randomized methods, could be used to compute the top singular vectors effectively. To the best of the authors' understanding, power iteration and GD both iteratively solve for the top eigenvector, with power iteration converging faster under higher SNR. However, power iteration is a technique specifically  designed for matrices/tensors, while GD is a general framework widely used in learning. We believe that studying the dynamics of GD can provide more insights into high-dimensional optimization problems. Following your suggestion, we have revised the corresponding statements regarding SVD and we kindly refer the reviewer to lines 145-149.
>
> - *Comparison with earlier works:* The reviewer is correct that the spirit of our analysis is similar to the approach in (Bodin & Macris, 2021). However, we would like to clarify that, due to the unique challenges of the covariance model as compared with the Wigner model, several key steps of our analysis employ different techniques from (Bodin & Macris, 2021), requiring substantial new  derivations and leading to new theoretical results.
>     - The structure of the resolvent becomes more complicated for the covariance model, and the time evolution involves correlation between the resolvent and the random matrix. We refer the reviewer to Eq. (36) (line 756) in the revised manuscript for more details. To the best of our knowledge, the almost sure convergence established in this work is not yet available in the literature.
>     - We rigorously establish the existence and uniqueness of the governing integro-differential equation in Theorem 3, which is not studied in earlier work (Bodin & Macris, 2021).  This framework reveals analytical properties of the solution and makes the convergence proof more rigorous and streamlined.
>     - To obtain the closed-form solution for the covariance model is more challenging. This is because the order of the poles in the Laplace transform increase, and their locations are defined implicitly. The broader set of basis functions also complicates the study of the large-time limit of the learning dynamics.
>
>
>   To clarify the novel contributions, we have summarized the above-mentioned technical difference in detail in Remark 6 of the revised manuscript. We kindly refer the reviewer to lines 295-310.

---

> > ### Author Response · Authors · 2025-11-21
> > **Response to Reviewer DPd9 (2/3)**
> >
> > > **Q1:** In lines 141-145, it is mentioned that  $ \operatorname{grad}(\cdot) $  is a "Riemannian gradient operator, which enforces the unit norm constraint". However, I do not see that the update equation of (4) enforces such a constraint. In some sense, this is statement is incompatible with the framework of Riemannian optimization as the Riemannian gradient lives in the tangent space onto Riemannian manifold and requires a retraction back onto the manifold (here, enforcing unit-norm vectors) to enforce the constraints. Can you clarify or correct this discussion? In particular, how does your studied gradient flow algorithm enforce the unit norm constraints throughout its flow?
> >
> > **R-Q1:** We would like to thank the reviewer for the insightful comments. We want to emphasize that  $ \text{grad}(.) $  is defined on the tangent space of the product manifold  $  S^{p-1} \times S^{n-1} $ , thereby inherently preserving the unit-norm constraints of the vectors  $ \boldsymbol{u}_t $  and  $ \boldsymbol{v}_t $ . In fact, we can build up the ordinary differential equations (ODE) regarding  $ f_u(t) = \lVert\boldsymbol{u}_t\rVert^2 - 1 $  and  $ f_v(t) = \lVert\boldsymbol{v}_t\rVert^2 - 1 $  by (4) and show that  $ f_u(t) = f_v(t) = 0 $  for any  $ t > 0 $  when  $ \lVert\boldsymbol{u}_0\rVert^2 =  \lVert\boldsymbol{v}_0 \rVert^2 = 1 $ . In the revised manuscript, we have clarified this (lines 149-157) and provided the proof in Appendix C.
> >
> > Additionally, we fully agree with the reviewer that within the standard framework of Riemannian optimization, the Euclidean gradient should first be projected to the tangent space, followed by a retraction operator.  To our knowledge, this formulation specifically applies to the discrete-time update case. In the revised manuscript, we have added a discussion in Remark 7 regarding the Riemannian GD update, where we define the projection and retraction operations. We kindly refer the reviewer to lines 311-328. We also state in the experiments that the simulation of GD is iteratively updated following Eq. (17).
> >
> > It is worth mentioning that the dynamic in Eq. (4) is in continuous time. This flow requires the Euclidean gradient to be projected onto the tangent space. To avoid confusion, we have added a clarification following Eq. (4) and defined the projection operator. We kindly refer the reviewer to lines 155-156.

---

> > > ### Author Response · Authors · 2025-11-21
> > > **Response to Reviewer DPd9 (3/3)**
> > >
> > > > **Q2:** What are the limitations of the presented analysis for higher-rank ground-truths? Where does your current analysis fail to go through?
> > >
> > > **R-Q2:** We would like to thank the reviewer for this insightful question. We agree that the current results can not be directly applied to the high-rank case, but the framework can be extended to analyze the multi-rank (finite) case. For example, the conclusions in Theorem 3 (lines 940-951) regarding the existence, uniqueness, and integral representation of the solution to the differential equation remain applicable. However, the multi-rank analysis introduces challenges as shown below
> > >
> > > - The ''correlation'' terms between different signal sub-spaces arise, which significantly increases the complexity of deriving the closed-form evaluation. Specifically, consider the information-plus-noise model  $ \boldsymbol{X} = \boldsymbol{Z} + \sum _ {i=1}^r \lambda_i \boldsymbol{u} _ i^* (\boldsymbol{v} _ i^* ) ^ {\top}  $  with  $ \lambda_i \geq \lambda_{i+1} $ , and the loss function  $ \mathcal{H}(\boldsymbol{U}, \boldsymbol{V}) = \frac{1}{2} \lVert \boldsymbol{X} - \boldsymbol{U} \boldsymbol{V}^{\top} \rVert_{F}^2 $ , where  $ \boldsymbol{U} \in \mathbb{R} ^ {p \times r} $ ,  $ \boldsymbol{V} \in \mathbb {R}^{n \times r} $ , and  $ \boldsymbol{U}^\top \boldsymbol{U} = \boldsymbol{V}^{\top} \boldsymbol{V} = \boldsymbol{I}_r  $ . The gradient flow with respective to  $ \boldsymbol{U}_t = [\boldsymbol{u}^1_t, \ldots, \boldsymbol{u}^r_t] $  is thus given by  $ \frac{d \boldsymbol{U}_t}{dt} = (\boldsymbol{I}_p - \boldsymbol{U}_t \boldsymbol{U}_t^{\top}) \boldsymbol{X} \boldsymbol{V}_t $ . According to random matrix theory, each  $ \boldsymbol{u}^j_t $  should align with  $ \boldsymbol{u}^*_j $ . When analyzing the differential equation, the correlation terms such as  $ \langle \boldsymbol{u}^j_t, \mathbf{Q}(z) \boldsymbol{u}^i_t \rangle $  and  $ \langle \boldsymbol{u}^j_t,  \mathbf{Q}(z) \boldsymbol{Z} \boldsymbol{v}^{i}_t\rangle $ ,  $ 1 \leq i, j \leq r $  arise. Hence, to obtain the Laplace transforms (see Eq. (79), lines 1196-1206), a multivariate system must be solved, leading to highly complex pole structure.
> > > - The multiplicity of the signal subspace (i.e., the multiplicity of  $ \lambda_j $ s) may change the order of the poles, further complicating the derivation for the closed-form approximation and the large time limit analysis.
> > >
> > > We believe that, as a first attempt for the signal-spike rectangular matrix model, our results provide valuable insights regarding the dynamics of gradient-based matrix recovery. In the future, we plan to investigate more general cases. Following your comment, we have added related discussions in the future works in the revision.
> > >
> > > Additionally, when preparing the response to your comments, we noticed that a factor of 1/2 is missing in the loss function. We have carefully proofread and corrected this throughout the paper, and this adjustment does not affect the theoretical results.
> > >
> > >
> > > **Thank you once again for your utter generosity with your precious time and insightful comments.**

---

### Author Response · Authors · 2025-12-03
**Summary to the Area Chairs**

Dear Area Chairs,

**First, we would like to thank the Area Chairs and Reviewers for their efforts and time in handling our manuscript.**  The suggestions from the reviewers have helped us significantly improve the quality of this manuscript.

We are encouraged that the reviewers recognized this work for providing *comprehensive insights* into high-dimensional matrix estimation (Reviewers DPd9, YymG, RZPg, he9F). Their feedback also highlighted the *new theoretical results* (Reviewers DPd9, YymG) and *strengthened link* between random matrix theory and learning dynamics (Reviewers YymG, he9F). We also sincerely appreciate their acknowledgment of the *solid mathematics* (Reviewers DPd9, YymG, RZPg, he9F) and the *well-written* presentation of the manuscript (Reviewers YymG, RZPg, he9F).


In the following, we provide a brief summary of the responses to reviewers' concerns and key changes made in the revised manuscript for your easy reference:

  1. *Comparison with related work* (Reviewers  DPd9,  YymG): To clarify the distinctions from prior work and highlight the technical contributions, we have provided a detailed comparison with (Bodin & Macris, 2021) in Remark 6. In particular, the resolvent structure for the rectangular model is fundamentally different from the symmetric Wigner model (Bodin & Macris, 2021), and we derived new convergence results in Lemma 4. We also analyzed the existence, uniqueness, and integral representation of solutions to the key differential equations in Theorem 3, which were not investigated in the previous work and have broader applications.
  2. *Continuous vs. discrete process* (Reviewers DPd9,  YymG): To demonstrate the practical applicability of the closed-form analysis, we have included a bound on the distance between the solution of gradient flow and gradient descent in Remark 7. This bound indicates that the deterministic approximation for the continuous random process in this work could be extended to discrete gradient descent with small learning rates.
  3. *Accuracy in low dimensions* (Reviewers DPd9, he9F): To demonstrate the accuracy of the theoretical analysis in low dimensions, we have added related numerical simulations in Figure 4 and plotted the error curves in Figure 5.
  4. *Discussion of additional insights* (Reviewers YymG, he9F): To situate the insights within the broader literature, we have compared the analysis of this work with that for approximate message passing in the introduction and discussed its connection to the Langevin dynamics of spin-glass systems in Remark 5.
  5. *Clarification on dynamic equations* (Reviewers DPd9, he9F): To avoid confusion, we have rewritten the Riemannian gradient flow equation (4) and included a proof for the unit norm constraint in Appendix C.
  6. *Clearer figure plotting* (Reviewer RZPg): To enhance clarity, we have replotted Figure 2.

During the discussion period, Reviewer he9F indicated that the revisions addressed their concerns, and raised the score to 6. Reviewer RZPg recognized our contributions and maintained the score of 8.

**Thank you once again for your efforts and we hope this summary will assist you for your assessment.**

---

### Meta-Review · Area_Chair_QkLG · 2026-01-07

**Summary:**

This paper looks at the learning dynamics for doing least squares denoising to recover rank 1 signal. The reviewers agree that the paper is well written, with interesting rigorous mathematical results.

Initially, the reviewers had a quite a favorable view of the paper with scores of 6 (DPd9), 6 (YymG), 8 (RZPg), and 4 (he9F). This opinion only seems to have increased after the rebuttal with the reviewer who gave it a 4 acknowledging that their concerns had been addressed.

The main concerns seem to be three fold.

1. Setting of the work is asymptotic and is for gradient flow not gradient descent

2. The comparison with prior work in the literature is incomplete and the results are potentially incremental compared to Bodin and Marcis (2021)

3. The experiments are simplistic and do not test the boundaries of the theory.

**Reviewer Concerns:**

The asymptotic setting is standard and I believe that the rebuttal adequately addresses this concern. In terms of comparisons with Bodin and Marcis, I believe that the rebuttal in response to reviewers DPd9 and YymG sufficiently address this concern and clearly demonstrate the novelty compared to Bodin and Marcis. Additionally, the new experiments that have been added address the experimental concerns with the paper as well.

As such I believe the majority of concerns have been addressed and all reviewers are in favor of acceptance.

Finally, for the authors’ convenience in updating the related-work discussion, here are a few additional papers that study closely related rank-1 matrix denoising / (linear) denoising autoencoder dynamics. This list is non-exhaustive, and the papers below (and their citation graphs) may point to further relevant work.

[1] Arnu Pretorius, Steve Kroon, and Herman Kamper. Learning Dynamics of Linear Denoising Autoencoders.
In Proceedings for the 35th International Conference on Machine Learning, 2018.

[2] Ham, Jonghyun, Maximilian Fleissner, and Debarghya Ghoshdastidar. "Impact of Bottleneck Layers and Skip Connections on the Generalization of Linear Denoising Autoencoders." arXiv preprint arXiv:2505.24668 (2025).

[3] Cui, Hugo, and Lenka Zdeborová. "High-dimensional asymptotics of denoising autoencoders." Advances in Neural Information Processing Systems 36 (2023): 11850-11890.

**Reviewer Scores:**

The reviewers started off with scores of 8, 6, 6, 4. The reviewer with a score of 4 acknowledged that their concerns have been resolved and was willing to increase their score. The reviewer with a score of 8 also said they would maintain their score. As such I think this paper is a clear accept.

---

### Decision · Program_Chairs · 2026-01-26

Accept (Poster)